# Cellulose nanofiber-mediated manifold dynamic synergy enabling adhesive and photo-detachable hydrogel for self-powered E-skin

Lei Zhang [1,4], Lu Chen[2,4], Siheng Wang [1,2], Shanshan Wang[3], Dan Wang[1], Le Yu[2], Xu Xu[3], He Liu [1] ✉ & Chaoji Chen [2] ✉

Self-powered skin attachable and detachable electronics are under intense development to enable the internet of everything and everyone in new and useful ways. Existing on-demand separation strategies rely on complicated pretreatments and physical properties of the adherends, achieving detachable-on-demand in a facile, rapid, and universal way remains challenging. To overcome this challenge, an ingenious cellulose nanofiber-mediated manifold dynamic synergy strategy is developed to construct a supramolecular hydrogel with both reversible tough adhesion and easy photodetachment. The cellulose nanofiber-reinforced network and the coordination between Fe ions and polymer chains endow the dynamic reconfiguration of supramolecular networks and the adhesion behavior of the hydrogel. This strategy enables the simple and rapid fabrication of strong yet reversible hydrogels with tunable toughness (($Value_{max}$-$Value_{min}$)/$Value_{max}$ of up to 86%), on-demand adhesion energy (($Value_{max}$-$Value_{min}$)/$Value_{max}$ of up to 93%), and stable conductivity up to 12 mS cm$^{-1}$. We further extend this strategy to fabricate different cellulose nanofiber/$Fe^{3+}$-based hydrogels from various biomacromolecules and petroleum polymers, and shed light on exploration of fundamental dynamic supramolecular network reconfiguration. Simultaneously, we prepare an adhesive-detachable triboelectric nanogenerator as a human-machine interface for a self-powered wireless monitoring system based on this strategy, which can acquire the real-time, self-powered monitoring, and wireless whole-body movement signal, opening up possibilities for diversifying potential applications in electronic skins and intelligent devices.

Existing skin-attachable electronics with autonomous powering ability are desired for obtaining accurate and reliable biological/physical information and can be reversibly attached to arbitrary surfaces and detached without leaving residues[1,2]. Thus the utilization of reversible adhesion hydrogels is of great significance for self-powered electronic skins (e-skins)[3,4]. In the past few years, great efforts have been devoted to realizing the reversible adhesion of hydrogel with a variety of hard and soft materials[5,6]. Reversible adhesion can be achieved using chemical connection consisting of reversible bonds[7,8], including dynamic covalent bonds, noncovalent bonds with specific chemical groups, and

physically topological entanglement through external stimuli, such as pH[9], temperature[10], current[11], and rays[12]. Suo's group[7] developed a smart solution adhesive, where the spreading of an aqueous solution of polymer chains on the surface of two adherends to crosslink them together for topological adhesion. To further shorten the interfacial adhesion time and afford the effective interfacial linking between the bulk hydrogel and adherends to transmit force and elicit energy dissipation, there is a need to design a dynamic adhesion strategy with quick and strong long-term adhesion while leaving a window period for reversible easy detachment.

In this work, we successfully constructed a supramolecular hydrogel with both reversible tough adhesion and easy photodetachment via an ingenious cellulose nanofiber (CNF)-mediated manifold dynamic synergy strategy (Fig. 1a, b). The comprehensive mechanical properties and cohesive strength of the supramolecular hydrogel can be facilely tuned through UV light-driven photo-Fenton-like (P.F.) reaction between CNF networks and $Fe^{3+}$ in the hydrogel

material (Fig. 1c, d). In this way, the interfacial toughness and viscoelasticity can be controlled facilely in a detachable-on-demand way. Combined with the rapid, reliable, and reproducible regulating method, as well as adhesion-on-demand properties based on supramolecular network engineering, the developed convertible hydrogel holds great potential for practical application in photo-detachable self-powered e-skins.

## Results

### Design strategy and photo-detachable mechanism of the CNF-DA/PAA@$Fe^{3+}$ hydrogel by the light-driven supramolecular network engineering

Cellulose—the most abundant polysaccharide on earth, featuring high mechanical strength, renewability, and biocompatibility, has been regarded as a multifunctional building block for developing high-performance materials[13,14]. Additionally, TEMPO-oxidized CNF and poly(acrylic acid) (PAA) chains with position-selective and abundant

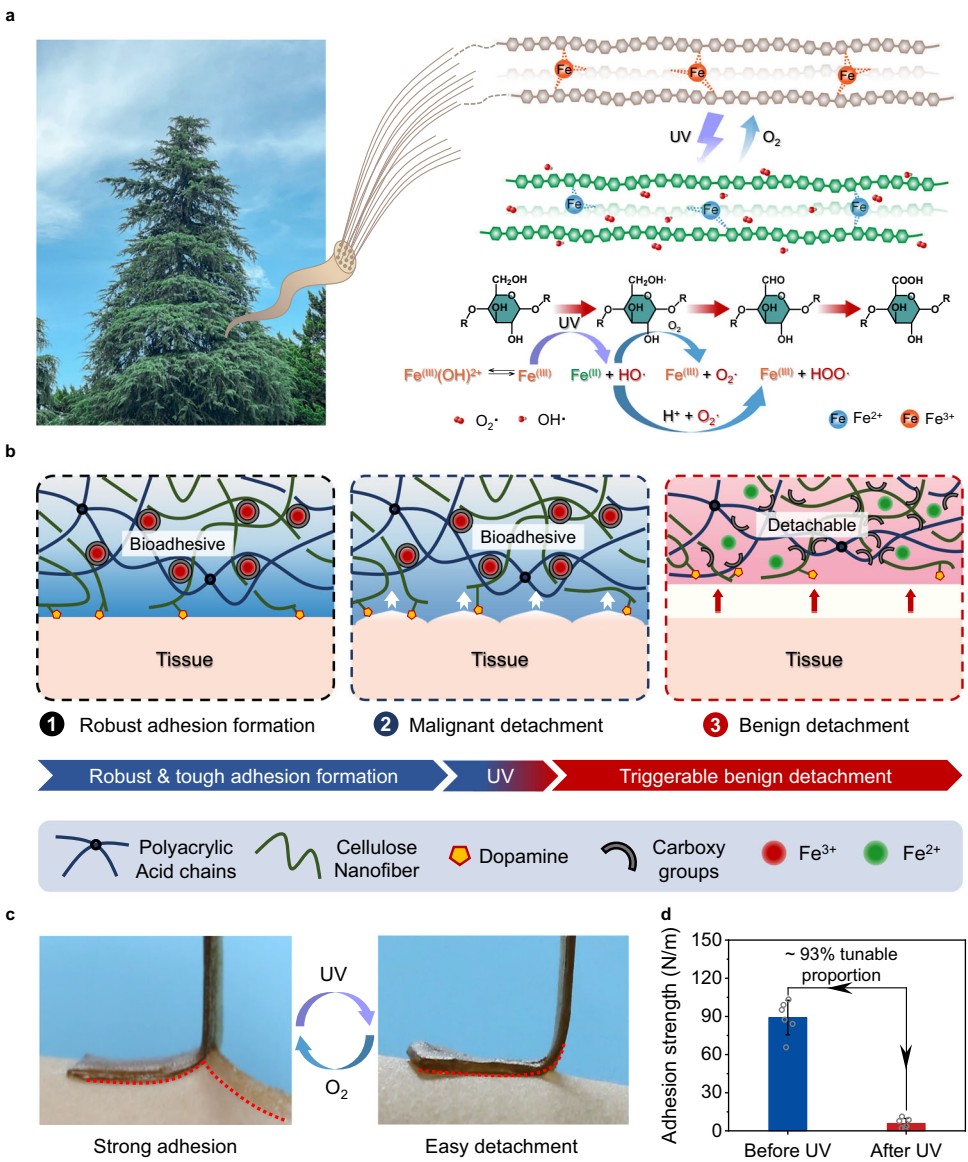

**Fig. 1 | Design of CNF-DA/PAA@$Fe^{3+}$ dynamic hydrogel based on a light-driven supramolecular network engineering strategy via photo-Fenton-like (P.F.) reaction. a** Schematic illustration of the design of the strong adhesion and photo-triggered detachment mechanism based on the P.F. reaction. **b** Schematic diagrams of a reversible structure of the adhesive and photo-detachable dynamic hydrogel with light-driven supramolecular network. **c** Photographs of the molecular switch being peeled off from human skins under UV irradiation and air oxidation. **d** Comparison of the adhesive strength of photo-detachable hydrogel before and after UV irradiation. Values in **d** represent their means ± SDs from *n* = 6 independent samples. Source data are provided as a Source Data file.

carboxy groups on the surface enable their coordination complexation with $Fe^{3+}$ ions. The CNF-reinforced supramolecular network, strong coordination interactions between $Fe^{3+}$ and polymer chains, as well as the dopamine (DA) adhesive groups collaboratively endow the excellent comprehensive mechanical properties and cohesive strength of the fabricated CNF-DA/PAA@$Fe^{3+}$ hydrogel. As directly exposed to UV light, $Fe^{3+}$ ions are reduced to $Fe^{2+}$ ions based on the P.F. reaction in the hydrogel system, where the CNF serves as a green Fenton-like reagent, leading to dissociation and reconfiguration of the Fe ions and CNF networks, thereby making the self-regulatable adhesive behavior of the hydrogel for on-demand photodetachment (Fig. 1b and Supplementary Figs. 1, 2). More attractively, by varying the exposure time in the air, we achieve the switchable operability of the above processes with associated reversible adhesion, holding great potentials for the reuse of the light-driven engineered hydrogel. Benefiting from the rich multi-type oxygen-containing functional groups of CNF chains and excellent mechanical properties, the hydrogel can stick quickly and conforms perfectly to the subject's skin surface as well as mild triggerable benign detachment with no visible residue and redness on the skin surface when exposed to the human-friendly UV irradiation (Fig. 1c, Supplementary Fig. 3, and Supplementary Movie 1).

The CNF-DA/PAA@$Fe^{3+}$ hydrogel with 2% $Fe^{3+}$ content was demonstrated to deliver the best comprehensive performance. All data analysis is based on the CNF-DA/PAA@$Fe^{3+}$ hydrogel sample with 2% $Fe^{3+}$ content (the mass fraction relative to the mass of CNF-DA is 2 wt%) and 80% water content (the mass fraction relative to the mass of the hydrogel is 80 wt%) if not specified otherwise (Supplementary Figs. 4–6). External properties of materials are strongly linked to their subtle structural changes. The confocal laser scanning microscope (CLSM) images and scanning electron microscopy (SEM) images of the CNF-DA/PAA@$Fe^{3+}$ hydrogel exhibited remarkable and flexible morphology transformation from the initial entangled and homogeneous dense microstructure (without UV irradiation) to a subsequently loose porous zone on its surface (after UV irradiation), and finally to a dense and compact microstructure again (after air-oxidation) (Fig. 2a and Supplementary Fig. 7). For facilitating the evident visualization of this structural transformation process, we performed in situ Raman analysis of the CNF-DA/PAA@$Fe^{3+}$ hydrogel by switching UV light on one side of the non-UV irradiated hydrogel during the dynamic introduction of UV light (Fig. 2b and Supplementary Fig. 8). Upon the UV irradiation, the peak intensity at 1680 $cm^{-1}$ (belong to carboxyl groups) exhibited a distinct enhancement, with the initial blue area (with low peak intensity value) being turned into green and completely to red (with high peak intensity value), indicative of the generation of more free carboxyl groups and thus the noticeable looser structure in the CNF-DA/PAA@$Fe^{3+}$ hydrogel triggered by UV light. As the hydrogel was placed in the air, the hydrogel showed an opposite trend, suggesting its reversible and tunable network structure. These results were also demonstrated by combined usage of ultraviolet-visible spectroscopy (UV-vis) spectra, small-angle X-ray scattering (SAXS), and X-ray diffraction (XRD) analysis of the CNF-DA/PAA@$Fe^{3+}$ hydrogel (Fig. 2c, d and Supplementary Figs. 9 and 10). Such a striking morphological transformation is associated with the significant differences in the interactions of CNF chains, $Fe^{3+}$ ions, and PAA chains, and thus supramolecular network evolution of the hydrogel triggered by UV irradiation and air oxidation[15–18].

To reveal the inherent driving mechanism of the structural transformation of the photo-detachable hydrogel based on the P.F. reaction, we carried out the X-ray photoelectron spectroscopy (XPS) analysis of the hydrogel during the UV irradiation and air-oxidation processes (Fig. 2e). The appearance of the peak at 710.6 eV of the hydrogel after UV irradiation demonstrated the generation of $Fe^{2+}$ in the system, verifying that $Fe^{3+}$ was partially reduced to $Fe^{2+}$ triggered by UV light, in which two free radicals ($O_2 \cdot^-$ and $HO \cdot^-$) also participate in the P.F. reaction process (Supplementary Fig. 11). As the photo-

detachable hydrogel was exposed to air, the $Fe^{2+}$ ions were partially oxidized to $Fe^{3+}$ ions, facilitating the self-recovery of the supramolecular network in the hydrogel. To identify the functional groups that participated in the photodetachment process, the two-dimensional correlation spectra (2D-COS) were employed to enhance FTIR spectra analysis of the hydrogel under UV irradiation (Fig. 2f, g)[19]. The cross-peak (1682, 3200) in the synchronous and asynchronous maps was both positive. According to Noda's judging rule, −COOH and −OH groups were the dominant functional groups that participated in the UV irradiation process, in which −COOH groups of CNF-DA chains were more susceptible to UV light. Additionally, the solid-state $^{13}C$ NMR spectra of the CNF-DA/PAA@$Fe^{3+}$ hydrogel exhibited that two characteristic peaks at 170–200 ppm, both assigned to −C=O, do appear in the hydrogel after UV irradiation, indicative of the environment variation of −COOH groups and the generation of −CHO in the hydrogel triggered by UV light (Fig. 2h and Supplementary Fig. 12)[20–22]. These results jointly revealed that $Fe^{3+}$ ions were transformed into $Fe^{2+}$ ions stimulated by UV light, and the coordination interactions of Fe ions with CNF-DA and PAA chains were resultantly dissociated; Subsequently, as the resultant hydrogel was exposed to air, the $Fe^{2+}$ ions were oxidized to $Fe^{3+}$ ions, leading to the reconfiguration of supramolecular network of the CNF-DA/PAA@$Fe^{3+}$ hydrogel in air. Along with this process, UV light acted as a functional factor in the P.F. reaction between CNF-DA chains and $Fe^{3+}$ ions, which triggered the valence state transformation of Fe ions in the system. Meanwhile, CNF served as a reducing agent in the P.F. reaction, which further contributed to the reversibility and stability of the coordination interactions in the CNF-DA/PAA@$Fe^{3+}$ hydrogel.

We further carried out a Gaussian simulation to investigate the dynamic transformation of the supramolecular network of the hydrogel based on the P.F. reaction process (Fig. 2i). The binding energy between $Fe^{3+}$ and −COOH on the CNF-DA chains was calculated to be −122.715 kcal/mol, which was lower than that between $Fe^{3+}$ and −OH on the CNF-DA chains (−101.121 kcal/mol), demonstrating that $Fe^{3+}$ ions preferentially coordinated with −COOH groups rather than −OH groups in the system (Supplementary Fig. 13). After UV irradiation, with $Fe^{3+}$ ions being partially reduced to $Fe^{2+}$ ions, the binding energy between $Fe^{2+}$ and −COOH groups was exactly higher than that between $Fe^{3+}$ and −COOH groups, facilitating the desirable dissociation of coordination complexes. These thermogravimetrics were in good agreement with the experimental results data. These experimental and theoretical simulation results confirm the strong coordination interactions between $Fe^{3+}$ and −COOH groups and subsequently UV light-induced supramolecular network reconstruction due to Fe ions valence transition based on the P.F. reaction.

## Mechanical performance of the CNF-DA/PAA@$Fe^{3+}$ hydrogel

Benefiting from strong coordination interactions between Fe ions and carboxyl groups, as shown in Fig. 3a, the CNF-DA/PAA@$Fe^{3+}$ hydrogel can be cyclically bent on fingers without visible fragmentation, suggesting its extraordinary flexibility and mechanical compliance. In addition, the CNF-DA/PAA@$Fe^{3+}$ hydrogel can be stretched 10 times under external force without breaking. Moreover, the CNF-DA/PAA@$Fe^{3+}$ hydrogel can still maintain its good flexibility even after being exposed to air for a long time of 96 h. Those combined demonstrate its strong stretchability, durability, and stability (Fig. 3b, Supplementary Figs. 14 and 15). The fabricated hydrogel, taking advantage of the dynamic and reversible metal coordination interactions, can still restore the original intact shape after being torn apart, suggesting its excellent self-healing performance (Fig. 3c and Supplementary Fig. 16).

To quantitatively demonstrate the mechanical properties of the obtained hydrogel, we conducted tensile and compressive stress-strain tests of the CNF-DA/PAA@$Fe^{3+}$ hydrogel before and after UV light stimulus. As shown in Fig. 3d, the introduction of $Fe^{3+}$ ions and

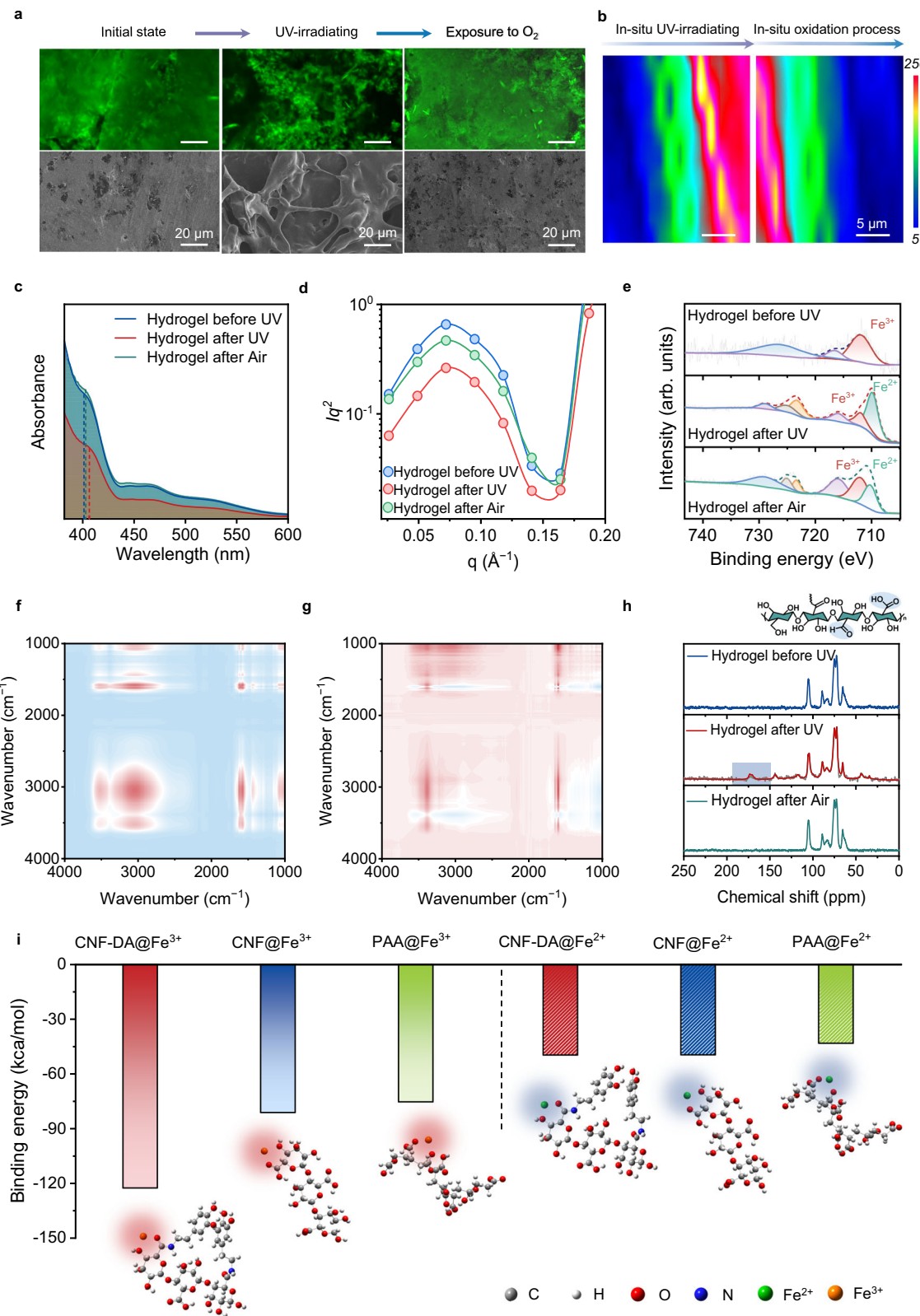

cellulose nanofibers, as well as their derivatives, can significantly enhance the stretchability of PAA-based hydrogels. Among them, especially, the CNF-DA/PAA@Fe$^{3+}$ hydrogel exhibited the maximum tensile stress of 0.053 MPa at a fracture strain of 1425%, and the maximum tensile stress and fracture strain was 5 times higher than those of the PAA hydrogel, respectively, demonstrating excellent mechanical properties of the CNF-DA/PAA@Fe$^{3+}$ hydrogel. Exposure to UV light, as

we expected, led to a drop in the tensile strength and fracture strain of the CNF-DA/PAA@Fe$^{3+}$ hydrogel by about 66% and 78%, respectively, owing to the dissociation of coordination complexes in the resulting hydrogel triggered by UV irradiation (Fig. 3e, Supplementary Figs. 17 and 18). More intuitively, the slight rise of the hydrogel with a 100 g weight suspending at its bottom after UV irradiation in the inset image demonstrated the degradation of its stretchability. Meanwhile,

**Fig. 2 | Photo-detachable mechanism of the CNF-DA/PAA@Fe³⁺ hydrogel by the light-driven supramolecular network engineering. a** CLSM images and SEM images of the CNF-DA/PAA@Fe³⁺ hydrogel during the UV irradiation and air-oxidation process showing the change of microstructures of the photo-detachable hydrogel. Scale bar, 20 μm. **b** Two-dimensional Raman image from the −COOH intensity (1680 cm⁻¹) of the CNF-DA/PAA@Fe³⁺ hydrogel during the UV irradiation and air-oxidation process. Scale bar, 5 μm. **c** UV-vis spectra of the CNF-DA/PAA@Fe³⁺ hydrogel during the UV irradiation and air-oxidation process. **d** SAXS curves for the photo-detachable hydrogels during the UV irradiation and air-oxidation process. **e** XPS of Fe 2*p* regions of the CNF-DA/PAA@Fe³⁺ hydrogel during the UV irradiation and air-oxidation process. **f**, **g** 2DCOS synchronous and asynchronous spectra generated from the FTIR spectra during the UV irradiation and air-oxidation process. In 2DCOS spectra, the warm color (red) represents positive intensities, while the cold color (blue) represents negative intensities. **h** ¹³C NMR spectra analysis of the CNF-DA/PAA@Fe³⁺ hydrogel during the UV irradiation and air-oxidation process. The illustration is the chemical structure of CNF, the blue shaded part is the carbonyl group (−C = O) during UV irradiation and air-oxidation process. **i** Binding energy between Fe³⁺ or Fe²⁺ and carboxyl groups in the PAA, CNF, and CNF-DA systems, respectively. The red and blue shading indicate Fe³⁺/Fe²⁺-coordination interactions.

the fracture energy of the CNF-DA/PAA@Fe³⁺ hydrogel after inducing by UV light presented a decay of 0.52−0.024 MJ/m³, a 22-fold drop, revealing the tunable stretchability of the hydrogel triggered by UV irradiation (Fig. 3f and Supplementary Fig. 19).

Similarly, the CNF-DA/PAA@Fe³⁺ hydrogel displayed a high compressive strength of 16.8 MPa at extreme compressibility up to 95% strain, 5 times higher than the PAA hydrogel (3.7 MPa) (Fig. 3g). After UV light stimulation, the compressive strength of the CNF-DA/PAA@Fe³⁺ hydrogel is transferred to 2.3 MPa at a compressive fracture strain of 87.3%, leading to the significant deformation of the hydrogel under heavy weight, indicating that UV light can regulate the compressibility of the hydrogel (Fig. 3h, i). In addition, the CNF-DA/PAA@Fe³⁺ hydrogel also shows excellent and tunable compression strength and toughness (($Value_{max}$-$Value_{min}$)/$Value_{max}$ of up to 88% for toughness) by adjusting the water or CNF contents of the hydrogel (Supplementary Figs. 20−22). These results demonstrated the mechanical properties tunability of the CNF-DA/PAA@Fe³⁺ hydrogel enabled by the transformation of Fe ions during the UV-stimulus process, which is of great significance to the self-powered e-skins with satisfactory mechanical performance.

## Photo-detachable adhesive performance of the CNF-DA/PAA@Fe³⁺ hydrogel

The CNF-DA/PAA@Fe³⁺ hydrogel exhibited splendid reversible adhesion properties, which plays an essential role in self-powered e-skins. Such dynamic adhesion is associated with the reversible supramolecular network transformation due to the Fe ions valence conversion based on the P.F. reaction. As shown in Fig. 4a, the CNF-DA/PAA@Fe³⁺ hydrogel enabled it to toughly bond to human fingers and different substrates without significant peeling gaps, suggesting excellent and broad-range adhesion properties of the fabricated hydrogel. Additionally, the CNF-DA/PAA@Fe³⁺ hydrogel can tightly adhere to various identical representative substrates including glass, aluminum, polyvinyl chloride (PVC), polyethylene terephthalate (PET), polycarbonate (PC), and polyamide, and easily support a 500 g weight suspended from the ends of the substrates, demonstrating its capacity for strong and stable adhesion (Supplementary Fig. 23 and Supplementary Movie 2).

To quantitatively evaluate the adhesion performance of the CNF-DA/PAA@Fe³⁺ hydrogel to various substrates, we conducted three types of mechanical tests, including the interfacial toughness by 90-degree peel tests, the shear strength by lap-strength, as well as the tensile strength by tensile tests (Fig. 4b, Supplementary Figs. 24 and 25). As shown in Fig. 4c−e, the CNF-DA/PAA@Fe³⁺ hydrogel works with various representative substrates, including skin, glass, polydimethylsiloxane (PDMS), aluminum, PET, polytetrafluoroethylene (PTFE), polyamide, and hydrogel, yielding high interfacial toughness, shear and tensile strength, demonstrating the strong and broad-range adhesion properties of the prepared hydrogel (Supplementary Figs. 26−28). Furthermore, the tunability of the adhesion performance of the CNF-DA/PAA@Fe³⁺ hydrogel to different substrates was investigated by being exposed to UV irradiation. The interfacial toughness, shear, and tensile strength of the resulting hydrogel exhibited a significant decrease stimulated by UV irradiation, facilitating easy photo-detachment of the

hydrogel from various substrates after UV irradiation. In particular, the interfacial toughness of the hydrogel adhered to conventional glass decreased from 94.92 J m⁻² to 10.41 J m⁻², the shear strength decreased from 166.68 kPa to 19.35 kPa, and the tensile strength decreased from 64.74 kPa to 16.59 kPa after UV irradiation. Similarly, when applied to the skin, the photo-detachable hydrogel presented a flexible tunability, showing a significant decrease (> 92%) in interfacial toughness (from 89.13 J m⁻² to 5.82 J m⁻²), shear strength (from 119.42 kPa to 6.91 kPa), as well as tensile strength (from 74.79 kPa to 13.10 kPa) after UV irradiation, respectively. Attractively, the adhesive properties of the CNF-DA/PAA@Fe³⁺ hydrogel can be further enhanced substantially by decreasing the water content of the hydrogel. The resultant hydrogel with a water content of 50 wt% presented a high interfacial toughness of 1218.23 J m⁻², shear strength of 1352.62 kPa, and tensile strength of 1245.03 kPa on the skin; interfacial toughness of 695.39 J m⁻², shear strength of 693.13 kPa, and tensile strength of 1299.22 kPa on the glass. It is worth noting that decreasing water content does not compromise the tunability of adhesive performance of the CNF-DA/PAA@Fe³⁺ hydrogel, as evidenced by its tunable adhesive performance ratio (Determined by ($Value_{max}$-$Value_{min}$)/$Value_{max}$) of up to 93% in terms of the interfacial toughness, shear strength, and tensile strength (Supplementary Figs. 29−34). Therefore, these impressive results demonstrated the strong adhesion yet easy photo-detachment and broad-range adhesion properties of the CNF-DA/PAA@Fe³⁺ hydrogel, thereby improving the adaptability and applicability of our fabricated hydrogel.

Beyond the rapid and effective UV-triggered photodetachment, the CNF-DA/PAA@Fe³⁺ hydrogel also demonstrated an extremely rapid and stable self-recovery adhesive performance after air oxidation of 5 min (Supplementary Figs. 35, 36). Additionally, the CNF-DA/PAA@Fe³⁺ hydrogel presented a strong adhesion to the freshly excised porcine skins over 2 h, and maintained a rapid and stable adhesion and photo-detachment even after 5 cycles of UV irradiation and air oxidation processes, indicating the stable and reversible adhesion performance of the developed hydrogel (Supplementary Fig. 37). Encouragingly, a wide range of adhesion strength of the CNF-DA/PAA@Fe³⁺ hydrogel can be achieved by tuning the UV intensity and irradiation time, and the hydrogel still exhibits excellent adhesive strength as the temperature varies, which also imparts the desired adhesion of the developed hydrogel for multi-scene applications (Supplementary Figs. 38 and 39). More attractively, our obtained hydrogel displayed a high tunability (> 90%) of adhesion strength, which is ascendant to most reported switchable adhesive hydrogels (Fig. 4f and Supplementary Table 1)[7,11,12,23−31]. Benefiting from the strong, stable, and broad-range adhesion performance, as well as tunable adhesion behavior triggered by friendly UV irradiation[32,33], the cell-compatible dynamic hydrogel shows promising prospects for diversifying potential applications, especially in green electronics such as wearable sensors and e-skins (Supplementary Figs. 40 and 41).

## Universality of the CNF-mediated photo-detachable adhesion strategy

To further verify the universality of the photo-detachable adhesion strategy based on the P.F. reaction, six additional hydrogels derived

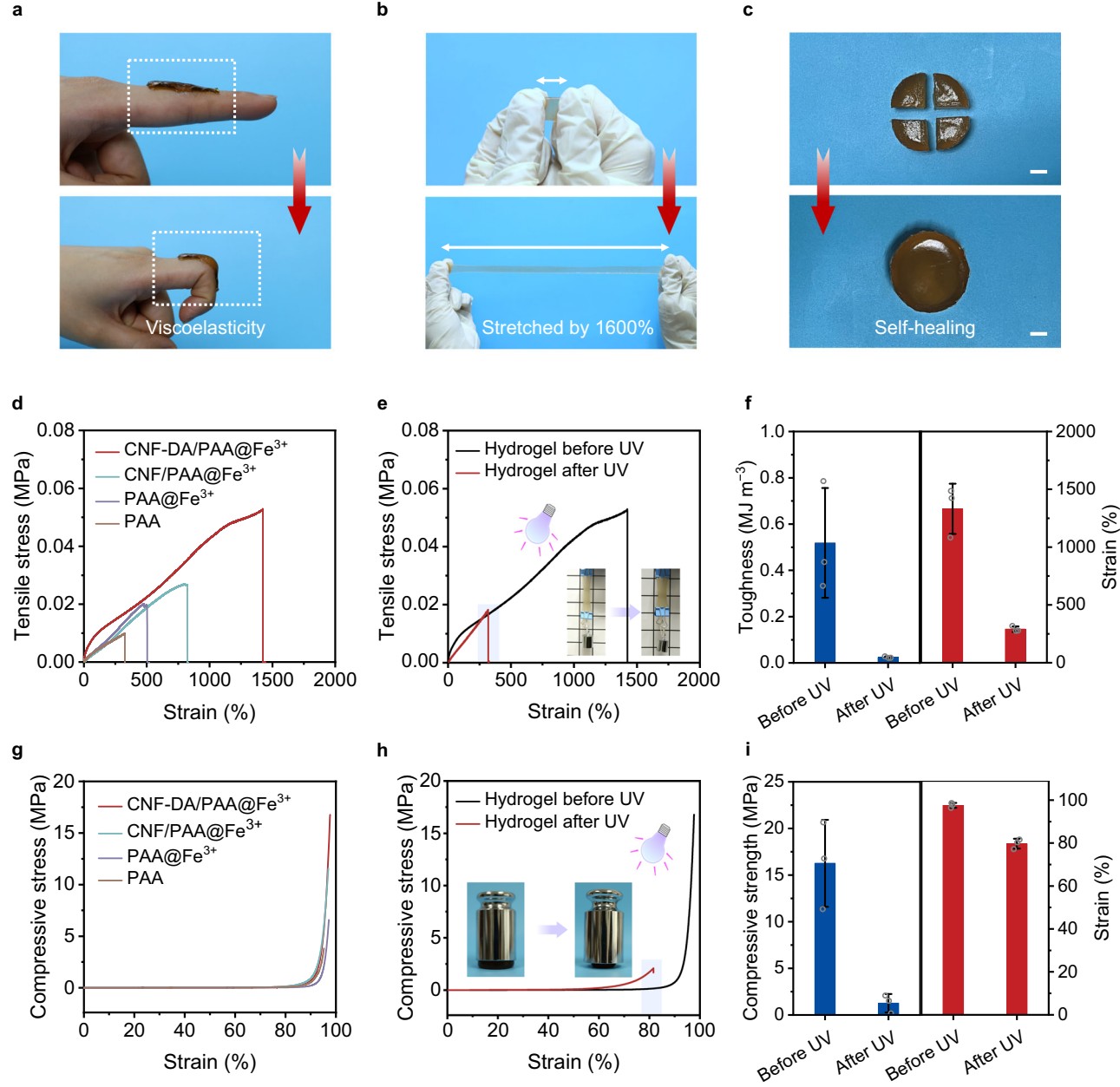

**Fig. 3 | Mechanical performance of the CNF-DA/PAA@Fe$^{3+}$ hydrogel.**
**a** Photographs of the CNF-DA/PAA@Fe$^{3+}$ hydrogel with excellent flexibility and adhesion ability to the human knuckle. The thickness of CNF-DA/PAA@Fe$^{3+}$ hydrogel is -2.5 mm. **b** Photographs of the CNF-DA/PAA@Fe$^{3+}$ hydrogel with high stretchability. The thickness of CNF-DA/PAA@Fe$^{3+}$ hydrogel is -1 mm. **c** Photographs of the self-healing performance of the CNF-DA/PAA@Fe$^{3+}$ hydrogel. The thickness of CNF-DA/PAA@Fe$^{3+}$ hydrogel is -25 mm. Scale bar, 1 cm. **d** Tensile stress-strain curves of the different hydrogels. **e** Tensile stress-strain curves of the CNF-DA/PAA@Fe$^{3+}$ hydrogel before and after the UV irradiation. Inset images display the hydrogel in the stretched state before and after UV irradiation. The purple shaded area indicates the fracture strain of the hydrogel after UV light.

**f** Comparison of the toughness and fracture strain of the CNF-DA/PAA@Fe$^{3+}$ hydrogel before and after the UV irradiation (*n* = 3). **g** Compressive stress-strain curves of the different hydrogels. **h** Compressive stress-strain curves of the CNF-DA/PAA@Fe$^{3+}$ hydrogel before and after the UV irradiation. Inset images show the corresponding hydrogel in the compressive state before and after the UV irradiation. The purple shaded area indicates the ultimate strain of the hydrogel after UV irradiation. **i** Comparison of the compressive strength and strain of the CNF-DA/PAA@Fe$^{3+}$ hydrogel before and after the UV irradiation. Data in (**f**, **i**) are reported as their means ± SDs from *n* = 3 and *n* = 3 independent samples, respectively. Source data are provided as a Source Data file.

from biomass and synthesized polymers were prepared. First, Gaussian simulations were performed to investigate the feasibility of the dynamic structural dissociation and reconfiguration based on the interactions between iron ions and six coordination polymer frameworks including gelatin, chitosan, alginate, starch, polyacrylamide (PAAm), and polyvinyl alcohol (PVA). It can be observed that the binding energy between Fe$^{3+}$ and gelatin (−127.64 kcal/mol), chitosan (−115.01 kcal/mol), alginate (−95.28 kcal/mol), starch

(−88.51 kcal/mol), PAAm (−94.61 kcal/mol), and PVA ( − 72.14 kcal/mol) was much lower than that between Fe$^{2+}$ and these polymers (Fig. 5a, b, Supplementary Fig. 42 and Supplementary Data 1). Such a difference enables the coordination dissociation of supramolecular networks via CNF-mediated manifold dynamic synergy, which decreases the cohesive strength of the hydrogels, thereby allowing triggerable detachment of these prepared hydrogels with the exposure to UV light.

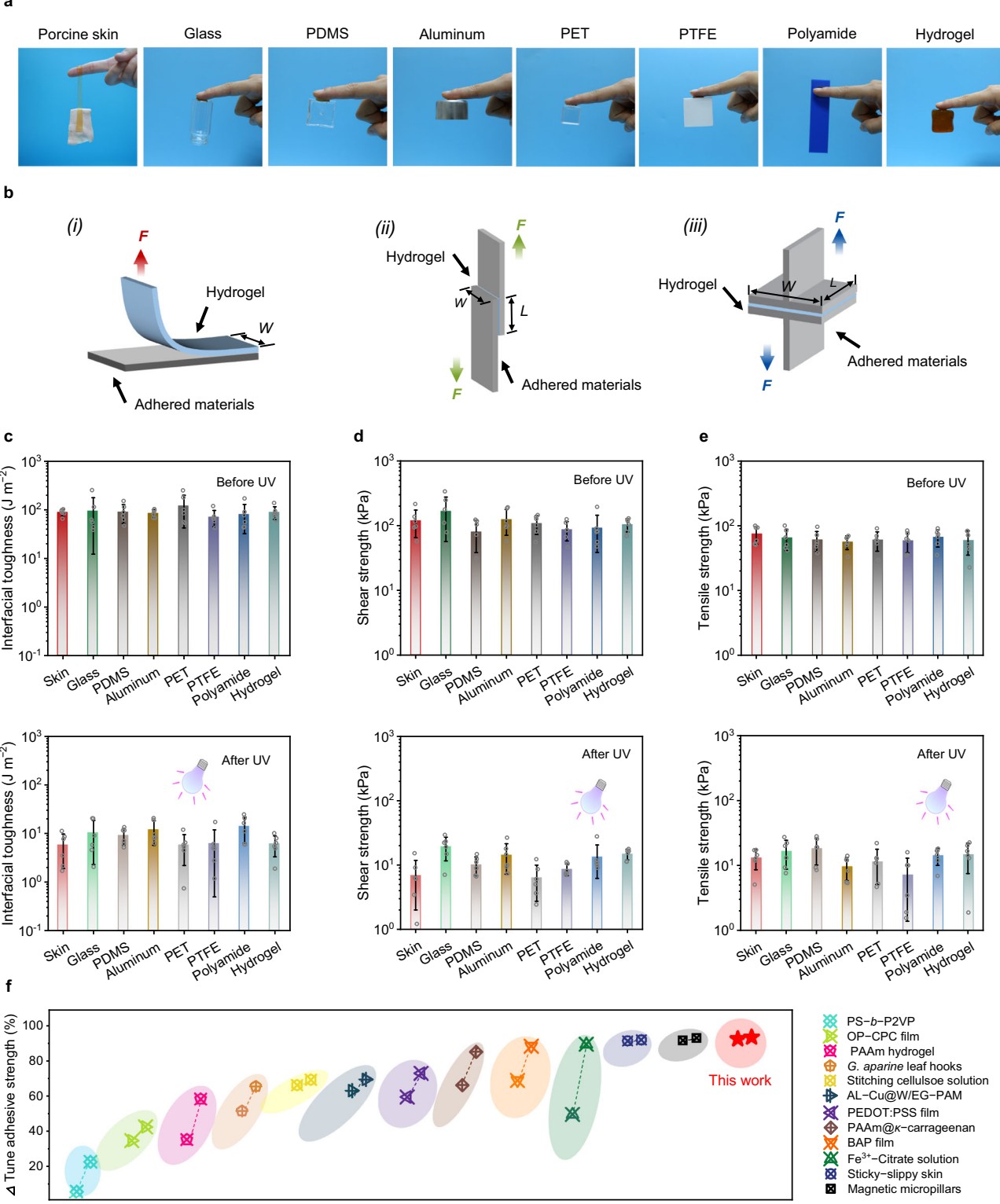

**Fig. 4 | Photo-detachable adhesive performance of the CNF-DA/PAA@Fe$^{3+}$ hydrogel. a** Photographs of the CNF-DA/PAA@Fe$^{3+}$ hydrogel with excellent adhesion to different representative substrates. **b** Setup schematics for mechanical testing of adhesion performance: (i) interfacial toughness by the standard 90-degree peeling test (ASTM D2861). (ii) shear strength by the standard lap-shear test (ASTM F2255). (iii) tensile strength by the standard tensile test (ASTM F2258). The red, green, and blue force present the 90-degree peeling force, shear force, and tensile force, respectively. **c**–**e** Interfacial toughness, shear strength, and tensile strength of the CNF-DA/PAA@Fe$^{3+}$ hydrogel to various representative substrates before and after UV irradiation. Values represent the mean and standard deviation. **f** Comparison of the tunable ratio in adhesive strength between CNF-DA/PAA@Fe$^{3+}$ hydrogels and other previously reported hydrogels. The data used are summarized in Supplementary Table 1. Data in (**c**–**e**) are reported as their means ± SDs from n = 6 independent samples. Source data are provided as a Source Data file.

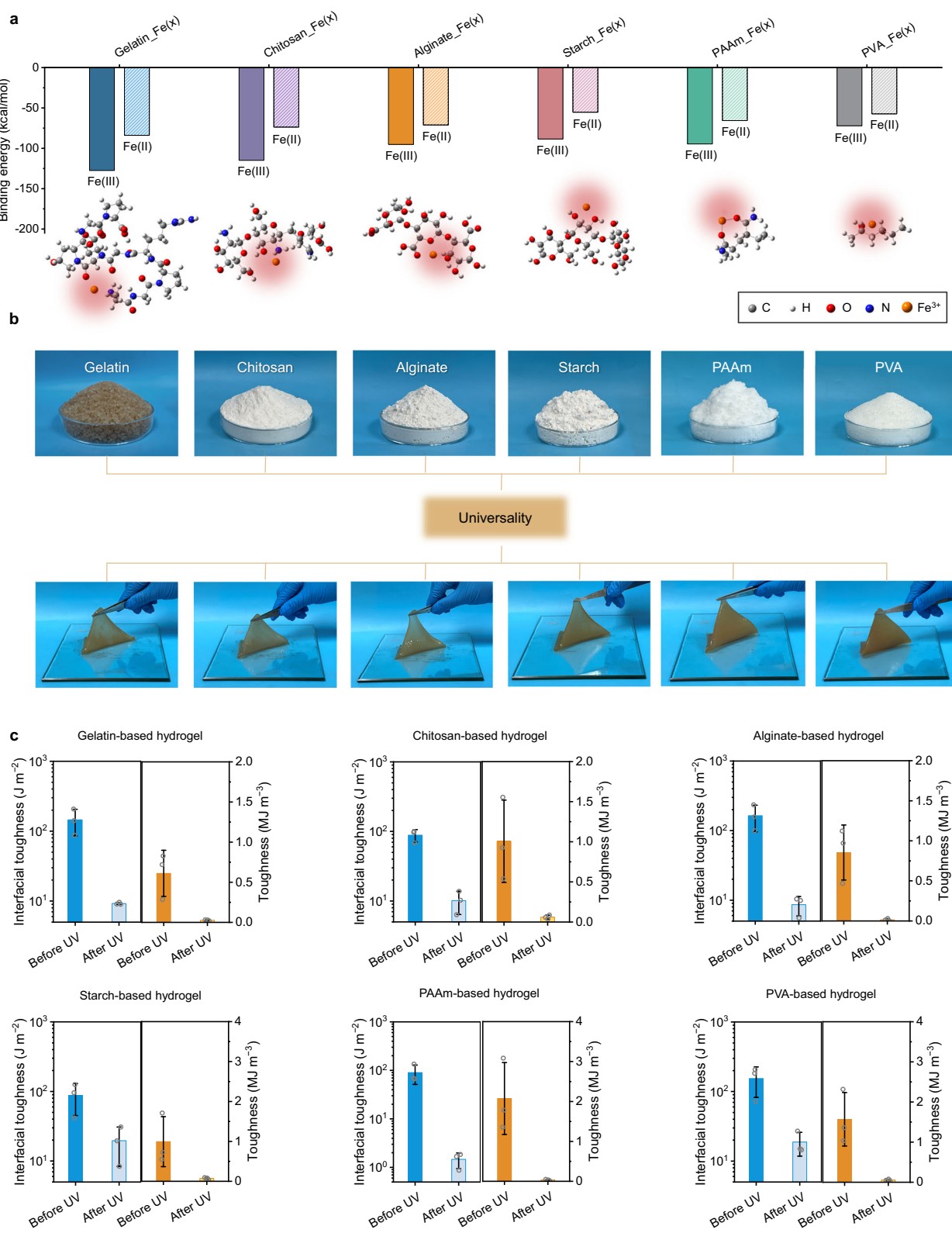

**Fig. 5 | Universality of the CNF-mediated photo-detachable adhesion strategy.**
**a** Binding energy among Fe ions ($Fe^{3+}$, $Fe^{2+}$) and gelatin, chitosan, alginate, starch, PAAm, and PVA. Fe (x), x represents the valence state of Fe ions, including Fe (III) and Fe (II). The illustrations are snapshots of $Fe^{3+}$ with gelatin, chitosan, alginate, starch, PAAm, and PVA chain, respectively. The red shading indicates Fe ions-coordination interactions. **b** Extending this photo-detachable adhesion strategy to different kinds of biomass materials and petroleum-based polymers, namely, gelatin, chitosan, alginate, starch, PAAm, and PVA. **c** Mechanical and adhesive impacts of the different kinds of biomass materials and petroleum-based polymers with and without UV irradiation. Data in **c** are reported as their means ± SDs from $n = 3$ independent samples. Source data are provided as Source Data file.

Beyond the theoretical simulations, all these fabricated hydrogels were experimentally shown to have strong yet reversible flexibility and interfacial toughness. Upon UV irradiation, the tensile toughness of the hydrogels decreased from 0.61 MJ m$^{-3}$ to 0.026 MJ m$^{-3}$ (for gelatin), 1.01 MJ m$^{-3}$ to 0.060 MJ m$^{-3}$ (for chitosan), 0.85 MJ m$^{-3}$ to 0.014 MJ m$^{-3}$ (for alginate), 0.99 MJ m$^{-3}$ to 0.073 MJ m$^{-3}$ (for starch), 2.07 MJ m$^{-3}$ to 0.033 MJ m$^{-3}$ (for PAAm), 1.57 MJ m$^{-3}$ to 0.055 MJ m$^{-3}$ (for PVA), respectively. As shown in Fig. 5c, Supplementary Figs. 43 and 44, the tensile toughness and interfacial toughness tunability ratios of all these hydrogels exceed 85%. These results demonstrated the manifold dynamic synergy strategy of the CNFs high-density interaction network, the physical entanglement between CNF fibers and PAA chains, and the strong coordination with Fe$^{3+}$ ions and CNF network endow hydrogels with strong interfacial binding and photo-detachment, which facilitates the reliable production of adhesive and photo-detachable hydrogels from a variety of resources, creating a wide-open space for diversifying potential applications, especially for self-powered electronic skins and intelligent devices.

### Photo-detachable adhesion-triboelectric nanogenerator (PdA-TENG) as a self-powered e-skin

The advantageous mechanical features, combined with the high ionic conductivity, biocompatibility, and photo-tunable adhesion behavior, enable our developed hydrogel as epidermal electronics under soft UV light. Notably, the use of human-friendly UV light ($\leq$ 40 mW cm$^{-2}$ for 5 min) for hydrogel-skin on-demand photodetachment is sufficient to achieve gentle detachment without any additional intervention. When applied to the skin as an ionic conductor, the hydrogel not only maintains robust contact and enhances signal clarity, but also offers benefits such as improved portability and conformability. More importantly, when the signal is collected, the device allows for its secure and swift removal without harming the skin. As a proof-of-concept demonstration, this hydrogel acting as an ionic conductor is effectively incorporated into flexible sensors or triboelectric nanogenerators as part of electronic skin systems that accurately track human movements and physiological signals with a high signal-to-noise ratio (Supplementary Figs. 45–47, Supplementary Movies 3 and 4). Specifically, a single-electrode mode PdA-TENG with a two-layer-like architecture (Fig. 6a, b, Supplementary Figs. 48 and 49) is fabricated to demonstrate the potential of the hydrogel via a series of measurements in long-term self-powered e-skin devices. The CNF/PAA@Fe$^{3+}$ hydrogel was used as the wearable substrate and ionic current collector and top-down encapsulated by a PDMS layer to improve its water retention properties (Supplementary Figs. 50–52 and Supplementary Movie 5). The obtained PdA-TENG enables excellent and steady output fluctuation, demonstrating its great stability and durability (Supplementary Figs. 53 and 54).

The human body is a multisensory system where different parts have special physiological signal characteristics (Supplementary Fig. 55). For more flexible detection of human movement, a wireless system composed of the PdA-TENG, wireless transceiver module, and wireless receiver for monitoring human activities was integrated (Fig. 6c). Four wearable PdA-TENGs integrated with a capacitor are regarded as a power source for the radio remote controller to harvest the mechanical energy from human motions[34]. A 0.1 mF capacitor is connected in parallel with four PdA-TENGs through a rectifier to test the charging capacity of PdA-TENG for an energy storage device.

The voltage of the capacitor is charged from 0 to 3 V in about five hours, which can drive the wireless transmitter to emit a trigger signal one time (Fig. 6d, e). As such, various human-body activities by mounting the devices can be readily monitored (Fig. 6f and supplementary Movie 6). For example, we record completely different signal characteristics with jug lifting and squatting by adhering PdA-TENGs to the elbows and the knees. Furthermore, we can also monitor body motion by recording signals of stands and squats, leg lifting, and

lunging. Various physiological features and motor behaviors can be transformed into readable, quantifiable, and real-time voltage signals by our self-powered e-skin, facilitating parallel and whole-body physiological and motor monitoring. Hence, our e-skin is expected to have broad application prospects in fields such as personal health monitoring, patient rehabilitation, athletic performance monitoring, and recreational human motion tracking.

## Discussion

In summary, we constructed a supramolecular CNF-DA/PAA@Fe$^{3+}$ hydrogel with both reversible tough adhesion and easy photodetachment via a CNF-mediated manifold dynamic synergy strategy. The CNF-reinforced network, the physical entanglement between CNF and PAA chains, as well as the strong coordination between Fe$^{3+}$ ions and CNF networks endow the excellent reversible mechanical and adhesive properties of the hydrogels. Additionally, the on-demand easy detachment was achieved through the dissociation of coordination complexes when the Fe$^{3+}$ ions was reduced to Fe$^{2+}$ ions upon exposure to UV light, allowing the adjustability of the supramolecular network to provide the hydrogel with switchable toughness and adhesion. The dynamic polymer network brings about the tunability of the mechanical properties and interfacial toughness (tunability ratio of up to 86%), and on-demand adhesion energy (tunability ratio of up to 93%). Strong and photo-detachable adhesion of the fabricated hydrogel works with diverse substrates, including skin, glass, PDMS, aluminum, PET, PTFE, polyamide, and hydrogel, demonstrating the attractive broad-range adhesion-detachment properties of our hydrogel. In addition, the photo-detachable adhesion strategy based on the P.F. reaction is applicable to other biomacromolecules and petrochemical polymers, including gelatin, chitosan, alginate, starch, PAAm, and PVA, where CNF/Fe$^{3+}$ serves as a Fenton-like reagent and the adhesive groups of DA provide initial adhesion. As a proof-of-concept demonstration, a self-healing and mechanically compliant PdA-TENG is fabricated using the CNF-DA/PAA@Fe$^{3+}$ hydrogel as the photo-detachable adhesion substrate and the electrode, and a PDMS elastomer as the electrification layer. The self-powered e-skin based on the single-electrode working mode PdA-TENG achieves complex whole-body physiological wireless signal monitoring including blinking, respiring, walking, and major joint motion detections such as knuckle, elbow, and knee. This study demonstrates the potential application of photo-detachable hydrogel in smart e-skins, self-powered biomechanical monitoring systems, and beyond.

## Methods
### Materials
(2,2,6,6-tetramethylpiperidin-1-yl)oxidanyl (TEMPO, 99.9%), Sodium hypochlorite (NaClO, 11.9%), Sodium hydroxide (NaOH, $\geq$98%), Sodium bromide (NaBr, > 99%), Acrylic acid (AA, 99%), Ammonium persulfate (APS, 98.5%), N, N'-methylenebis(acrylamide) (MBAA, $\geq$ 99%), Iron chloride hexahydrate (FeCl$_3 \cdot$6H$_2$O, 99%), 1-(3-Dimethylaminopropyl)−3-ethylcarbodiimide hydrochloride (EDC, 98%), N-Hydroxy succinimide (NHS, 99%), Dopamine hydrochloride (DA, 98%) were obtained from Aladdin, and all purchased chemicals were of analytical purity. It was prepared for cellulose nanofibrils by the combination of TEMPO oxidation (5 mmol of NaClO per gram of cellulose) of Masson pine needle pulp which was acquired from Masson pine wood powder (Shijiazhuang Teng Bang Mineral Products Co. Ltd.) and intimate mixing[6]. CNF which was oxidized by TEMPO embodies 1.2 mmol g$^{-1}$ surface carboxylate groups (Supplementary Figs. 56 and 57).

### Preparation of CNF-DA solution
CNF (2.00 g, 1.2 mmol g$^{-1}$ − COOH) was dispersed uniformly in 200 mL of deionized water. Before the addition of DA (the molar ratio of amino to carboxyl groups was 4), EDC and NHS (the molar ratio of EDC to NHS

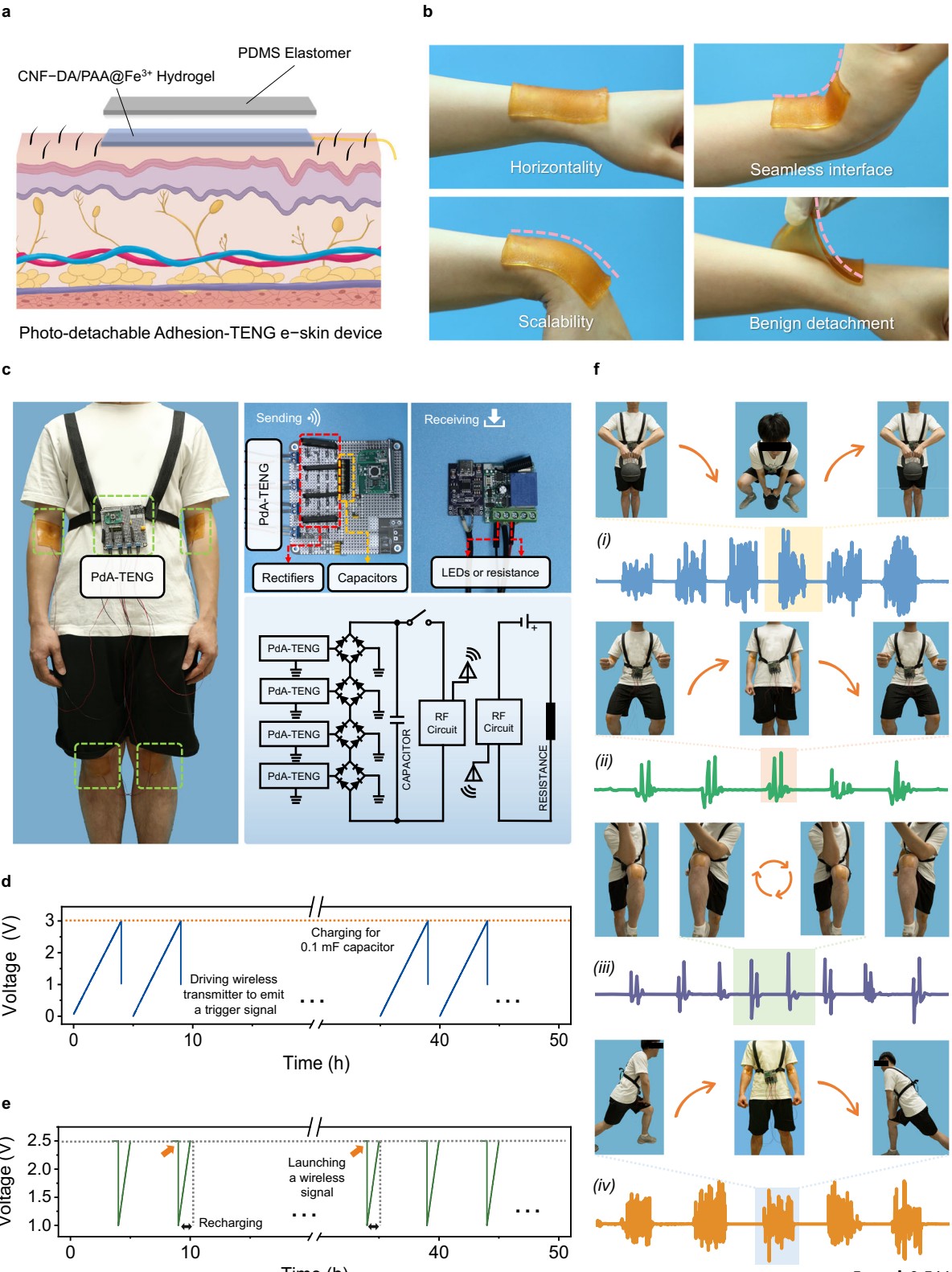

**Fig. 6 | PdA-TENG is used as a self-powered e-kin device for whole-body physiological and motion monitoring. a** Scheme of the photo-detachable adhesion-triboelectric nanogenerator (PdA-TENG) with a two-layer structure. **b** A piece of stretchable PdA-TENG can attach and conform to a human external wrist even under a large tilting action. **c** The photograph is composed of a schematic diagram of integrated energy harvesting, integrated wireless transmitter, wireless receiver, and a simple circuit diagram of the motion monitoring system. **d** Voltage changes of a 0.1 mF capacitor charged by PdA-TENG used to power a wireless transmitter to emit a trigger signal. **e** Recharging a 0.1 mF capacitor by PdA-TENG after the transmitter launches a wireless signal. **f** Detection of body exercises and the corresponding images (inset), jug lifting (i), squatting (ii), leg lifting (iii), and lunging (iv). The shading area represents the electrical signal for different movements, respectively.

was 1) were added separately and reacted in an ice bath for 0.5 h to activate the carboxyl group. After 24 h mixing at room temperature, it was poured into the dialysis bag (Mw = 3500) and dialyzed in the DI water for 72 h. Then the product was evaporated, which was labeled as CNF-DA[35]. The solid-state $^{13}$C nuclear magnetic resonance (NMR) spectra of the CNF and CNF-DA are confirmed (Supplementary Fig. 58).

### Synthesis of CNF-DA/PAA@Fe$^{3+}$ hydrogel
Chemically crosslinked CNF-DA/PAA@Fe$^{3+}$ hydrogel was synthesized via polymerization at 70 °C. A precursor solution was prepared by adding 1 mg/mL chemical cross-linker MBAA and 0.0625 mL/mL thermal initiator APS (with respect to the volume of CNF-DA solution) to a solution containing 5 mL CNF-DA solution, 1.25 mL AA, 0.01 g FeCl$_3$ and 2 mL DI water. The precursor solution was then polymerized under 70 °C for 40 min to form the CNF-DA/PAA@Fe$^{3+}$ hydrogel. The CNF/PAA@Fe$^{3+}$ hydrogel was polymerized under 70 °C for 40 min via adding 1 mg/mL chemical cross-linker MBAA and 0.0625 mL/mL thermal initiator APS (with respect to the volume of CNF solution) to a solution containing 5 mL CNF solution, 1.25 mL AA, 0.01 g FeCl$_3$, and 2 mL DI water (See Supplementary Methods 1 for details on the synthesis of the CNF/PAA@Fe$^{3+}$, PAA hydrogels). The FTIR spectra of the hydrogel are confirmed (Supplementary Fig. 59).

### Characterizations and measurements
Fourier transform infrared (FTIR) spectra of the dried hydrogels, and other freeze-dried samples were analyzed by a Fourier transform infrared spectrometer (Nicolet iS50, Thermo Fisher Scientific, USA) over the range from 4000 cm$^{-1}$ to 400 cm$^{-1}$. A $^{13}$C solid-state nuclear resonance (NMR) spectroscopy (AVANCE NEO, 600 MHz, Germany) was used to determine the structure of the samples, and X-ray photoelectron spectroscopy (XPS) was determined by the analysis of the chemical compositions and structures. The performance of the hydrogels was evaluated using UV-vis spectroscopy (SHIMADZU CORPORATION, UV-2450, Japan). Raman spectroscopy was performed using a Renishaw inVia Raman microscope (Renishaw, inVia, UK) equipped with an objective (Leica, 63 ×) and a 1200 L/mm grating. The wavelength of the excitation laser was 785 nm. The X-ray diffraction (XRD) patterns of the dried bulk hydrogel before and after UV irradiation in the diffractometer (Siemens, Germany) with a scanning speed of 3°/min. The SAXS of hydrogel samples was tested using an instrument (Anton Paar, Graz, Austria) with Cu/Mo X-ray radiation. SEM images of dried bulk hydrogels were acquired with environmental scanning electron microscopy (JSM-7600 Fs, JEOL, Japan), and confocal laser scanning microscopy (CLSM) (LSM710, Zeiss, Germany) was used to test the bulk structure of the hydrogels stained with fluorescent dye solution. Atomic force microscopy (AFM) was used to characterize the surface morphologies of the cellulose nanofibers. A differential scanning calorimeter (DSC, NETZSCH, Germany) was used to heat from 10 to 250 °C at a rate of 10 °C/min for the freeze-dried samples. The PdA-TENG-based e-skin was implemented by a linear motor (Linmot E1100) for electrical measurements. A multimeter (Keithley, 7510, USA) were driven to test the open-circuit voltage, short-circuit current, transferred charges, and the open-voltage of long-term motion cycles.

### Adhesion tests
Interfacial toughness was obtained by measuring adhered samples with a width of 2.5 cm using a mechanical testing machine (100 N load-cell, SUNS, UTM6503, China) by standard 90-degree peeling test (ASTM D2861). All the tests were performed at a constant peeling speed of 50 mm min$^{-1}$, and when the peeling process reached a steady state, the measured force also reached a steady state. The interfacial toughness was determined by dividing the plateau force (for 90-degree peeling test) via the width of the samples. Shear strength was obtained by measuring adhered samples with a width of 2.5 cm using a

mechanical testing machine (5.0 kN load-cell, SUNS, UTM6503, China) by standard lap-shear test (ASTM F2255). All the tests were performed at a constant shear speeding of 50 mm min$^{-1}$. The shear strength was determined by dividing the maximum force (for lap-shear test) via the adhesive area of the samples. Tensile strength was obtained by measuring adhered samples with a width of 2.5 cm using a mechanical testing machine (5.0 kN load-cell, SUNS, UTM6503, China) by standard tensile test (ASTM F2258). All the tests were performed at a constant tensile speeding of 50 mm min$^{-1}$. The tensile strength was determined by dividing the maximum force (for tensile test) via the adhesive area of the samples. Of note, the adhesive performance (e.g., interfacial toughness, shear strength, tensile strength, adhesion energy) tunability ratio is determined by (Value$_{max}$-Value$_{min}$)/Value$_{max}$, where Value$_{max}$ is the maximum value of the performance and Value$_{min}$ is the minimum value of the performance.

### Gaussian simulation
All DFT theoretical calculations have been carried out using the Gaussian 09 program package[36]. The B3LYP density functional method with the D3(BJ) dispersion correction density functional method was employed in this work to carry out all the computations. The 6–31 G(d) basis set was used for the small atoms and the Stuttgart-Dresden (SDD) basis set for Fe atoms[37]. All structures have been optimized considering solvent effects using the polarizable continuum model (PCM) for water[38]. Vibrational frequency analysis of all optimized structures at the same level of theory was performed to characterize stationary points as local minima. The single-point energy calculations were carried out using the def2-TZVP basis set to provide better energy correction[39].

### Fabrication of the PdA-TENG
The PdA-TENG consists of the photo-detachable adhesion CNF-DA/PAA@Fe$^{3+}$ hydrogel substrate and the negative electrification PDMS layer. The commercially available SYLGARD 184 PDMS (part A and part B with a weight ratio of 10:1), as a well-mixed mixture, was poured into a rectangular groove mold, and further cured at 80 °C for 2 h to form a PDMS elastomer layer. Then, the CNF-DA/PAA@Fe$^{3+}$ precursor is injected into the previous PDMS groove before curing. Owing to being prone to the connection between the hydrogel electrode and external circuits, the signal readout wire was immersed in the hydrogel solution, which could be cured and bonded inside the hydrogel. All the devices used in the work are 4 × 3 cm.

### Fabrication of wireless motion monitoring system
A wireless motion monitoring system was constructed of the wireless launch part and the wireless receive part. Four integrated wearable PdA-TENGs fixed on a volunteer were connected with a 0.1 mF capacitor through four rectifier bridges respectively. The capacitor was connected to the wireless launch module as a power source. A 5 V lithium battery was connected to the wireless receiver module for the power supply.

### Demonstration for real-life applications
The experiments were performed with the assistance of volunteers and informed consent was obtained for the publication of images and data. The assembled sensors were attached to different parts of the human body, such as fingers, eyes, feet, and knees. Voltage changes caused by different movements were recorded in real time through a digital electrometer. The volunteers participated in the experiments as approved by the Shanghai Ninth People's Hospital, Shanghai Jiao Tong University School of Medicine Ethics Committee (SH9H-2022-T357-1).

### Statistics and reproducibility
All experiments were repeated independently with similar results at least three times.

**Reporting summary**

Further information on research design is available in the Nature Portfolio Reporting Summary linked to this article.

## Data availability

The data supporting the findings of this study are available with the article and its supplementary files. Any additional requests for information can be directed to, and will be fulfilled by, the corresponding authors. Source data are provided in this paper. Source data are provided with this paper.

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

## Acknowledgements

This research was undertaken, in part, thanks to funding from the National Natural Science Foundation of China (Grant No. 31890774) to H.L., the Forestry Science and Technology Innovation and Extension

Project of Jiangsu Province (No. LYKJ[2021]04) to H.L., the National Natural Science Foundation of China (Grant No. 52273091) and the Fundamental Research Funds for the Central Universities (Grant No. 691000003) to C.C.

## Author contributions

C.C., H.L. and X.X. conceived the concept, processing, and structure details and supervised the work. L.Z. and L.C. carried out most experiments and co-wrote the manuscript. C.C., H.L., L.Z., L.C., L.Y. and D.W. revised the manuscript. S.H.W. assisted in completing the photographs of samples in the manuscript. S.S.W. carried out the computational simulation and analyzed the results. All authors commented on the submitted version of the manuscript.

## Competing interests

The authors declare no competing interests.

## Additional information

[1]Jiangsu Key Laboratory of Biomass Energy and Material, Institute of Chemical Industry of Forest Products, Chinese Academy of Forestry, 210042 Nanjing, China. [2]Hubei Biomass-Resource Chemistry and Environmental Biotechnology Key Laboratory, School of Resource and Environmental Sciences, Wuhan University, 430079 Wuhan, China. [3]Jiangsu Co–Innovation Center of Efficient Processing and Utilization of Forest Resources, College of Chemical Engineering, Nanjing Forestry University, 210037 Nanjing, China. [4]These authors contributed equally: Lei Zhang, Lu Chen. ✉e-mail: liuhe.caf@gmail.com; chenchaojili@whu.edu.cn

