## [Peer Review File · Nature Communications]

Cellulose Nanofiber-Mediated Manifold Dynamic Synergy
Enabling Adhesive and Photo-Detachable Hydrogel for Self-
Powered E-SkinEditorial Note: Parts of this Peer Review File have been redacted as indicated to remove third-party material where no permission to publish could be obtained.

Reviewers' comments:

Reviewer #1 (Remarks to the Author):

The authors report a photo-Fenton-like strategy to achieve both reversible tough adhesion and easy detachment in a CNF-DA/PAA@Fe³⁺ dynamic hydrogel upon exposure to UV light. The authors further prepared an adhesive-detachable triboelectric nanogenerator (TENG) as a human-machine interface for a self-powered wireless monitoring system which broaden the promising application in e-skin filed.

1. It is an interesting strategy to achieve highly reversible adhesion upon exposure to UV light. However, the same mechanism has been well researched and published by Suo group, who took UV light to enable the transformation of Fe³⁺ to Fe²⁺ for the regulation of the adhesion energies from 1400 and 10 J m⁻² (Adv. Mater. 2019, 31, 1806948). Therefore, the novelty of this work is highly weakened in comparison with the previous work.

2. It seems that the adhesion energy of CNF-DA/PAA@Fe³⁺ in various substrates including skin (88 J/m²), glass (60 J/m²), PDMS (65 J/m²), aluminum (70 J/m²), and hydrogel (80 J/m²) was pretty low in comparison with existing interfacial adhesion models, which could reach >1000 J/m². This low adhesion energy has been able to allow hydrogel to be easily removed without special treatments (UV treatment may be negligible). Is it possible to improve the interfacial adhesion energy?

3. Interfacial adhesion could be rebuilt upon exposure to white light by air oxidization, how long the oxidization process will take?

4. Dose the coordination interaction in the CNF-DA/PAA@Fe³⁺ hydrogel rebuild after air oxidization process? Small-angle X-ray scattering test, UV-vis spectra tests, XPS tests could be carried out to give a demonstration of the coordination interaction in the hydrogel after air oxidization process.

5. Direct exposure to UV light for adhesion weakening was not friendly to human skin when the hydrogel was used as e-skin.

6. The mechanical strength of hydrogel was < 0.05 MPa, which was low and hard to enable the high adhesion at various surfaces.

Reviewer #2 (Remarks to the Author):

The manuscript reported the design and fabrication of a hydrogel with tunable adhesive as self-powered e-skin. The manuscript is well-organized with plenty of data. However, there are some issues need to added.

1. The author mentioned “topological network structure”, but didn’t explain clearly about what’s the topological network in this system and how it changes towards stimuli.
2. In figure 3a,b,c, the hydrogel seems have different color. Whether the amount of Fe would influence the color of the hydrogel and the performance of the hydrogel?
3. In the past, there are many work using Fe ion coordinated with COOH to fabricate high performance hydrogels. The novelty of this work should be emphasized.

Reviewer #3 (Remarks to the Author):

Recommendation: Minor revision

Comments to the authors:

In this manuscript, the authors reported a light-driven supramolecular topology network engineering strategy for the construction of a dynamic hydrogel with adhesive and photo-detachable performance and evaluated its function and application potential as self-powered e-skins. This strategy used UV light to drive the supramolecular topology network transformation via the photo-Fenton-like reaction (P.F. reaction) of common copolymers with cellulose nanofibrils, which in turn triggers the conversion of Fe³⁺ and Fe²⁺ ions to regulate the properties of hydrogels. This strategy is interesting and practical, and the resulting CNF-DA/PAA@Fe³⁺ hydrogel is superior to existing dynamically adhesive hydrogels in terms of adhesive properties while maintaining high tunability. The obtained results and discussions for designing and fabricating the CNF-DA/PAA@Fe³⁺ hydrogel were provided through extensive state-of-the-art experimental characterizations and theoretical calculations. The manuscript is well-written. I would like to recommend the publication of this work in Nature Communications after some minor revisions to address the following points.

1. In Fig. 2d, the diagram should be drawn in the form of q-lq² to analyze the formation of dispersible microstructure in hydrogel with UV light.
2. In Fig. 2e, the peak-splitting curves in the XPS spectrum should be better represented.
3. Both experimental and theoretical simulation show strong coordination interactions between Fe³⁺ and -COOH groups. However, the Fe³⁺-catechol coordination also play an important role in the topology and photo-detachable property of hydrogel. Please discuss the interactions of Fe³⁺-catechol coordination and hydrogen bonding in the hydrogel in details.
4. The catechol groups could be oxidized gradually by O₂ and increase the crosslinking density of hydrogel. How about the stability of CNF-DA/PAA@Fe³⁺ hydrogel.
5. How to exclude the influence of temperature on the performance of adjusted adhesion with UV light?
6. How does photo-detachable cellulosic CNF-DA/PAA@Fe³⁺hydrogel relate to its application in self-powered electronic skin?
7. What is the benefit of using cellulose nanofibrils? Please explain in more details.
8. The presented photo-Fenton-like reaction induced reversible transformation of supramolecular topology network strategy is interesting. I am interested to know whether such strategy is applicable for

other hydrogel or ionic gel materials as well? The authors are suggested to provide further discussion on this point.

Itemized list of response to the Referees' remarks (Black: Referees' remarks; Blue type: Our response)

Referee #1:

Comments:

The authors report a photo-Fenton-like strategy to achieve both reversible tough adhesion and easy detachment in a CNF-DA/PAA@Fe³⁺ dynamic hydrogel upon exposure to UV light. The authors further prepared an adhesive-detachable triboelectric nanogenerator (TENG) as a human-machine interface for a self-powered wireless monitoring system which broaden the promising application in e-skin filed.

Reply to the Referee: We thank Referee #1 for the constructive comments, especially for pointing out the interesting phenomenon and novel concept of properties enhancement by a photo-Fenton-like strategy of our work.

1. It is an interesting strategy to achieve highly reversible adhesion upon exposure to UV light. However, the same mechanism has been well researched and published by Suo group, who took UV light to enable the transformation of Fe³⁺ to Fe²⁺ for the regulation of the adhesion energies from 1400 and 10 J m⁻² (Adv. Mater. 2019, 31, 1806948). Therefore, the novelty of this work is highly weakened in comparison with the previous work.

Reply to the Referee: We thank Referee #1 for the valuable suggestions and positive comment on our interesting strategy to achieve highly reversible adhesion upon exposure to UV light. However, We regret not being able to agree with the referee's views on this point as **our work is substantially different from the previous work**, as briefly quoted below.

Ref.1 Adv. Mater. 2019, 31, 1806948. (Aqueous solution for photodetachable adhesion)

Abstract

Peeling from strong adhesion is hard, and sometimes painful. Herein, an approach is described to achieve both strong adhesion and easy detachment. The latter is triggered, on-demand, through an exposure to light of a certain frequency range. The principle of photodetachable adhesion is first demonstrated using two hydrogels as adherends. Each hydrogel has a covalent polymer network, but does not have functional groups for bonding, so that the two hydrogels by themselves adhere poorly. The two hydrogels, however, adhere strongly when an aqueous solution of polymer chains is spread on the surfaces of the hydrogels and is triggered to form a stitching polymer network in situ, in topological entanglement with the pre-existing polymer networks of the two hydrogels. The two hydrogels detach easily when the stitching polymer network is so functionalized that it undergoes a gel-sol transition in response to a UV light. For example, two pieces of alginate-polyacrylamide hydrogels achieve adhesion energies about 1400 and 10 J m⁻², respectively, before and after the UV radiation. Experiments are conducted to study the physics and chemistry of this

strong and photodetachable adhesion, and to adhere and detach various materials, including hydrogels, elastomers, and inorganic solids.

[REDACTED]

Figure 1. Photodetachable adhesion of two pieces of polyacrylamide (PAAm) hydrogels.

In this work (**Ref.1**), the two PAAm hydrogels with no functional groups for bonding adhere poorly. An aqueous solution of PAA chains with certain rheology was added on the surface of two pieces of pristine hydrogels and waited for its permeation into hydrogels. In these circumstances, another aqueous solution of Fe^{3+} and citric acid with a controlled pH value of 1.5~6 was needed to diffuse into the hydrogel to form the coordination complexes to strongly adhere the two hydrogels. Under these conditions, concerning the time-consuming permeation process and accessibility of the diffusion substrate, this aqueous solution-enabled adhesion can only be obtained in the aqueous adherend system after a long waiting time. Particularly, given the acid environment and pre-prepared permeation of the PAA solution, it is extremely difficult to apply this photo-detachable system to human skin, leading to limited applications in self-powered e-skin. More importantly, after the UV radiation, the Fe^{3+} -cross-linked PAA network dissociated into PAA chains again, and the dense interfacial zone between the two hydrogels enabled by cross-linking of the PAA network disappeared for photodetachment.

Our work (CNF-mediated photo-Fenton reaction for the reversible photo-detachable adhesion)

Abstract

Self-powered skin attachable and detachable electronics are under intense development to enable the internet of everything and everyone in new and useful ways. Existing on-demand separation strategies rely on complicated pretreatments and physical properties of the adherends, achieving detachable-on-demand in a facile, rapid, and universal way remains challenging. To overcome this

challenge, we propose a rapid, reliable, and universal strategy to achieve both reversible tough adhesion and easy detachment by regulating its supramolecular network via the cellulose nanofiber (CNF)/Fe³⁺ (as a Fenton-like reagent)-mediated dopamine-poly(acrylic acid) dynamic hydrogel. This strategy enables the simple and rapid fabrication of strong yet reversible hydrogels with tunable toughness $((Value_{max}-Value_{min})/Value_{max})$ of up to 88%, on-demand adhesion energy $((Value_{max}-Value_{min})/Value_{max})$ of up to 98%, and stable conductivity up to 12 mS m⁻¹. We further extend this strategy to fabricate different CNF/Fe³⁺-based hydrogels from various biomacromolecules and petroleum polymers, and shed light on exploration of fundamental dynamic supramolecular network reconfiguration. Simultaneously, we prepare an adhesive-detachable triboelectric nanogenerator (TEENG) as a human-machine interface for a self-powered wireless monitoring system based on this strategy, which can acquire the real-time, self-powered monitoring, and wireless whole-body movement signal, opening up possibilities for diversifying potential applications in e-skins and intelligent devices.

Figure 1. Design of CNF-DA/PAA@Fe³⁺ dynamic hydrogel based on a light-driven supramolecular network engineering strategy via photo-Fenton reaction.

In our work, we propose a **rapid, reliable, and universal strategy to achieve both reversible tough adhesion and easy photodetachment of the hydrogel by regulating its supramolecular network via the cellulose nanofibers (CNFs)-mediated photo-Fenton-like reaction.** The developed strongly adhesive CNF-DA/PAA@Fe³⁺ hydrogel is composed of Fe³⁺, CNF-DA, and PAA networks, in which the CNF network serves as a Fenton-like reagent and the supporting framework of the hydrogel, and produces coordination interactions with Fe³⁺ for excellent mechanical and adhesive performance. Upon the UV radiation, the Fe³⁺ ions are reduced to Fe²⁺ ions in the photo-Fenton-like reaction, leading to dissociation and reconstruction of the Fe ions and CNF networks, further the reconfiguration of the supramolecular network inside the hydrogel. As such, we achieve a strong and immediate adhesion on various substrates ranging from aqueous and nonaqueous systems, and rapid and human-friendly detachment of the hydrogel. More attractively, the mechanical and adhesion energy is tunable for diversifying applications, and the detached hydrogel can be easily self-recovered by air oxidation of the hydrogel within 5 min for reuse. Furthermore, the biocompatible and human-friendly hydrogel as an ionic conductor in flexible sensors or triboelectric nanogenerators can be applied in electronic skins to stably monitor human movements and physiological signals with a high signal-to-noise ratio, creating a wide-open space for potential applications, especially for personal health monitoring, patient rehabilitation, athletic performance monitoring, and recreational human motion tracking.

To illustrate the originality of our current work, we summarize the major differences between our work and the Suo group's previous work (Adv. Mater. 2019, 31, 1806948), as shown below.

1. **CNFs-mediated supramolecular reconfiguration regulatory mechanism (Our work) VS interfacial iron-carboxyl chemistry regulatory mechanism (Ref.1 Adv. Mater. 2019, 31, 1806948).** In our work, cellulose nanofibrils (CNFs), as the supporting framework of hydrogel, are also the key to the regulation of the valence state of iron ions. CNF served as a reducing agent in the photo-Fenton reaction and meanwhile contributed to the reversibility and stability of coordination interactions in the CNF-DA/PAA@Fe³⁺ hydrogel. Upon the UV radiation, due to the transformation of the valence state of Fe ions, the coordination interaction among Fe ions, CNFs, and PAA chains is dynamically dissociated and reconstructed, affecting the supramolecular network reconfiguration inside the hydrogel, further self-regulating the cohesive energy of the hydrogel. This is the first time that the photo-Fenton reaction involving the transformation of the Fe ions valence via CNFs is introduced to regulate the supramolecular network reconfiguration within the hydrogel, thus regulating the mechanical and adhesive properties of the hydrogel. By sharp contrast, Ref.1 focused on the interfacial iron-carboxyl chemistry with Fe³⁺-citric acid and PAA aqueous solution and adherends. UV radiation triggered the dissociation of permeated PAA chains into the two PAAm hydrogels, rather than the photo-Fenton reaction.
2. **CNFs-mediated hydrogel with widely tunable mechanical and adhesive performance (Our work) VS single photo-detachable adhesion performance (Ref.1 Adv. Mater. 2019, 31, 1806948).** In our work, the introduction of CNFs endows the hydrogel with excellent yet tunable mechanical and adhesive properties. For the first time, we easily utilize the photo-Fenton-like specialty of CNFs and iron ions in combination with UV light radiation to rapidly obtain dynamic hydrogels with tunable mechanical and adhesive properties, instead of monotonous photo-detachable glue in Ref.1. The mechanical and adhesive performance of the hydrogel is tunable by tuning the Fe contents, water content, and CNF/DA contents in the hydrogel, indicating excellent flexibility, adaptability, and applicability of our hydrogel. However, the adhesion energy is related to the permeation and gelation of PAA and the pH of the Fe³⁺-citric acid serving as a monotonous adhesive glue in Ref.1, which can hardly satisfy the requirements of adjustable mechanical and adhesive properties for wide applications.
3. **Fast adhesion process and rapid self-recovery adhesion performance (Our work) VS time-consuming and unrecoverable adhesion process (Ref.1 Adv. Mater. 2019, 31, 1806948).** The hydrogels prepared in our work exhibit adhesion immediately, which is much faster and more efficient than the adhesion process reported in Ref. 1 (Less than 1 s VS 10 min). The abundant oxygen-containing functionalized groups on CNFs (especially catechol) can directly produce strong interactions for adhesion with the adherends, which is an efficient path to rapidly form a conformal and perfect adhesion. By sharp contrast, the whole adhesion process reported in Ref. 1 is extremely time-consuming because the PAA aqueous solution needs to diffuse and gelate in the two hydrogels and the Fe³⁺-citric acid solution needs to penetrate the adherends and thus generate coordination interactions with PAA chains. More importantly, the interfacial adhesion of our hydrogel could be rebuilt upon exposure to white light by air oxidization. The adhesion strength and energy of the hydrogels can be recovered to their original state with 5-minute exposure to open air.

4. **Universality of CNFs-mediated photo-detachable adhesion strategy (Our work) VS aqueous adhesion-detachable strategy (Ref.1 Adv. Mater. 2019, 31, 1806948).** In our work, benefiting from introducing polymer chains with versatile functional groups, this strategy applies to a wide variety of hydrogels and adherends. As such, the hydrogel works as a universal gel for facile adhesion to diverse substrates (both aqueous and nonaqueous systems), without any additional interfacial chemical designs or pretreatments for adherends. On the other hand, the photo-detachable adhesion strategy based on the P.F. reaction is applicable to other biomacromolecules and petrochemical polymers, including gelatin, chitosan, alginate, starch, acrylamide (PAAm), and polyvinyl alcohol (PVA), in which CNF/Fe³⁺ serves as a Fenton-like reagent. By contrast, Fe³⁺-citric acid aqueous solution in Ref. 1 as an adhesive glue is inapplicable to nonaqueous adherends, and the adhesion performance depends on the choice between adherends types and operations.
5. **Multifunctional applications (Our work) VS limited applications (Ref.1 Adv. Mater. 2019, 31, 1806948).** With excellent reversible and tunable features of viscoelasticity, self-healing, and ionic conductance, our developed dynamic gel demonstrates great potential as a flexible, transparent, designable, and biocompatible sensor device for real-world applications including e-skins, soft robots, energy storage, and intelligent devices. However, the accessible permeation of the adherends and Fe³⁺-citric acid aqueous solution with controlled pH value of 1.5~6 in Ref. 1 has limited its applications due to the specific and applicable environment of the iron-carboxyl chemistry for the Fe³⁺-citric acid aqueous solution.

These major differences between our work and previous works are further listed in the following Table R1 for a clear and detailed comparison.

Table R1. Comprehensive comparisons (differences) of our work with the Suo group's previously reported paper.

Product (Study object)	Fabrication Method	Photo-detachable adhesion time (min)	Micro-structure	Photodetachment mechanism	Universality of adherends and strategy	Properties	Applications	Refs
CNF-mediated photo-detachable adhesive hydrogel	In-situ synthesis of hydrogels by copolymerization of CNF-DA and iron ions as the P. F. reagent with acrylic acid	Immediate adhesion; detaching after 5 mins (UV light, 40 mW cm ⁻²)	Reverse transformation, from the dense to loose micro-structure	Supramolecular network reconfiguration inside the CNF-mediated hydrogel to further enable self-regulation of the cohesive energy	1. Diverse substrates (both aqueous and non-aqueous systems) 2. Photo-detachable adhesion strategy applies to other biomacromolecules and petrochemical polymers	1. Reversible mechanical properties; 2. Tunable adhesive properties; 3. Self-healing properties; 4. Ionic conductivity 5. Biocompatibility	1. E-skins, soft robots, energy storage, and intelligent devices; 2. Wound dressing and transdermal drug delivery	This work
Fe ³⁺ -citric acid aqueous solution	FeCl ₃ and citric acid were dissolved in deionized water.	Adhesion by 10 mins penetration; detaching after 3 mins (UV light, 60 mW cm ⁻²)	Solution	Interfacial iron-carboxyl chemistry	Only aqueous systems	Photodetachable adhesive properties	Need to transfer thin films of devices from a donor to a target substrate	Ref. 1

References

[1] Gao, Y., Wu, K. & Suo, Z. *Adv. Mater.* **2019**, *31*, 1806948.

In brief, our universal light-driven supramolecular reconfiguration approach to regulate the adhesion properties of CNF-mediated hydrogels is faster, safer, more reliable, scalable, and designable compared to the methods of regulating the adhesion between two hydrogels through the iron-carboxyl chemistry as reported in Ref.1. Our CNF-mediated hydrogel through this light-driven supramolecular reconfiguration strategy has the combined features of reversible and excellent viscoelasticity, flexible scalability, and satisfied designability, thus **substantially different** from that of the Fe^{3+} -citric acid aqueous solution in Suo group's previous work.

Moreover, to give a clearer picture of our innovative work according to the comments of Referee #1, we summarize the novelty of our work as follows:

(a) CNFs-mediated photo-Fenton reaction for the reversible photo-detachable dynamic adhesion

Available position-selective and abundant carboxy groups on the CNF surface offer abundant active sites to bond with Fe ions for coordination complexes in the hydrogel. Upon UV radiation without external additives, the environment of $-\text{COOH}$ groups changes and $-\text{CHO}$ groups are obtained in the CNF-DA/PAA@ Fe^{3+} hydrogel, meanwhile, Fe^{3+} ions were transformed into Fe^{2+} ions in the photo-Fenton reaction. As such, the coordination interaction between CNF-DA, Fe ions, and PAA chains is dynamically transformed through the valence state switching of Fe ions, and the supramolecular network structure inside the hydrogel undergoes the dissociation-reconfiguration, facilitating the triggerable benign detachment of the hydrogel; Attractively, as the resultant hydrogel was exposed to air, the Fe^{2+} ions were oxidized to Fe^{3+} ions, leading to the reconfiguration of supramolecular network of the CNF-DA/PAA@ Fe^{3+} hydrogel in air for strong yet reversible adhesion of the hydrogel.

(b) CNFs-mediated hydrogels with widely tunable properties (i.e., self-healing properties, mechanical properties, reversible photo-detachable adhesion behavior, and self-recovery adhesive performance)

The strong coordination interactions and CNF serving as the supporting skeleton of the internal network structure enable the excellent mechanical properties of the hydrogel. The CNF-DA/PAA@ Fe^{3+} hydrogel with strong stretchability, durability, and stability displays excellent self-healing performance. Additionally, the mechanical properties and adhesive performance of the hydrogel are tunable by regulating the water or CNF contents in the hydrogel. The developed hydrogel exhibited the maximum mechanical properties (0.57 MJ m^{-3} for toughness, 34.8 MPa for compressive strength) and adhesive performance (1739 J m^{-2} for interfacial toughness, 1017 kPa for shear strength, 1286 kPa for tensile strength). Upon the UV radiation, the hydrogel presents dramatical tunability ratios of the interfacial adhesion energy (by up to 94%) and toughness (by up to 83%) and thus realizing the reversible photo-detachment of the hydrogel with various substrates. Encouragingly, a wide range of adhesion strength of the hydrogel can be achieved (more than 1739 J m^{-2} for interfacial toughness, more than 99% for tunable adhesion ratio) by tuning the UV intensity and radiation time or hydrogel water content, imparting the desired adhesion of the developed hydrogel for multi-scene applications. More attractively, being exposed to air for 5 min, we can achieve the switchable operability of the above processes with associated reversible adhesion, suggesting the great potential of the light-driven engineered hydrogel in rapid and efficient reuse.

(c) Universal and scalable photo-detachable adhesion strategy

The photo-Fenton reagent formed by CNF and iron ions can be introduced into a wide variety of hydrogel matrices, benefiting from the dynamic regulation of CNF chains with multifunctional functional groups and its internal network, and can facilely adhere and detach to diverse substrates (like aqueous and nonaqueous systems), without requiring any interfacial chemical design or pretreatment for adherends (more than 1739 J m^{-2} for interfacial toughness, 99% for adhesion adjustable ratio). CNF/ Fe^{3+} can be served as a Fenton-like reagent to realize the transformation between Fe^{3+} - Fe^{2+} ions under UV radiation, and then enables a tunable supramolecular hydrogel. We prepared different hydrogels consisting of CNF/ Fe^{3+} as a Fenton-like reagent, including those based on gelatin, chitosan, alginate, starch acrylamide (PAAm), and polyvinyl alcohol (PVA). All fabricated hydrogels show good potential in strong yet reversible flexibility and interfacial toughness, and the tensile toughness and interfacial toughness tunability ratios of all these hydrogels exceed 88%. The universality of this strategy will facilitate the reliable production of photo-detachable dynamic hydrogels from a variety of resources ranging from biomass to synthesized polymers.

In addition, we appreciate Referee #1's constructive suggestions and comments regarding the current manuscript. At the same time, we are aware that the academic quality of our manuscript should be further improved to dispel these doubts, and that is what we've been working on in the past few months. While addressing all the comments, we have further improved this work given more comprehensive experiments and simulations (Figure R1). **The highlights of our revisions added in the Revised Manuscript are summarized below:**

I. Computational simulation to evaluate the binding energies between Fe ions (Fe^{3+} , Fe^{2+}) and other macromolecules and polymers including gelatin, chitosan, alginate, starch, PAAm, and PVA

We further investigated the applicability of this strategy to different kinds of biomacromolecules and polymers to confirm its universality. Gaussian simulations were employed to get further insights into the binding energy between Fe ions (Fe^{3+} , Fe^{2+}) and gelatin, chitosan, alginate, starch, PAAm, and PVA, which creates a chance of preparation of reversible photo-detachable dynamic hydrogel from a variety of resources ranging from biomass to synthesized polymers.

II. Photo-detachable adhesive tests of a variety of CNFs-mediated adhesive hydrogels

We carried out tensile tests and 90-degree peeling tests to quantitative analysis of the stretchability and interfacial toughness of gelatin-based, chitosan-based, alginate-based, starch-based, PAAm-based, PVA-based hydrogels. All developed hydrogels formed by CNF- Fe^{3+} photo-Fenton reagents and biomass materials or petroleum-based polymers exhibit excellent yet reversible flexibility and photodetachment performance.

III. Demonstration of the coordination interaction in the hydrogel after the air oxidization process

We added a series of compressive experimental analyses including in-situ Raman spectroscopy, X-ray photoelectron spectroscopy (XPS), 2D-COS synchronous and asynchronous spectroscopy, confocal (CLSM) and scanning electron microscopy (SEM) images, small-angle X-ray scattering

(SAXS), the solid-state ^{13}C NMR spectroscopy, and ultraviolet-visible (UV-vis) spectroscopy to further demonstrate supramolecular reconfiguration in the CNF-DA/PAA@Fe $^{3+}$ hydrogel before and after UV radiation as well as after air oxidation (See Revised Figure 2).

IV. Tunable and stable mechanical and adhesive performance by regulating water contents of the hydrogel

We carried out tensile, compression, and 90-degree peeling tests to explore the mechanical and adhesive performance of the CNF-DA/PAA@Fe $^{3+}$ hydrogel with a water content ranging from 50 wt% to 80 wt%. The CNF-DA/PAA@Fe $^{3+}$ hydrogel showed the maximum values of mechanical (0.57 MJ m $^{-3}$ for toughness, 34.8 MPa for compressive strength) and adhesive properties (1739 J m $^{-2}$ for interfacial toughness, 1017 kPa for shear strength, 1286 kPa for tensile strength).

V. Tunable and stable mechanical and adhesive performance by regulating CNF contents of the hydrogel

We fabricated a series of CNF-DA/PAA@Fe $^{3+}$ hydrogels with a fixed water content of 50 wt% and different CNF-DA contents to further improve the mechanical properties of the hydrogel. According to the CNF-DA content to the mass of the total hydrogel as x, the prepared hydrogel is denoted as CNF-DA/PAA@Fe $^{3+}$ _x, where x is 7.4 wt%, 10 wt%, 12 wt%, 15 wt%. For the case of CNF-DA/PAA@Fe $^{3+}$ hydrogels with varying CNF-DA contents, the mechanical properties of the hydrogels are correspondingly improved with the increase of CNF-DA contents.

VI. Tunable and stable mechanical and adhesive performance by regulating Fe contents of the hydrogel

To investigate the influence of Fe $^{3+}$ contents on the mechanical and adhesive performance of CNF-DA/PAA@Fe $^{3+}$ hydrogels, we prepared series of hydrogels with different Fe $^{3+}$ contents. According to the ratio of Fe $^{3+}$ to the mass of the CNF-DA as x, the prepared hydrogel is denoted as CNF-DA/PAA@Fe $^{3+}$ _x, where x is 0.5 wt%, 1 wt%, 2 wt%, and 4 wt%. We conducted compression and tensile tests to investigate the mechanical performance of the CNF-DA/PAA@Fe $^{3+}$ hydrogel with varying Fe contents. To explore the adhesive properties of these CNF-DA/PAA@Fe $^{3+}$ hydrogels, we performed 90-degree peeling, lap-shear, and tensile tests on the substrates of freshly excised porcine skin and engineering glass.

VII. Self-recovery adhesive performance of the CNF-DA/PAA@Fe $^{3+}$ hydrogel over the air oxidation time

To investigate the adhesive recovery behavior of the photodetached hydrogel on the freshly excised porcine skin in terms of the adhesive strength during the air-oxidation process, we conducted the 90-degree peel, lap-shear, and tensile tests of the hydrogels with different air oxidation time on the substrate of the freshly excised porcine skin.

VIII. Excellent and stable adhesive performance of the CNF-DA/PAA@Fe $^{3+}$ hydrogel under different environment temperatures

To exclude the influence of temperature on the hydrogel, we evaluated the adhesion performance of the CNF-DA/PAA@Fe³⁺ hydrogel on the skin at different temperatures (20, 25, 30, and 35 °C). Furthermore, we investigated the adhesion strength of the CNF-DA/PAA@Fe³⁺ hydrogel on glass under varied UV intensity, and it can be observed that the adhesion strength displays a significant decreasing trend with the increase of UV intensity.

IX. Stable mechanical and adhesive performance of the hydrogel over the air oxidation time

To explore the mechanical and adhesive stability of the CNF-DA/PAA@Fe³⁺ hydrogel, we placed the hydrogel in the atmosphere (temperature is 25 °C and relative humidity is 70%) for 12, 24, 48, and 96 h, respectively, and then quantitatively tested mechanical and adhesive properties of these hydrogels. For the different oxidation times, we denote the hydrogel as CNF-DA/PAA@Fe³⁺_x, where x is 12, 24, 48, and 96 h. As a control, the initially prepared hydrogel is denoted as CNF-DA/PAA@Fe³⁺_0 h.

I. Computational simulation to evaluate the binding energies between Gelatin, Chitosan, Alginate, Starch, PAAm, PVA and Fe ions (Fe³⁺, Fe²⁺)

II. Detailed 90-degree peeling and tensile toughness tests

III. Detailed air-oxidation process investigation

IV. Excellent yet reversible mechanical and adhesive properties of CNF-DA/PAA@Fe³⁺ hydrogel prepared with different water content

(i) High yet reversible photodetachable adhesion of CNF-DA/PAA@Fe³⁺ hydrogel prepared with different water content

(ii) Strong yet tunable flexibility and compressibility of CNF-DA/PAA@Fe³⁺ hydrogel prepared with different water content

V. Excellent mechanical properties of CNF-DA/PAA@Fe³⁺ hydrogel prepared with different CNF ratio

(i) Robust and strong mechanical properties of CNF-DA/PAA@Fe³⁺ hydrogel prepared with different CNF ratio

VI. Excellent yet reversible mechanical and adhesive properties of CNF-DA/PAA@Fe³⁺ hydrogel prepared with different iron content

(i) Excellent yet reversible stretchability and photodetachable adhesion of CNF-DA/PAA@Fe³⁺ hydrogel prepared with different iron content

VII. Strong adhesive properties of CNF-DA/PAA@Fe³⁺ hydrogel with the extension of oxidation time on the freshly excised porcine skin

(i) Strong adhesive properties of CNF-DA/PAA@Fe³⁺ hydrogel with the extension of oxidation time on the freshly excised porcine skin

VIII. Adhesion strength as a function of several variables

(i) Excellent photodetachable adhesion of CNF-DA/PAA@Fe³⁺ hydrogel with different environment temperature after UV light

IX. Stable mechanical and adhesive properties of CNF-DA/PAA@Fe³⁺ hydrogel placed at different time

(i) Stable stretchability, compressability and adhesive properties of CNF-DA/PAA@Fe³⁺ hydrogel placed at different time

Figure R1 Summary of our experimental and simulation efforts during the past few months to provide more solid evidences towards wide, excellent yet tunable mechanical and adhesive properties and insights of the structure-property-function relationship of the developed dynamic hydrogel.

Updates to the revised manuscript: According to Referee #1's comments, we have added a more careful description to further explain the supramolecular network in the revised manuscript on Pages 2-4 and 17-18.

“Self-powered skin attachable and detachable electronics are under intense development to enable the internet of everything and everyone in new and useful ways. Existing on-demand separation strategies rely on complicated pretreatments and physical properties of the adherends, achieving detachable-on-demand in a facile, rapid, and universal way remains challenging. To overcome this challenge, we propose a rapid, reliable, and universal strategy to achieve both reversible tough adhesion and easy detachment by regulating its supramolecular network via the cellulose nanofiber (CNF)/Fe³⁺ (as a Fenton-like reagent)-mediated dopamine-poly(acrylic acid) dynamic hydrogel. This strategy enables the simple and rapid fabrication of strong yet reversible hydrogels with tunable toughness ((Value_{max}-Value_{min})/Value_{max} of up to 88%), on-demand adhesion energy ((Value_{max}-Value_{min})/Value_{max} of up to 98%), and stable conductivity up to 12 mS m⁻¹. We further extend this strategy to fabricate different CNF/Fe³⁺-based hydrogels from various biomacromolecules and petroleum polymers, and shed light on exploration of fundamental dynamic supramolecular network reconfiguration. Simultaneously, we prepare an adhesive-detachable triboelectric nanogenerator (TENG) as a human-machine interface for a self-powered wireless monitoring system based on this strategy, which can acquire the real-time, self-powered monitoring, and wireless whole-body movement signal, opening up possibilities for diversifying potential applications in e-skins and intelligent devices.” On Page 2.

“Existing skin-attachable electronics with autonomous powering ability are desired for obtaining accurate and reliable biological/physical information and can be reversibly attached to arbitrary surfaces and detached without leaving residues^{1, 2}. Thus the utilization of reversible adhesion hydrogels is of great significance for self-powered electronic skins^{3, 4}. In the past few years, great efforts have been devoted to realizing the reversible adhesion of hydrogel with a variety of hard and soft materials^{5, 6}. Reversible adhesion can be achieved using chemical connection consisting of reversible bonds^{7, 8}, including dynamic covalent bonds, noncovalent bonds and specific chemical groups, and physically topological entanglement through external stimuli, such as pH⁹, temperature¹⁰, current¹¹, and rays¹². These current hydrogel adhesion-detachment strategies suffer from general limitations in their universality. For reversible adhesion through reversible bonds, this method only works for the hydrogels that have been chemically designed or the adherends that can be surface modified. The procedures for hydrogel synthesis with tailored chemical structure and modification of adherends usually need rigorous reaction conditions, time-consuming pretreatments, or the usage of toxic agents. For adhesion through physical topological entanglement, it requires the adherends to have a porous microstructure. This method only works for porous adherend like hydrogels and living tissues. In this regard, the reversible integration of hydrogels and diverse materials calls for a fast, facile, and universal strategy.”

Here we report a rapid, reliable, and universal strategy to achieve both reversible tough adhesion and easy detachment by regulating the supramolecular network of the dynamic hydrogel via the cellulose nanofiber (CNF)/Fe³⁺ (as a Fenton-like reagent)-mediated photo-Fenton (P.F.) reaction for self-powered e-skins (Fig. 1a and Supplementary Fig. 1). CNF derived from biomass cellulose—the most abundant polysaccharide on earth, featuring high mechanical strength, renewability, and biocompatibility, has been regarded as a multifunctional building block for applications in electronics and medicine. Furthermore, TEMPO-oxidized CNF contains position-selective and abundant carboxy groups on the surface, which offers abundant active sites to bond with Fe ions for coordination complexes. Strong coordination interactions between Fe³⁺ and carboxyl groups and the presence of catechol groups in the CNF-DA/PAA@Fe³⁺ hydrogel allow tough adhesion of the hydrogel. As directly exposed to UV light without external additives, through the transformation of the valence state of iron ions, the coordination interaction among Fe ions, CNFs, and PAA chains is dynamically dissociated and reconstructed, leading to the reconfiguration of the supramolecular network inside the hydrogel, and further making the cohesive strength of hydrogel self-regulatable (Fig. 1b and Supplementary Fig. 2). More attractively, by varying the exposure time in the air, we achieve the switchable operability of the above processes with associated reversible adhesion, representing great potentials for the reuse of the light-driven engineered hydrogel. Benefiting from the rich multi-type oxygen-containing functional groups of CNF chains and excellent mechanical properties, the hydrogel is able to stick quickly and conforms perfectly to the subject's skin surface as well as mild triggerable benign detachment with no visible residue and redness on the skin surface when exposed to the human-friendly UV radiation (Fig. 1c, Supplementary Fig. 3, and Supplementary Movie 1). With a switchable light-driven supramolecular network, the photo-detachable hydrogel exhibits excellent tunable adhesive strength change (before UV radiation: after UV radiation = 15.6 times) (Fig. 1d). Combined with the rapid, reliable, and reproducible regulating method, as well as adhesion-on-demand properties based on supramolecular network engineering, the developed convertible hydrogel holds great potential for practical application in photo-detachable self-powered e-skins” On Pages 3-4.

“To further verify the universality of the photo-detachable adhesion strategy based on the P.F. reaction, six additional hydrogels derived from biomass and synthesized polymers were prepared. First, Gaussian simulations were performed to investigate the feasibility of the dynamic structural dissociation and reconfiguration based on the interactions between iron ions and six coordination polymer frameworks including gelatin, chitosan, alginate, starch, polyacrylamide (PAAm), and polyvinyl alcohol (PVA). It can be observed that the binding energy between Fe³⁺ and gelatin (−127.64 kcal/mol), chitosan (−115.01 kcal/mol), alginate (−95.28 kcal/mol), starch (−88.51 kcal/mol), PAAm (−94.61 kcal/mol), and PVA (−72.14 kcal/mol) was much higher than that between Fe²⁺ and these polymers (Fig. 5a, b and Supplementary Fig. 40). Such a difference enables the desirable dissociation of coordination complexes involved in the P.F. reaction and thus triggerable detachment of these prepared hydrogels with the exposure to UV light.

Beyond the theoretical simulations, all these fabricated hydrogels were experimentally shown to be strong yet reversible flexibility and interfacial toughness. Upon UV radiation, the tensile toughness of the hydrogels decreased from 0.47 MJ m⁻³ to 0.057 MJ m⁻³ (for gelatin), 0.33 MJ m⁻³ to 0.050 MJ m⁻³ (for chitosan), 0.45 MJ m⁻³ to 0.044 MJ m⁻³ (for alginate), 0.40 MJ m⁻³ to 0.047 MJ m⁻³ (for starch), 0.42 MJ m⁻³ to 0.043 MJ m⁻³ (for PAAm), 0.31 MJ m⁻³ to 0.050 MJ m⁻³ (for PVA), respectively. As shown in Fig. 5c and Supplementary Fig. 41, the tensile toughness and interfacial toughness tunability ratios of all these hydrogels exceed 88%. These results demonstrated the facile yet versatile strategy for the fabrication of adhesive and photo-detachable dynamic hydrogels for reversible, strong and photo-detachable adhesion, which facilitates the reliable production of adhesive and photo-detachable hydrogels from a variety of resources, creating a wide-open space for diversifying potential applications, especially for self-powered electronic skins and intelligent devices.” On Pages 17-18.

2. It seems that the adhesion energy of CNF-DA/PAA@Fe³⁺ in various substrates including skin (88 J/m²), glass (60 J/m²), PDMS (65 J/m²), aluminum (70 J/m²), and hydrogel (80 J/m²) was pretty low in comparison with existing interfacial adhesion models, which could reach >1000 80 J/m². This low adhesion energy has been able to allow hydrogel to be easily removed without special treatments (UV treatment may be negligible). Is it possible to improve the interfacial adhesion energy?

Reply to the Referee: We thank Referee #1 for the thoughtful suggestions. The fabricated CNF-DA/PAA@Fe³⁺ hydrogel has equilibrium water contents of 80 wt% in the manuscript, whose adhesion energy in various substrates including skin (88 J/m²), glass (60 J/m²), PDMS (65 J/m²), aluminum (70 J/m²), and hydrogel (80 J/m²) were obtained. The water content in the hydrogel materials plays a crucial role in the effort to improve the mechanical and adhesive properties of the hydrogel. In this regard, we fabricated a series of CNF-DA/PAA@Fe³⁺ hydrogels with different water contents for the expected enhancement of the interfacial adhesion energy of the developed hydrogel. According to the water content to the mass of the total hydrogel as x, the prepared hydrogel is denoted as CNF-DA/PAA@Fe³⁺_x, where x is 50 wt%, 60 wt%, 70 wt%, and 80 wt%. As an auxiliary verification, we investigated the interfacial toughness, shear strength, and tensile strength of these hydrogels to illustrate their adhesion performance on the substrates of freshly excised porcine skin and engineering glass before, after UV, and after air exposure using 90-degree peeling, lap-shear, tensile tests.

(1) Digital images of the CNF-DA/PAA@Fe³⁺ hydrogels with different water contents

As shown in Figure R2, the CNF-DA/PAA@Fe³⁺_80% hydrogel, which can be easily removed from the engineering glass, shows weak adhesive strength. By contrast, the CNF-DA/PAA@Fe³⁺_50% hydrogel exhibits a pull on the glass observed without visible fracture when detached from the glass, and the hydrogel surface appeared with the obvious bristle-like patterns, indicating an enhanced adhesive strength and mechanical toughness because of the reduction of the hydration layer on the hydrogel surface, exposing more interfacial adhesive groups, and hence increasing the number of adhesion sites on the interface with the object to adhere, as well as improved interfacial interaction.

Figure R2 The digital images of the CNF-DA/PAA@Fe³⁺ hydrogel with different water contents.

(2) Adhesion properties of the CNF-DA/PAA@Fe³⁺ hydrogels with different water contents

Furthermore, we quantitatively evaluated the adhesive properties of prepared hydrogels with different water contents by 90-degree peeling, lap-shear, and tensile tests on the substrates of freshly excised porcine skin and engineering glass during the in-situ UV irradiation and air oxidation process (Figures R3 and R4). The peel force-displacement, shear force-displacement, and tensile force-displacement curves of hydrogels on the skin before UV light are shown in Figure R3a–c. The plateau force, shear force, and tensile force of CNF-DA/PAA@Fe³⁺_80% hydrogel are around 2.2 N, 17.0 N, and 42.5 N on the skin (1.6 N for plateau force, 14.1 N for shear force, and 30.3 N for tensile force on the glass) (Figure R4a–c), showing inherent weak adhesion strength. With the reduction of water content, the platform force, shear force, and tensile force of the CNF-DA/PAA@Fe³⁺ hydrogels display an upward trend, among which, the CNF-DA/PAA@Fe³⁺_50% hydrogel exhibits the highest platform force, shear force, and tensile force before UV light (47.4 N for plateau force, 251.6 N for shear force, and 803.4 N for tensile force on the skin; 26.5 N for plateau force, 191.8 N for shear force, and 981.4 N for tensile force on the glass) (Figure R4a–c).

(3) Adhesion properties of the CNF-DA/PAA@Fe³⁺ hydrogels with different water contents after the UV radiation and air oxidation

More importantly, after UV radiation, the plateau force, shear force, and tensile force of the CNF-DA/PAA@Fe³⁺_80% hydrogel are around 0.45 N, 1.4 N, and 2.7 N on the skin (0.31 N for plateau force, 1.3 N for shear force, and 2.2 N for tensile force on the glass) (Figures R3d–f and 4d–f), which are 4.8 times for plateau force, 12.1 times for shear force, and 15.7 times for tensile force of the CNF-DA/PAA@Fe³⁺_80% hydrogel on the skin before UV (5 times for plateau force, 10.8 times for shear force, and 13.8 times for tensile force on the glass). In particular, with the reduction of water content, the platform force, shear force, and tensile force of the CNF-DA/PAA@Fe³⁺ hydrogel after UV radiation display a downward trend, among which, the CNF-DA/PAA@Fe³⁺_50% hydrogel exhibits the highest tunable adhesive force ratios after UV light (comparing before UV, 105.3 times for plateau force, 81.2 times for shear force, and 200.9 times for tensile force on the skin; 44.2 times for plateau force, 91.3 times for shear force, and 223.0 times N for tensile force on the glass) (Figures R3d–f and R4d–f). These 90-degree peeling, lap-shear, and tensile tests results after air oxidation are consistent

with the peeling, shear, and tensile behavior of hydrogels before UV radiation from the skin and glass shown in Figures R3g–i and R4g–i.

Figure R3 a–c 90-degree peel, lap-shear, and tensile force-displacement curves of the CNF-DA/PAA@Fe³⁺ hydrogels with different water content on the substrate of freshly excised porcine skin before UV radiation. d–f 90-degree peel, lap-shear, and tensile force-displacement curves of the CNF-DA/PAA@Fe³⁺ hydrogels with different water content on the substrate of freshly excised porcine skin after UV radiation. g–i 90-degree peel, lap-shear, and tensile force-displacement curves of the CNF-DA/PAA@Fe³⁺ hydrogels with different water content on the substrate of freshly excised porcine skin after air oxidation.

Figure R4 a-c 90-degree peel, lap-shear, and tensile force-displacement curves of the CNF-DA/PAA@Fe³⁺ hydrogels with different water contents on the substrate of engineering glass before UV radiation. **d-f** 90-degree peel, lap-shear, and tensile force-displacement curves of the CNF-DA/PAA@Fe³⁺ hydrogels with different water contents on the substrate of engineering skin after UV radiation. **g-i** 90-degree peel, lap-shear, and tensile force-displacement curves of the CNF-DA/PAA@Fe³⁺ hydrogels with different water contents on the substrate of engineering glass after air oxidation.

(4) Comparison of the adhesion performance of various CNF-DA/PAA@Fe³⁺ hydrogels with different water contents

To objectively evaluate the reversible adhesive performance of the CNF-DA/PAA@Fe³⁺ hydrogels with varying water contents, we compared the interfacial toughness, shear strength, and tensile strength during the UV radiation and air oxidation process (Figure R5). Compared to CNF-

DA/PAA@Fe³⁺_80% hydrogel before UV radiation, the interfacial toughness, shear strength, and tensile strength of CNF-DA/PAA@Fe³⁺_50% hydrogel on the substrate of freshly excised porcine skin increase 19.8-fold (1739 J m⁻² vs. 88 J m⁻²), 12.7-fold (1017 kPa vs 80 kPa), and 16.1-fold (1286 kPa vs 80 kPa) (16.6-fold for interfacial toughness, 13.7-fold for shear strength, and 32.1-fold for tensile strength on the engineering glass), respectively. With exposure to UV light, the interfacial toughness, shear strength, and tensile strength of the CNF-DA/PAA@Fe³⁺_50% hydrogel significantly decrease, and its tunable interfacial toughness, shear strength, and tensile strength ratios are 98.8%, 98.7% and 99.5%, which is higher than that of CNF-DA/PAA@Fe³⁺_80% hydrogel (94.0%, 93.2%, 94.5%). Subsequently, when the hydrogel is oxidized in the air, the adhesive performance of the hydrogel recovers to that before UV radiation. More attractively, our obtained hydrogel displayed a high adhesion strength, which is superior to most reported switchable adhesive hydrogels (Figure R6).

Generally, these impressive results demonstrated the tunable adhesion yet easy photo-detachment performance by tuning the water content in the hydrogel. Strong adhesion yet easy photo-detachment, and broad-range adhesion properties of the CNF-DA/PAA@Fe³⁺ hydrogel can be achieved by reducing the water content of the hydrogel, promising to improve the adaptability and applicability of our hydrogel.

Figure R5 a, b Comparison of interfacial toughness, shear strength, and tensile strength of the CNF-DA/PAA@Fe³⁺ hydrogels with different water contents on the substrates of skin and glass before UV radiation. **c, d** Comparison of interfacial toughness, shear strength, and tensile strength of the CNF-DA/PAA@Fe³⁺ hydrogels with different water contents on the substrates of skin and glass after UV radiation. **e, f** Comparison of interfacial toughness, shear strength, and tensile strength of the CNF-DA/PAA@Fe³⁺ hydrogels with different water contents on the substrates of skin and glass after air

exposure. Values represent the mean and the standard deviation ($n = 3$).

Figure R6 Comparison of the adhesive strength with other adhesive materials and CNF-DA/PAA@Fe³⁺ hydrogel prepared with different water contents.¹⁻³

Updates to the revised manuscript: According to Referee #1's comments, we have added a more detailed description to further explain the supramolecular network in the revised manuscript on Page 14.

“Attractively, the adhesive properties of the CNF-DA/PAA@Fe³⁺ hydrogel can be further enhanced substantially by decreasing the water content of the hydrogel. The resultant hydrogel with a water content of 50 wt% presented a high interfacial toughness of 1739 J m⁻², shear strength of 1017 kPa, and tensile strength of 1286 kPa on the skin; interfacial toughness of 1062 J m⁻², shear strength of 777 kPa, and tensile strength of 1557 kPa on the glass. It is worth noting that the decreasing of water content does not compromise the tunability of adhesive performance of the CNF-DA/PAA@Fe³⁺ hydrogel, as evidenced by its tunable adhesive performance ratio (Determined by $(Value_{max}-Value_{min})/Value_{max}$) of up to 98% in terms of the interfacial toughness, shear strength, and tensile strength (Supplementary Figs. 28–32).”

References:

1. Deng, J. et al. A bioinspired medical adhesive derived from skin secretion of *andrias davidianus* for wound healing. *Adv. Funct. Mater.* **29**, 1809110 (2019).
2. Cui, C., Shao, C., Meng, L. & Yang, J. High-strength, self-adhesive, and strain-sensitive chitosan/poly (acrylic acid) double-network nanocomposite hydrogels fabricated by salt-soaking strategy for flexible sensors. *ACS Appl. Mater. Inter.* **11**, 39228–39237 (2019).
3. Ma, Y. et al. Liquid bandage harvests robust adhesive, hemostatic, and antibacterial performances as a first-aid tissue adhesive. *Adv. Funct. Mater.* **30**, 2001820 (2020).

3. *Interfacial adhesion could be rebuilt upon exposure to white light by air oxidization, how long the oxidization process will take?*

Reply to the Referee: We thank Referee #1 for the thoughtful suggestions. The adhesive strength of CNF-DA/PAA@Fe³⁺ hydrogel gradually recovered with the extension of air-oxidation time, and as the oxidation time reached 5 minutes, the adhesion strength of the hydrogel was the same as that before UV radiation. As an auxiliary verification, we explored the adhesive properties of CNF-DA/PAA@Fe³⁺ hydrogel (prepared with a water content of 80%) under different air oxidation times by performing 90-degree peeling, lap shear, and tensile tests of the hydrogel.

To determine the recovery time of the CNF-DA/PAA@Fe³⁺ hydrogel (prepared with a water content of 80%) in terms of the adhesive strength during the air-oxidation process, we conducted the 90-degree peel, lap-shear, and tensile tests of the hydrogels with different air oxidation time on the substrate of the freshly excised porcine skin (Figure R6). The peel, shear, and force-displacement curves of CNF-DA/PAA@Fe³⁺ hydrogels are shown in Figure R11a–c, and the plateau, shear, and tensile force of CNF-DA/PAA@Fe³⁺ hydrogel with an oxidation time of 5 minutes are around 2.1 N, 16.8 N, 42.0 N, respectively (2.2 N, 17.0 N and 42.5 N before UV radiation), exhibiting fast and excellent recovery capability of the adhesive strength of the hydrogel. Additionally, we explored the adhesion energy and adhesive strength of the CNF-DA/PAA@Fe³⁺ hydrogels within 1 hour. It can be seen that the adhesion strength and adhesion energy of the hydrogels has recovered to that before UV radiation with the exposure to the air-oxidation for 5 minutes, and remained unchanged after 5 minutes, demonstrating the flexibly tunable adhesion properties of the CNF-DA/PAA@Fe³⁺ hydrogel.

Figure R6 a, b, c 90-degree peel, lap-shear, and tensile force-displacement curves of the CNF-DA/PAA@Fe³⁺ hydrogels with the extension of oxidation time on the skin. **d** Comparison of adhesive

energy and adhesive strength of the CNF-DA/PAA@Fe³⁺ hydrogels with the extension of oxidation time on the skin. Values represent the mean and the standard deviation (n = 3).

Updates to the revised manuscript: According to Referee #1's comments, we have added a more careful description to further explain the supramolecular network in the revised manuscript on Page 14.

“Beyond the rapid and effective UV-triggered photodetachment, the CNF-DA/PAA@Fe³⁺ hydrogel also demonstrated an extremely rapid and stable self-recovery adhesive performance after air oxidation of 5 min (Supplementary Figs. 33-34).”

4. Dose the coordination interaction in the CNF-DA/PAA@Fe³⁺ hydrogel rebuild after air oxidization process? Small-angle X-ray scattering test, UV-vis spectra tests, XPS tests could be carried out to give a demonstration of the coordination interaction in the hydrogel after air oxidization process.

Reply to the Referee: We thank Referee #1 for the constructive comments. The coordination interaction in the CNF-DA/PAA@Fe³⁺ hydrogel was rebuilt after the air-oxidation process. we have added the small-angle X-ray scattering test, UV-vis spectra tests, and XPS tests to demonstrate the coordination interaction rebuild in the CNF-DA/PAA@Fe³⁺ hydrogel after the air-oxidation process.

The confocal images and scanning electron microscopy (SEM) images of the CNF-DA/PAA@Fe³⁺ hydrogel exhibited the remarkable and flexible morphology transformation from the initial entangled and homogeneous dense microstructure (without UV radiation) to a subsequently loose porous zone on its surface (after UV radiation), and finally to a dense and compact microstructure again (after air-oxidation) (Revised Fig. 2a). For facilitating the evident visualization of this structural transformation process, we performed in situ Raman analysis of the CNF-DA/PAA@Fe³⁺ hydrogel by switching UV light on one side of the non-UV radiated hydrogel during the dynamic introduction of UV light (Revised Fig. 2b). The peak intensity at 1680 cm⁻¹ (belong to carboxyl groups) enhanced and displayed the graded distinction (after exposure to UV light for 5 min); with the continuation of UV radiation, the initial blue area (with lower peak intensity value) completely turned into red (with higher peak intensity value), indicative of the noticeable increase of free carboxyl groups in the CNF-DA/PAA@Fe³⁺ hydrogel as UV light gradually penetrated; finally when the resultant CNF-DA/PAA@Fe³⁺ hydrogel was placed in the air, the red area (with higher peak intensity value) gradually returns to the original blue, suggesting its reversible and tunable network structure. Additionally, the right shift of the UV absorbance band from 402 nm (before UV) to 406 nm (after UV), and then left to 402 nm (after air), suggesting the structure transformation of the hydrogel (Revised Fig. 2c). Small-angle X-ray scattering (SAXS) was further conducted to investigate the subtle transformation of the supramolecular network triggered by UV radiation (Revised Figure 2d). A sharp SAXS peak at $q \approx 0.04 \text{ \AA}^{-1}$ was observed in the fabricated hydrogel before UV radiation, indicating a uniform and strong coordination-bond supramolecular network structure in the hydrogel

in the non-UV state, Fe^{3+} ions are transformed into Fe^{2+} ions after being stimulated by UV light, which makes the tightly entangled supramolecular network of CNF-DA and PAA chains loosely deformed. These results were also demonstrated by the XRD analysis of the CNF-DA/PAA@ Fe^{3+} hydrogel. Such a striking morphological transformation is associated with the significant differences in the interactions of CNF chains, Fe^{3+} ions, and PAA chains, and thus supramolecular network evolution of the hydrogel triggered by UV radiation and air oxidation.

To reveal the inherent driving mechanism of structural transformation of the photo-detachable hydrogel based on the P.F. reaction, we carried out the X-ray photoelectron spectroscopy (XPS) analysis of the hydrogel during the UV radiation and air-oxidation processes (Revised Fig. 2e). The appearance of the peak at 710.6 eV of the hydrogel after UV radiation demonstrated the generation of Fe^{2+} in the system, verifying that Fe^{3+} was partially reduced to Fe^{2+} triggered by UV light, in which two free radicals ($\text{O}_2^{\cdot-}$ and $\text{HO}^{\cdot-}$) also participate in the P.F. reaction process (Supplementary Fig. 6). As the photo-detachable hydrogel was exposed to air, the Fe^{3+} was partially oxidized to Fe^{2+} , facilitating the self-recovery of the supramolecular network in the hydrogel. To identify the functional groups that participated in the photodetachment process, the two-dimensional correlation spectra (2D-COS) was employed to enhance FTIR spectra analysis of the hydrogel under the UV radiation¹ (Revised Fig. 2f, g). The cross-peak (1682, 3200) in the synchronous and asynchronous maps was both positive. According to Noda's judging rule, $-\text{COOH}$ and $-\text{OH}$ groups were the dominant functional groups that participated in the UV radiation process, in which $-\text{COOH}$ groups of CNF-DA chains were more susceptible to UV light. Additionally, the solid-state ^{13}C NMR spectra of the CNF-DA/PAA@ Fe^{3+} hydrogel exhibited that two characteristic peaks at 170 and 200 ppm, assigned to $-\text{COOH}$ and $-\text{CHO}$, respectively, do appear in the hydrogel after UV radiation, indicative of the environment variation of $-\text{COOH}$ groups and the generation of $-\text{CHO}$ in the hydrogel triggered by UV light (Revised Fig. 2h). These results jointly revealed that Fe^{3+} ions were transformed into Fe^{2+} ions stimulated by UV light, and the coordination interactions with CNF-DA and PAA chains were resultantly dissociated; Subsequently, as the resultant hydrogel was exposed to air, the Fe^{2+} ions were oxidized to Fe^{3+} ions, leading to the reconfiguration of supramolecular network of the CNF-DA/PAA@ Fe^{3+} hydrogel under air. Along with this process, UV light acted as a functional factor in the photo-Fenton reaction between CNF-DA chains and Fe^{3+} ions, which triggered the valence state transformation of Fe ions in the system. Meanwhile, the introduction of CNF served as a reducing agent in the photo-Fenton reaction and meanwhile contributed to the reversibility and stability of coordination interactions in the CNF-DA/PAA@ Fe^{3+} hydrogel.

Revised Figure 2 **a** Confocal images and SEM images of the CNF-DA/PAA@Fe³⁺ hydrogel during the UV radiation and air-oxidation process showing the change of microstructures of the photo-detachable hydrogel. Scale bar, 20 μm . **b** Two-dimensional Raman image from the $-\text{COOH}$ intensity (1680 cm^{-1}) of the CNF-DA/PAA@Fe³⁺ hydrogel during the UV radiation and air-oxidation process. Scale bar, 5 μm . **c** Ultraviolet-visible spectroscopy (UV-vis) spectra of the CNF-DA/PAA@Fe³⁺ hydrogel during the UV radiation and air-oxidation process. **d** SAXS curves for the photo-detachable hydrogels during the UV radiation and air-oxidation process. **e** XPS of Fe 2p regions of the CNF-DA/PAA@Fe³⁺ hydrogel during the UV radiation and air-oxidation process. **f, g** 2DCOS synchronous and asynchronous spectra generated from the FTIR spectra during the UV radiation and air-oxidation process. In 2DCOS spectra, the warm color (red) represents positive intensities, while the cold color

(blue) represents negative intensities. **h** ^{13}C NMR spectra analysis of the CNF-DA/PAA@Fe $^{3+}$ hydrogel during the UV radiation and air-oxidation process.

Updates to the revised manuscript: According to Referee #1's comments, we have added a more careful description to further explain the supramolecular network in the revised manuscript on Pages 6–8.

“The CNF-DA/PAA@Fe $^{3+}$ hydrogel with 2% Fe $^{3+}$ content was demonstrated to deliver the best comprehensive performance. All data analysis is based on the CNF-DA/PAA@Fe $^{3+}$ hydrogel sample with 2% Fe $^{3+}$ content (the mass fraction relative to the mass of CNF-DA is 2 wt%) and 80% water content (the mass fraction relative to the mass of the hydrogel is 80 wt%) if not specified otherwise (Supplementary Figs. 4–6). External properties of materials are strongly linked to their subtle structural changes. The confocal images and scanning electron microscopy (SEM) images of the CNF-DA/PAA@Fe $^{3+}$ hydrogel exhibited remarkable and flexible morphology transformation from the initial entangled and homogeneous dense microstructure (without UV radiation) to a subsequently loose porous zone on its surface (after UV radiation), and finally to a dense and compact microstructure again (after air-oxidation) (Fig. 2a and Supplementary Fig. 7). For facilitating the evident visualization of this structural transformation process, we performed in situ Raman analysis of the CNF-DA/PAA@Fe $^{3+}$ hydrogel by switching UV light on one side of the non-UV radiated hydrogel during the dynamic introduction of UV light (Fig. 2b and Supplementary Fig. 8). Upon the UV radiation, the peak intensity at 1680 cm^{-1} (belong to carboxyl groups) exhibited a distinct enhancement, with the initial blue area (with low peak intensity value) being turned into green and completely to red (with high peak intensity value), indicative of the generation of more free carboxyl groups and thus the noticeable looser structure in the CNF-DA/PAA@Fe $^{3+}$ hydrogel triggered by UV light. As the hydrogel was placed in the air, the hydrogel showed an opposite trend, suggesting its reversible and tunable network structure. These results were also demonstrated by combined usage of ultraviolet-visible spectroscopy (UV-vis) spectra, small-angle X-ray scattering (SAXS), and X-ray diffraction (XRD) analysis of the CNF-DA/PAA@Fe $^{3+}$ hydrogel (Fig. 2c, 2d and Supplementary Figs. 9 and 10). Such a striking morphological transformation is associated with the significant differences in the interactions of CNF chains, Fe $^{3+}$ ions, and PAA chains, and thus supramolecular network evolution of the hydrogel triggered by UV radiation and air oxidation^{13–18}.”

“To reveal the inherent driving mechanism of structural transformation of the photo-detachable hydrogel based on the P.F. reaction, we carried out the X-ray photoelectron spectroscopy (XPS) analysis of the hydrogel during the UV radiation and air-oxidation processes (Fig. 2e). The appearance of the peak at 710.6 eV of the hydrogel after UV radiation demonstrated the generation of Fe $^{2+}$ in the system, verifying that Fe $^{3+}$ was partially reduced to Fe $^{2+}$ triggered by UV light, in which two free radicals ($\text{O}_2^{\cdot-}$ and $\text{HO}^{\cdot-}$) also participate in the P.F. reaction process (Supplementary Fig. 11). As the photo-detachable hydrogel was exposed to air, the Fe $^{3+}$ was partially oxidized to Fe $^{2+}$, facilitating the self-recovery of the supramolecular network in the

hydrogel. To identify the functional groups that participated in the photodetachment process, the two-dimensional correlation spectra (2D-COS) were employed to enhance FTIR spectra analysis of the hydrogel under UV radiation (Fig. 2f, g)¹⁹. The cross-peak (1682, 3200) in the synchronous and asynchronous maps was both positive. According to Noda's judging rule, -COOH and -OH groups were the dominant functional groups that participated in the UV radiation process, in which -COOH groups of CNF-DA chains were more susceptible to UV light. Additionally, the solid-state ¹³C NMR spectra of the CNF-DA/PAA@Fe³⁺ hydrogel exhibited that two characteristic peaks at 170 and 200 ppm, assigned to -COOH and -CHO, respectively, do appear in the hydrogel after UV radiation, indicative of the environment variation of -COOH groups and the generation of -CHO in the hydrogel triggered by UV light (Fig. 2h and Supplementary Fig. 12)^{20, 21, 22}. These results jointly revealed that Fe³⁺ ions were transformed into Fe²⁺ ions stimulated by UV light, and the coordination interactions of iron ions with CNF-DA and PAA chains were resultantly dissociated; Subsequently, as the resultant hydrogel was exposed to air, the Fe²⁺ irons were oxidized to Fe³⁺ irons, leading to the reconfiguration of supramolecular network of the CNF-DA/PAA@Fe³⁺ hydrogel in air. Along with this process, UV light acted as a functional factor in the photo-Fenton reaction between CNF-DA chains and Fe³⁺ ions, which triggered the valence state transformation of Fe ions in the system. Meanwhile, CNF served as a reducing agent in the photo-Fenton reaction, which further contributed to the reversibility and stability of the coordination interactions in the CNF-DA/PAA@Fe³⁺ hydrogel."

References:

1. Wang, J., Wu, B., Wei, P., Sun, S. & Wu, P. Fatigue-free artificial ionic skin toughened by self-healable elastic nanomesh. *Nat. Commun.* **13**, 4411 (2022).
5. Direct exposure to UV light for adhesion weakening was not friendly to human skin when the hydrogel was used as e-skin.

Reply to the Referee: We thank Referee #1 for the thoughtful suggestions. In our system, the experimental conditions are mild, that is, UV intensity is 40 mW cm⁻² and its exposure time is 5 min. The World Meteorological Organization has designed one UV index unit as 25 mW cm⁻², so the UV index of our system is between levels 1-2,¹⁻³ which is minimal damage to human skin and the exposure time is only 5 min. Moreover, recently many hydrogels have been developed for biomedical applications with the assistance of UV light.^{4,5} For example, Feng Zhou and co-workers reported a hydrogel coating implanted in natural cattle esophagus with UV-assisted conditioning to validate practical applicability.⁶ Hongwei Ouyang and co-workers developed a hydrogel that can be in-situ polymerized on the pig carotid artery and heart for wound healing under UV-induced, and the treated pigs survived after hemostatic treatments with this hydrogel, which is well-tolerated and appears to offer a significant clinical advantage as a traumatic wound sealant⁷ (See Table R2).

Table R2 Comparison between the UV radiation condition of our work and the EPA UV index exposure categories.

UV wavelength	UV radiation conditions	UV index	Exposure level	Refs
290 nm ~ 400 nm	25 mW m ⁻² , 1 hour	8.7	High (1 – 2, Minimal; 3 – 4, Low; 5 – 6, Moderate; 7 – 9, High; 10 and greater, Very high)	Ref. 1
365 nm	40 mW m ⁻² , 5 minutes	1 ~ 2	Minimal	This work

Updates to the revised manuscript: According to Referee #1's comments, we have added a more careful description to further explain the supramolecular network in the revised manuscript on Pages 4 and 19.

“Benefiting from the rich multi-type oxygen-containing functional groups of CNF chains and excellent mechanical properties, the hydrogel is able to stick quickly and conforms perfectly to the subject’s skin surface as well as mild triggerable benign detachment with no visible residue and redness on the skin surface when exposed to the human-friendly UV radiation (Fig. 1c, Supplementary Fig. 3, and Supplementary Movie 1).” On Page 4

“Therefore, we exploit the supramolecular network reconfiguration as UV light-mediated valence transition of iron ions to regulate hydrogel-skin on-demand adhesion-detachment. Of note, human-friendly UV light ($\leq 40 \text{ mW cm}^{-2}$, 5 min) alone is required to trigger benign detachment without additional activity.” On Page 19

References:

1. Kinney, J. P., Long, C. S. & Geller, A. C. The Ultraviolet Index: a useful tool. *Dermatology Online J.* **6** (2000).
2. Coldiron, B. M. The UV index: a weather report for skin. *Clin. Dermatol.* **16**, 441–446 (1998).
3. Italia, N. & Rehfuss, E. A. Is the global solar UV Index an effective instrument for promoting sun protection? A systematic review. *Health Educ.* **27**, 200–213 (2012).
4. Foyt, D. A., Norman, M. D. A., Yu, T. T. L. & Gentleman, E. Exploiting advanced hydrogel technologies to address key challenges in regenerative medicine. *Adv. Healthc. Mater.* **7**, e1700939 (2018).
5. Jayakumar, A., Jose, V. K. & Lee, J. M. Hydrogels for medical and environmental applications. *Small Methods* **4**, 1900735 (2020).
6. Xu, R., et al. A universal strategy for growing a tenacious hydrogel coating from a sticky initiation layer. *Adv. Mater.* e2108889 (2022).
7. Hong, Y., et al. A strongly adhesive hemostatic hydrogel for the repair of arterial and heart bleeds. *Nat. Commun.* **10**, 2060 (2019).

6. *The mechanical strength of hydrogel was < 0.05 MPa, which was low and hard to enable the high adhesion at various surfaces.*

Reply to the Referee: We thank Referee #1 for the constructive comments. The mass ratio in the hydrogel materials plays a crucial role in the mechanical and adhesive properties of the developed hydrogel. As such, we fabricated a series of CNF-DA/PAA@Fe³⁺ hydrogels with different water contents and CNF-DA contents for the expected enhancement of the mechanical strength of the hydrogel.

(1) Enhanced mechanical strength by tuning the water contents in the hydrogel

According to the water content to the mass of the total hydrogel as x, the prepared hydrogel is denoted as CNF-DA/PAA@Fe³⁺_x, where x is 50 wt%, 60 wt%, 70 wt%, and 80 wt%. Compared with CNF-DA/PAA@Fe³⁺_80% hydrogel, CNF-DA/PAA@Fe³⁺_50% hydrogel with lower water content has a significant advantage in mechanical properties, which is attributed that the decreased water contents inside the hydrogel lead to a weak barrier between water molecules and CNF-DA, Fe³⁺ ions, and PAA chains, further making the enhanced coordination interaction among them, and thus tighter the supramolecular network of the CNF-DA/PAA@Fe³⁺ hydrogel. As an auxiliary verification, we investigated the tensile strength, toughness, tensile Young's modulus, and compressive strength of hydrogels with different water contents and CNF-DA contents to further explain the reversible mechanical performance of the hydrogels using tensile and compressive tests after the UV radiation and air-oxidation. As shown in Figure R7a, the CNF-DA/PAA@Fe³⁺_80% hydrogel deforms greatly under external force. In sharp contrast, the CNF-DA/PAA@Fe³⁺_50% hydrogel maintains high integrity without distinct fractures and cracks under external force, indicating its excellent mechanical strength and dimensional stability. The tensile and compressive stress-strain curves of various hydrogels before UV radiation are shown in Figure R7b,e. It can be observed that both the ultimate tensile and compressive stress of the CNF-DA/PAA@Fe³⁺ hydrogel display an increasing trend as the water content is gradually decreasing. In particular, the CNF-DA/PAA@Fe³⁺_50% hydrogel exhibits the best tensile and properties, that is, the tensile Young's modulus is 0.045 MPa corresponding to the toughness of 0.25 MJ m⁻³, 8.3 times higher than the CNF-DA/PAA@Fe³⁺_80% hydrogel (0.0054 MPa); and the compressive ultimate strength is 29.5 MPa corresponding to the compressive fracture strain of 95%, 1.6 times higher than the CNF-DA/PAA@Fe³⁺_80% hydrogel (18.1 MPa) (Figure R8a). These jointly indicated that the coordination interaction between CNF-DA, Fe³⁺ ions, and PAA chains is enhanced, and the supramolecular network structure of the hydrogel is tighter when water molecules as a blocker inside the hydrogel are reduced.

With exposure to UV light, the mechanical performance of all CNF-DA/PAA@Fe³⁺ hydrogels with different water contents is greatly degraded (Figure R7c,f). The CNF-DA/PAA@Fe³⁺_50% hydrogel displaced low tensile Young's modulus of 0.033 MPa corresponding to the toughness of 0.030 MJ m⁻³, and the compressive ultimate strength of 5.1 MPa, showing that UV-triggered regulation of the stretchability and compressibility of the hydrogels (Figure R8b), which is beneficial to benign photo-

detachment of the hydrogel. Finally, when the UV-radiated hydrogel is exposed to the air, the mechanical performance of the hydrogel recovers to that before UV radiation (Figures R7d, g, and R8c). These results demonstrated the mechanical strength of the hydrogel is tunable by regulating the water contents in the hydrogel, and the hydrogel with excellent mechanical strength still enables reversible photo-detachment by UV radiation.

Figure R7 a Digital images of the CNF-DA/PAA@Fe³⁺ hydrogels with different water contents before and after compression. b–d Tensile stress-strain curves of the CNF-DA/PAA@Fe³⁺ hydrogels with different water contents during the UV irradiation and air-oxidation process. e–g Compressive stress-strain curves of the CNF-DA/PAA@Fe³⁺ hydrogels with different water contents during the UV irradiation and air-oxidation process.

Figure R8 a–c Comparison of toughness, Young’s modulus, and compressive strength of the CNF-DA/PAA@Fe³⁺ hydrogels with different water contents during the UV radiation and air-oxidation process. Values represent the mean and the standard deviation (n = 3).

(2) Enhanced mechanical strength by tuning the CNF-DA contents in the hydrogel

On the other hand, we also prepared a series of CNF-DA/PAA@Fe³⁺ hydrogels with a fixed 50% water contents and different CNF-DA contents to further improve the mechanical properties of the hydrogel. According to the CNF-DA content to the mass of the total hydrogel as x, the prepared hydrogel is denoted as CNF-DA/PAA@Fe³⁺_x, where x is 7.4%, 10%, 12%, 15%. for the case of CNF-DA/PAA@Fe³⁺ hydrogels with varying CNF-DA contents, the mechanical properties of the hydrogel are correspondingly improved with the increase of CNF-DA contents. As shown in Figure R9a, the CNF-DA/PAA@Fe³⁺_15% hydrogel maintains high integrity without distinct fractures and cracks under external force, indicating excellent mechanical strength and dimensional stability. By contrast, the CNF-DA/PAA@Fe³⁺_7.4% hydrogel deforms slightly under external force. Additionally, the tensile and compressive stress-strain curves of various hydrogels after the UV radiation and air-oxidation process are shown in Figure R9b–g. It can be observed that both the ultimate tensile and compressive stress of the CNF-DA/PAA@Fe³⁺ hydrogel displays an increasing trend along with the increase of CNF-DA contents. In particular, the CNF-DA/PAA@Fe³⁺_15% hydrogel exhibits the best tensile and compressive properties, that is, the toughness of 0.57 MJ m⁻³ corresponding to the tensile Young’s modulus is 1.2 MPa, 2.4 times higher than the CNF-DA/PAA@Fe³⁺_7.4% hydrogel (0.24 MJ m⁻³); and the compressive ultimate strength is 34.8 MPa corresponding to the compressive fracture strain of 95%, 1.2 times higher than the CNF-DA/PAA@Fe³⁺_7.4% hydrogel (29.5 MPa). With exposure to UV light and further air-oxidation process, the strong yet reversible stretchability and compression of the CNF-DA/PAA@Fe³⁺_15% hydrogel recovers to that before UV radiation (Figure R10a–c). these impressive results demonstrate the strong flexibility, stretchability, and excellent compressibility of CNF-DA/PAA@Fe³⁺ hydrogels. More attractively, by adjusting the water content of the hydrogel or adjusting the CNF contents, the mechanical properties of the hydrogel have made a qualitative leap. Such strong and photo-detachable adhesion may enable broad applications (Table R3).

Figure R9 a Digital images of the CNF-DA/PAA@Fe³⁺ hydrogels prepared with different CNF-DA content before and after compression (50% for water content). **b–d** Tensile stress-strain curves of the CNF-DA/PAA@Fe³⁺ hydrogels prepared with different CNF-DA content (50% for water content) during the UV irradiation and air-oxidation process. **e–g** Compressive stress-strain curves of the CNF-DA/PAA@Fe³⁺ hydrogels prepared with different CNF-DA content (50% for water content) during the UV irradiation and air-oxidation process.

Figure R10 a-c Comparison of toughness, Young's modulus, and compressive strength of the CNF-DA/PAA@Fe³⁺ hydrogels prepared with different CNF-DA content (water content: 50 wt%) during the UV irradiation and air-oxidation process. Values represent the mean and the standard deviation (n = 3).

Table R3 Comparison of the mechanical and adhesive properties of CNF-mediated photo-detachable adhesive hydrogel.

Product (Study object)	Mechanical properties	Adhesive properties	Applications	Refs
CNF-mediated photo-detachable adhesive hydrogel	46 ~ 1012 kPa for failure tensile strength	80 ~ 1017 kPa for adhesion strength on the skin	1. E-skins, soft robots, energy storage, and intelligent devices; 2. Wound dressing and transdermal drug delivery	This work
HV gel	400 ~ 900 kPa for failure tensile strength	20 ~ 70 kPa for adhesion strength on the skin	E-skins	Ref. 1
PEI/PAA hydrogel powder	28 ~ 120 kPa for failure tensile strength	20 ~ 80 kPa for adhesion strength on the skin	Wet tissue bioadhesives	Ref. 2

Updates to the revised manuscript: According to Referee #1's comments, we added more careful descriptions to further explain the supramolecular network in the revised manuscript (Page 11).

“In addition, the CNF-DA/PAA@Fe³⁺ hydrogel also shows excellent and tunable compression strength and toughness ((Value_{max}-Value_{min})/Value_{max} of up to 88% for toughness) by adjusting the water or CNF contents of the hydrogel (Supplementary Figs. 20 and 21). These results demonstrated the mechanical properties tunability of the CNF-DA/PAA@Fe³⁺ hydrogel during the UV-stimulus process, which is of great significance to the self-powered e-skins with satisfactory mechanical performance.”

References:

1. Cho, K. G., et al. Block Copolymer-based supramolecular ionogels for accurate on-skin motion monitoring. *Adv. Funct. Mater.* **31**, 2102386 (2021).
2. Peng, X., et al. Ultrafast self-gelling powder mediates robust wet adhesion to promote healing of gastrointestinal perforations. *Sci. Adv.* **7**, eabe8739 (2021).

Referee #2:

Comments: *The manuscript reported the design and fabrication of a hydrogel with tunable adhesive as self-powered e-skin. The manuscript is well-organized with plenty of data. However, there are some issues need to added.*

Reply to the Referee: We thank Referee #2 for the positive comments on our work, especially pointing out its extraordinary performances and scientific interest, and the suggestion of publication of our work in *Nature Communications*.

1. The author mentioned “topological network structure”, but didn’t explain clearly about what’s the topological network in this system and how it changes towards stimuli.

Reply to the Referee: We genuinely thank Referee #2 for the valuable comments. Our photo-detachable adhesion mechanism is that, through the coordination interaction among iron ions, CNFs, and PAA chains is dynamically dissociated and restructured along with the transformation of the valence state of iron ions, imparting the reconfiguration of the supramolecular network inside the hydrogel, further making the cohesive energy of hydrogel self-regulatable. We are sorry for the confusing description in our original manuscript. We have modified the “topological network structure” to “supramolecular network structure” in the revised manuscript.

The confocal images and scanning electron microscopy (SEM) images of the CNF-DA/PAA@Fe³⁺ hydrogel exhibited the remarkable and flexible morphology transformation from the initial entangled and homogeneous dense microstructure (without UV radiation) to a subsequently loose porous zone on its surface (after UV radiation), and finally to a dense and compact microstructure again (after air-oxidation) (Revised Fig. 2a). For facilitating the evident visualization of this structural transformation process, we performed in situ Raman analysis of the CNF-DA/PAA@Fe³⁺ hydrogel by switching UV light on one side of the non-UV radiated hydrogel during the dynamic introduction of UV light (Revised Fig. 2b). The peak intensity at 1680 cm⁻¹ (belong to carboxyl groups) enhanced and displayed the graded distinction (after exposure to UV light for 5 mins); with the continuation of UV radiation, the initial blue area (with lower peak intensity value) completely turned into red (with higher peak intensity value), indicative of the noticeable increase of free carboxyl groups in the CNF-DA/PAA@Fe³⁺ hydrogel as UV light gradually penetrated; finally when the resultant CNF-DA/PAA@Fe³⁺ hydrogel was placed in the air, the red area (with higher peak intensity value) gradually returns to the original blue, suggesting its reversible and tunable network structure. Additionally, the right shift of the UV absorbance band from 402 nm (before UV) to 406 nm (after UV), and then left to 402 nm (after air), suggesting the structure transformation of the hydrogel (Revised Fig. 2c). Small-angle X-ray scattering (SAXS) was further conducted to investigate the subtle transformation of the supramolecular network triggered by UV radiation (Revised Figure 2d). A sharp SAXS peak at $q \approx 0.04 \text{ \AA}^{-1}$ was observed in the fabricated hydrogel before UV radiation, indicating a uniform and strong coordination-bond supramolecular network structure in the hydrogel

in the non-UV state, Fe^{3+} ions are transformed into Fe^{2+} ions after being stimulated by UV light, which makes the tightly entangled supramolecular network of CNF-DA and PAA chains loosely deformed. These results were also demonstrated by the XRD analysis of the CNF-DA/PAA@ Fe^{3+} hydrogel. Such a striking morphological transformation is associated with the significant differences in the interactions of CNF chains, Fe^{3+} ions, and PAA chains, and thus supramolecular network evolution of the hydrogel triggered by UV radiation and air oxidation.

To reveal the inherent driving mechanism of structural transformation of the photo-detachable hydrogel based on the P.F. reaction, we carried out X-ray photoelectron spectroscopy (XPS) analysis of the hydrogel during the UV radiation and air-oxidation processes (Revised Fig. 2e). The appearance of the peak at 710.6 eV of the hydrogel after UV radiation demonstrated the generation of Fe^{2+} in the system, verifying that Fe^{3+} was partially reduced to Fe^{2+} triggered by UV light, in which two free radicals ($\text{O}_2\cdot^-$ and $\text{HO}\cdot^-$) also participate in the P.F. reaction process (Supplementary Fig. 6). To identify the functional groups that participated in this process, the two-dimensional correlation spectra (2D-COS) was employed to enhance FTIR spectra analysis of the hydrogel under the UV radiation¹ (Revised Fig. 2f, g). The cross-peak (1682, 3200) in the synchronous and asynchronous maps was both positive. According to Noda's judging rule, $-\text{COOH}$ and $-\text{OH}$ groups were the dominant functional groups that participated in the UV radiation process, in which $-\text{COOH}$ groups of CNF-DA chains were more susceptible to UV light. Additionally, the solid-state ^{13}C NMR spectra of the CNF-DA/PAA@ Fe^{3+} hydrogel exhibited that two characteristic peaks at 170 and 200 ppm, assigned to $-\text{COOH}$ and $-\text{CHO}$, respectively, do appear in the hydrogel after UV radiation, indicative of the environment variation of $-\text{COOH}$ groups and the generation of $-\text{CHO}$ in the hydrogel triggered by UV light (Revised Fig. 2h). These results jointly revealed that Fe^{3+} ions were transformed into Fe^{2+} ions stimulated by UV light, and the coordination interactions with CNF-DA and PAA chains were resultantly dissociated; Subsequently, as the resultant hydrogel was exposed to air, the Fe^{2+} ions were oxidized to Fe^{3+} ions, leading to the reconfiguration of supramolecular network of the CNF-DA/PAA@ Fe^{3+} hydrogel under air. Along with this process, UV light acted as a functional factor in the photo-Fenton reaction between CNF-DA chains and Fe^{3+} ions, which triggered the valence state transformation of Fe ions in the system. Meanwhile, the introduction of CNF served as a reducing agent in the photo-Fenton reaction and meanwhile contributed to the reversibility and stability of coordination interactions in the CNF-DA/PAA@ Fe^{3+} hydrogel.

Revised Figure 2 **a** Confocal images and SEM images of the CNF-DA/PAA@Fe³⁺ hydrogel during the UV radiation and air-oxidation process showing the change of microstructures of the photo-detachable hydrogel. Scale bar, 20 μm . **b** Two-dimensional Raman image from the $-\text{COOH}$ intensity (1680 cm^{-1}) of the CNF-DA/PAA@Fe³⁺ hydrogel during the UV radiation and air-oxidation process. Scale bar, 5 μm . **c** Ultraviolet-visible spectroscopy (UV-vis) spectra of the CNF-DA/PAA@Fe³⁺ hydrogel during the UV radiation and air-oxidation process. **d** SAXS curves for the photo-detachable hydrogels during the UV radiation and air-oxidation process. **e** XPS of Fe 2p regions of the CNF-DA/PAA@Fe³⁺ hydrogel during the UV radiation and air-oxidation process. **f, g** 2DCOS synchronous and asynchronous spectra generated from the FTIR spectra during the UV radiation and air-oxidation process. In 2DCOS spectra, the warm color (red) represents positive intensities, while the cold color

(blue) represents negative intensities. **h** ^{13}C NMR spectra analysis of the CNF-DA/PAA@Fe $^{3+}$ hydrogel during the UV radiation and air-oxidation process.

Updates to the revised manuscript: According to Referee #2's comments, we have added a more careful description to further explain the supramolecular network in the revised manuscript on Pages 6–8.

“The CNF-DA/PAA@Fe $^{3+}$ hydrogel with 2% Fe $^{3+}$ content was demonstrated to deliver the best comprehensive performance. All data analysis is based on the CNF-DA/PAA@Fe $^{3+}$ hydrogel sample with 2% Fe $^{3+}$ content (the mass fraction relative to the mass of CNF-DA is 2 wt%) and 80% water content (the mass fraction relative to the mass of the hydrogel is 80 wt%) if not specified otherwise (Supplementary Figs. 4–6). External properties of materials are strongly linked to their subtle structural changes. The confocal images and scanning electron microscopy (SEM) images of the CNF-DA/PAA@Fe $^{3+}$ hydrogel exhibited remarkable and flexible morphology transformation from the initial entangled and homogeneous dense microstructure (without UV radiation) to a subsequently loose porous zone on its surface (after UV radiation), and finally to a dense and compact microstructure again (after air-oxidation) (Fig. 2a and Supplementary Fig. 7). For facilitating the evident visualization of this structural transformation process, we performed in situ Raman analysis of the CNF-DA/PAA@Fe $^{3+}$ hydrogel by switching UV light on one side of the non-UV radiated hydrogel during the dynamic introduction of UV light (Fig. 2b and Supplementary Fig. 8). Upon the UV radiation, the peak intensity at 1680 cm^{-1} (belong to carboxyl groups) exhibited a distinct enhancement, with the initial blue area (with low peak intensity value) being turned into green and completely to red (with high peak intensity value), indicative of the generation of more free carboxyl groups and thus the noticeable looser structure in the CNF-DA/PAA@Fe $^{3+}$ hydrogel triggered by UV light. As the hydrogel was placed in the air, the hydrogel showed an opposite trend, suggesting its reversible and tunable network structure. These results were also demonstrated by combined usage of ultraviolet-visible spectroscopy (UV-vis) spectra, small-angle X-ray scattering (SAXS), and X-ray diffraction (XRD) analysis of the CNF-DA/PAA@Fe $^{3+}$ hydrogel (Fig. 2c, 2d and Supplementary Figs. 9 and 10). Such a striking morphological transformation is associated with the significant differences in the interactions of CNF chains, Fe $^{3+}$ ions, and PAA chains, and thus supramolecular network evolution of the hydrogel triggered by UV radiation and air oxidation^{13–18}.”

“To reveal the inherent driving mechanism of structural transformation of the photo-detachable hydrogel based on the P.F. reaction, we carried out the X-ray photoelectron spectroscopy (XPS) analysis of the hydrogel during the UV radiation and air-oxidation processes (Fig. 2e). The appearance of the peak at 710.6 eV of the hydrogel after UV radiation demonstrated the generation of Fe $^{2+}$ in the system, verifying that Fe $^{3+}$ was partially reduced to Fe $^{2+}$ triggered by UV light, in which two free radicals ($\text{O}_2^{\cdot-}$ and $\text{HO}^{\cdot-}$) also participate in the P.F. reaction process (Supplementary Fig. 11). As the photo-detachable hydrogel was exposed to air, the Fe $^{3+}$ was partially oxidized to Fe $^{2+}$, facilitating the self-recovery of the supramolecular network in the

hydrogel. To identify the functional groups that participated in the photodetachment process, the two-dimensional correlation spectra (2D-COS) were employed to enhance FTIR spectra analysis of the hydrogel under UV radiation (Fig. 2f, g)¹⁹. The cross-peak (1682, 3200) in the synchronous and asynchronous maps was both positive. According to Noda's judging rule, -COOH and -OH groups were the dominant functional groups that participated in the UV radiation process, in which -COOH groups of CNF-DA chains were more susceptible to UV light. Additionally, the solid-state ¹³C NMR spectra of the CNF-DA/PAA@Fe³⁺ hydrogel exhibited that two characteristic peaks at 170 and 200 ppm, assigned to -COOH and -CHO, respectively, do appear in the hydrogel after UV radiation, indicative of the environment variation of -COOH groups and the generation of -CHO in the hydrogel triggered by UV light (Fig. 2h and Supplementary Fig. 12)^{20, 21, 22}. These results jointly revealed that Fe³⁺ ions were transformed into Fe²⁺ ions stimulated by UV light, and the coordination interactions of iron ions with CNF-DA and PAA chains were resultantly dissociated; Subsequently, as the resultant hydrogel was exposed to air, the Fe²⁺ irons were oxidized to Fe³⁺ irons, leading to the reconfiguration of supramolecular network of the CNF-DA/PAA@Fe³⁺ hydrogel in air. Along with this process, UV light acted as a functional factor in the photo-Fenton reaction between CNF-DA chains and Fe³⁺ ions, which triggered the valence state transformation of Fe ions in the system. Meanwhile, CNF served as a reducing agent in the photo-Fenton reaction, which further contributed to the reversibility and stability of the coordination interactions in the CNF-DA/PAA@Fe³⁺ hydrogel."

References:

1. Wang, J., Wu, B., Wei, P., Sun, S. & Wu, P. Fatigue-free artificial ionic skin toughened by self-healable elastic nanomesh. *Nat. Commun.* **13**, 4411 (2022).
2. In figure 3a,b,c, the hydrogel seems have different color. Whether the amount of Fe would influence the color of the hydrogel and the performance of the hydrogel?

Reply to the Referee: We thank Referee #2 for the thoughtful suggestions. In our hydrogel hybrid system, Fe³⁺ content did affect the color, mechanical, and adhesion performance of the fabricated hydrogel. As a validation, we explored the color, mechanical and adhesive performance of CNF-DA/PAA@Fe³⁺ hydrogels with various Fe³⁺ contents. According to the mass ratio of Fe³⁺ to the mass of the CNF-DA as x, the prepared hydrogel is denoted as CNF-DA/PAA@Fe³⁺_x, where x is 0.5%, 1%, 2%, and 4%. Also, we replenished a detailed elaboration on the effect of Fe³⁺ on the mechanics.

(1) Color of CNF-DA/PAA@Fe³⁺ hydrogels with different Fe contents.

The digital image of the various hydrogels with different Fe³⁺ ratios is shown in Figure R11. It can be observed that the overall color of the hydrogels changes from light yellow to dark brown with the gradual increase of Fe³⁺ content in the various CNF-DA/PAA@Fe³⁺ hydrogel.

Figure R11 Digital image of the CNF-DA/PAA@Fe³⁺ hydrogels with different Fe³⁺ contents.

(2) Mechanical properties of CNF-DA/PAA@Fe³⁺ hydrogels with different Fe contents.

As shown in Figure R12a, the CNF-DA/PAA@Fe³⁺_0.5% hydrogel is flattened and has small cracks in the hydrogel center after being compressed under the action of external force, showing significant irrecoverability and poor mechanical properties. In sharp contrast, CNF-DA/PAA@Fe³⁺_2% hydrogel maintains high integrity without distinct fracture and cracks under external force despite larger deformation, indicating its excellent mechanical strength and good recoverability. These impressive physical behaviors suggest that the corporation of Fe³⁺ has a positive effect on enhancing the mechanical strength of the CNF-DA/PAA@Fe³⁺ hydrogel.

To further explore the Fe³⁺ on the mechanical properties of hydrogels, we performed the quantitative analysis of these hydrogels in tensile and compressive tests. As shown in Figure R12b, the ultimate stress of the CNF-DA/PAA@Fe³⁺_0.5% hydrogel is 0.013 MPa at a maximum fracture strain of 1022%. It can be seen that both the maximum tensile stress and strain of the hydrogel show an improving trend with the increase of Fe³⁺ contents. In particular, when the mass of doped Fe³⁺ accounted for 2% of the mass of the CNF-DA, the resultant CNF-DA/PAA@Fe³⁺_2% hydrogel exhibits a fracture strain of 1850% and maximum stress of 0.046 MPa, showing 3.5-fold and 45% increase compared to the CNF-DA/PAA@Fe³⁺_0.5% hydrogel, suggesting a positive role of Fe³⁺ for the mechanical properties of the developed hydrogel. Meanwhile, the toughness of CNF-DA/PAA@Fe³⁺_2% hydrogel is 0.55 MJ m⁻³, respectively, which is 7.3 times of the corresponding CNF-DA/PAA@Fe³⁺_0.5% hydrogel (0.075 MJ m⁻³), implying a significant improved mechanical toughness. However, the tensile properties and toughness of the obtained hydrogels decrease when Fe³⁺ content is continuously added, which is due to the heterogeneous system caused by excess Fe³⁺.

Furthermore, this interesting mechanical behavior concerning the positive effect of Fe³⁺ was also found on the compressive performance of the hydrogel, that is, with the increase of Fe³⁺ content, the compressive strength of the hydrogel showed a trend of first increasing and then decreasing (Figure R12c). Especially, the compressive strength of the CNF-DA/PAA@Fe³⁺_2% hydrogel is 18.1 MPa, an increase of 1.5 times compared to that of 12.3 MPa for the CNF-DA/PAA@Fe³⁺_0.5% hydrogel (Figure R12d), indicating the increase of Fe³⁺ content improves the compressive strength, which is also good agreement with the results of the mechanical demonstration in Figure R12a.

Figure R12 a Digital images of the CNF-DA/PAA@Fe³⁺ hydrogels before and after compression. **b** Tensile stress-strain curves of the CNF-DA/PAA@Fe³⁺ hydrogels with different Fe contents. **c** Compressive stress-strain curves of the CNF-DA/PAA@Fe³⁺ hydrogels with different Fe contents. **d** Comparison of toughness and compressive strength of the CNF-DA/PAA@Fe³⁺ hydrogels with different Fe contents. Values represent the mean and the standard deviation (n = 3).

(3) Adhesive properties of CNF-DA/PAA@Fe³⁺ hydrogels with different Fe contents.

To explore the adhesive properties of various CNF-DA/PAA@Fe³⁺ hydrogels with different Fe contents, we performed 90-degree peeling, lap-shear, and tensile tests on the substrates of freshly excised porcine skin and engineering glass. As shown in Figure R13a, the CNF-DA/PAA@Fe³⁺_0.5% hydrogel can be easily removed from the engineering glass, showing a weak adhesive strength. In sharp contrast, the CNF-DA/PAA@Fe³⁺_2% hydrogel exhibits a pull on the glass observed without visible fracture when detached from the glass, indicating an enhanced adhesive strength and mechanical toughness because CNF-DA and Fe³⁺ form a strong coordination interaction and reinforced the interface interaction. However, the CNF-DA/PAA@Fe³⁺_4% hydrogel presents a decayed peeling behavior compared to the CNF-DA/PAA@Fe³⁺_2% hydrogel, since an excess of Fe³⁺

leads to a non-uniform network system.

Furthermore, we quantitatively evaluated the adhesive properties of prepared hydrogels with different CNF-DA contents by 90-degree peeling, lap-shear, and tensile tests on the substrates of freshly excised porcine skin and engineering glass (Figure R13b–g). According to these force-displacement curves, we can find that, when the mass ratio of added Fe^{3+} to CNF-DA is less than 2%, as the Fe^{3+} content increases, the peel force, shear force, and tensile force all show a rising trend, which because that the introduction of Fe^{3+} has a positive effect on the cohesive energy of the hydrogel, facilitating more active groups to bond with functional groups on the substrates of skin surface and glass. As expected, the peeling force, shear force, and tensile force of hydrogels show an attenuation when Fe^{3+} is continued to be added, which is ascribed to the excess Fe^{3+} resulting in a non-uniform hydrogel, enabling it easily friable to a weak adhesion on the skin and glass.

More intuitively, the interfacial toughness, shear strength, and tensile strength of various CNF-DA/PAA@ Fe^{3+} hydrogels on the substrates of skin and glass were correspondingly obtained, respectively, and a similar adhesion trend to that of the force-displacement curves could be found (Figure R13h, i). Specifically, the CNF-DA/PAA@ Fe^{3+} _2% hydrogel exhibits 88 J m^{-2} for interfacial toughness, 80 kPa for shear strength, and 80 kPa for shear strength on the skin (80 J m^{-2} for interfacial toughness, a 77 kPa for shear strength, and a 68 kPa for shear strength on the glass). These impressive adhesion observations reveal a CNF-DA/PAA@ Fe^{3+} hydrogel with excellent adhesion properties through optimal Fe content, that is, the ratio of Fe^{3+} to the mass of the CNF-DA as 2%.

Combined with these pictorial techniques and quantitative tests of mechanical and adhesive performance of developed hydrogels with different Fe contents, jointly demonstrated the fact that Fe^{3+} plays a vital role in enhancing mechanical and adhesive properties of the CNF-DA/PAA hydrogel, enabling to provide strong complexation interaction between cellulose chains and PAA chains, which is conducive to the improvement of the cohesive energy of the hydrogel system to promote its overall performance and practical use.

Figure R13 **a** Digital images of the CNF-DA/PAA@Fe³⁺ hydrogels peel from engineering glass. **b–d** 90-degree peel, lap-shear, and tensile force-displacement curves of the CNF-DA/PAA@Fe³⁺ hydrogels prepared with different material ratios on the substrate of freshly excised porcine skin. **e–g** 90-degree peel, lap-shear, and tensile force-displacement curves of the CNF-DA/PAA@Fe³⁺

hydrogels prepared with different material ratios on the substrate of engineering glass. **h, i** Comparison of interfacial toughness, shear strength, and tensile strength of the CNF-DA/PAA@Fe³⁺ hydrogels prepared with different material ratios on the substrates of skin and glass. Values represent the mean and the standard deviation (n = 3).

Benefiting from strong coordination interactions between Fe ions and carboxyl groups, as shown in Figure R14a, the CNF-DA/PAA@Fe³⁺_2% hydrogel can be cyclically bent on fingers without visible fragmentation, suggesting its extraordinary flexibility and mechanical compliance. In addition, the CNF-DA/PAA@Fe³⁺_2% hydrogel enables it to be stretched by 10 times under external force without breaking, demonstrating its strong stretchability and durability (Figure R14b). Thanks to the dynamic and reversible metal coordination, the fabricated hydrogel can still restore the original intact shape after being torn apart, presenting an excellent self-healing performance of the CNF-DA/PAA@Fe³⁺_2% hydrogel (Figure R14c). In addition, the color of the CNF-DA/PAA@Fe³⁺_2% hydrogel with different thicknesses is different. As the thickness of the CNF-DA/PAA@Fe³⁺_2% hydrogel increase, the color of the CNF-DA/PAA@Fe³⁺_2% hydrogel changes from light yellow to dark brown. As shown in Figure R14, the thickness of the hydrogel is ~2.5 mm in Figure 2a, ~1 mm in Figure 2b, ~25 mm in Figure 2c, respectively. Of note, the CNF-DA/PAA@Fe³⁺_2% hydrogel with a larger thickness presents more excellent viscoelasticity and self-healing properties.

Figure R14 **a** Photographs of the CNF-DA/PAA@Fe³⁺_2% hydrogel with excellent flexibility and adhesion ability to the human knuckle. The thickness of CNF-DA/PAA@Fe³⁺_2% hydrogel is ~2.5 mm. **b** Photographs of the CNF-DA/PAA@Fe³⁺_2% hydrogel with high stretchability. The thickness of CNF-DA/PAA@Fe³⁺_2% hydrogel is ~1 mm. **c** Photographs of the self-healing performance of the CNF-DA/PAA@Fe³⁺_2% hydrogel. The thickness of CNF-DA/PAA@Fe³⁺_2% hydrogel is ~25 mm.

Updates to the revised manuscript: According to Referee #2's comments, we have added more careful descriptions in the revised manuscript on Pages 6 and 12.

"The CNF-DA/PAA@Fe³⁺ hydrogel with 2% Fe³⁺ content was demonstrated to deliver the best comprehensive performance. All data analysis is based on the CNF-DA/PAA@Fe³⁺ hydrogel

sample with 2% Fe^{3+} content (the mass fraction relative to the mass of CNF-DA is 2 wt%) and 80% water content (the mass fraction relative to the mass of the hydrogel is 80 wt%) if not specified otherwise (Supplementary Figs. 4–6). External properties of materials are strongly linked to their subtle structural changes. The confocal images and scanning electron microscopy (SEM) images of the CNF-DA/PAA@ Fe^{3+} hydrogel exhibited remarkable and flexible morphology transformation from the initial entangled and homogeneous dense microstructure (without UV radiation) to a subsequently loose porous zone on its surface (after UV radiation), and finally to a dense and compact microstructure again (after air-oxidation) (Fig. 2a and Supplementary Fig. 7).” On Page 6

“**Fig. 3 Mechanical performance of the photo-detachable hydrogel.** **a** Photographs of the CNF-DA/PAA@ Fe^{3+} hydrogel with excellent flexibility and adhesion ability to the human knuckle. The thickness of CNF-DA/PAA@ Fe^{3+} hydrogel is ~2.5 mm. **b** Photographs of the CNF-DA/PAA@ Fe^{3+} hydrogel with high stretchability. The thickness of CNF-DA/PAA@ Fe^{3+} hydrogel is ~1 mm. **c** Photographs of the self-healing performance of the CNF-DA/PAA@ Fe^{3+} hydrogel. The thickness of CNF-DA/PAA@ Fe^{3+} hydrogel is ~25 mm.” on Page 12

3. In the past, there are many work using Fe ion coordinated with COOH to fabricate high performance hydrogels. The novelty of this work should be emphasized.

Reply to the Referee: We thank Referee #2 for the valuable suggestions. We propose a rapid, reliable, and universal strategy to achieve both reversible tough adhesion and easy detachment by regulating its supramolecular network reconfiguration via the cellulose nanofiber (CNF)/ Fe^{3+} (as a Fenton-like reagent)-mediated dopamine-poly(acrylic acid) dynamic hydrogel. Despite many similar achievements as mentioned, so far, we fail to find a similarity between these works and our current effort, except for the incorporation of Fe ions and COOH. To illustrate the novelty of our study, two representative works (**Ref. 1:** Gao, Y., Wu, K. & Suo, Z. Adv. Mater. 2019, 31, 1806948; **Ref. 2:** Xu, R., et al. Adv. Mater. 2022, 34, 2108889) are selected for detailed discussion. The CNF-mediated photo-Fenton reaction for the reversible photo-detachable dynamic adhesion in our work is substantially different from the previous work, as briefly quoted below.

Ref. 1 Adv. Mater. 2019, 31, 1806948. (Photodetachable adhesion aqueous solution)

Abstract

Peeling from strong adhesion is hard, and sometimes painful. Herein, an approach is described to achieve both strong adhesion and easy detachment. The latter is triggered, on-demand, through an exposure to light of a certain frequency range. The principle of photodetachable adhesion is first demonstrated using two hydrogels as adherends. Each hydrogel has a covalent polymer network, but does not have functional groups for bonding, so that the two hydrogels by themselves adhere poorly. The two hydrogels, however, adhere strongly when an aqueous solution of polymer chains is spread on the surfaces of the hydrogels and is triggered to form a stitching polymer network in

situ, in topological entanglement with the pre-existing polymer networks of the two hydrogels. The two hydrogels detach easily when the stitching polymer network is so functionalized that it undergoes a gel–sol transition in response to a UV light. For example, two pieces of alginate–polyacrylamide hydrogels achieve adhesion energies about 1400 and 10 J m⁻², respectively, before and after the UV radiation. Experiments are conducted to study the physics and chemistry of this strong and photodetachable adhesion, and to adhere and detach various materials, including hydrogels, elastomers, and inorganic solids.

[REDACTED]

Figure 1. Photodetachable adhesion of two pieces of polyacrylamide (PAAm) hydrogels.

In this work (**Ref.1**), the two PAAm hydrogels with no functional groups for bonding adhere poorly. An aqueous solution of PAA chains with certain rheology was added on the surface of two pieces of pristine hydrogels and waited for its permeation into hydrogels. In these circumstances, another aqueous solution of Fe³⁺ and citric acid with a controlled pH value of 1.5~6 was needed to diffuse into the hydrogel to form the coordination complexes to strongly adhere the two hydrogels. Under these conditions, concerning the time-consuming permeation process and accessibility of the diffusion substrate, this aqueous solution-enabled adhesion can only be obtained in the aqueous adherend system after a long waiting time. Particularly, given the acid environment and pre-prepared permeation of the PAA solution, it is extremely difficult to apply this photo-detachable system to human skin, leading to limited applications in self-powered e-skin. More importantly, after the UV radiation, the Fe³⁺-cross-linked PAA network dissociated into PAA chains again, and the dense interfacial zone between the two hydrogels enabled by cross-linking of the PAA network disappeared for photodetachment.

Ref. 2 Adv. Mater. 2022, 34, 2108889. (A universal strategy for growing a tenacious hydrogel coating from a sticky initiation layer)

Abstract

Controllably coating the surfaces of substrates/medical devices with hydrogels exhibits great application potential, but lacks universal techniques. Herein, a new method, namely ultraviolet-triggered surface catalytically initiated radical polymerization (UV-SCIRP) from a sticky initiation layer (SIL) (SIL@UV-SCIRP), is proposed for growing hydrogel coatings. The method involves three key steps: 1) depositing a sticky polydopamine/Fe³⁺ coating on the surface of the substrates-SIL, 2) reducing Fe³⁺ ions to Fe²⁺ ions as active catalysts by UV illumination with the assistance of citric acid, and 3) conducting SCIRP in a monomer solution at room temperature for growing hydrogel coatings. In this manner, practically any substrate's surface (natural or artificial materials) can be modified by hydrogel coatings with controllable thickness and diverse compositions. The hydrogel coatings exhibit good interface bonding with the substrates and enable easy changes in their wettability and lubrication performances. Importantly, this novel method facilitates the smooth growth of uniform hydrogel lubrication coatings on the surface of a range of medical devices with complex geometries. Finally, as a proof-of-concept, the slippery balls coated with hydrogel exhibited smooth movement within the catheter and esophagus. Hence, this method can prove to be a pioneering universal modification tool, especially in surface/interface science and engineering.

[REDACTED]

Figure 1. Mechanism of SIL@UV-SCIRP method.

In this work (Ref.2), to in-situ fabrication the hydrogel layer on the surface of substrates/medical devices, the deposition of PDA coating on the substrate and presence of Fe³⁺ and citric acid with a control pH value of 4.0 enables the substrate deposited with PDA coordinate with Fe³⁺. Upon the UV radiation, The Fe³⁺ ions within PDA/CA-Fe³⁺ can then be reduced to Fe²⁺ ions using UV illumination with the assistance of citric acid, generating an active sticky catalyst initiation layer (SIL). Subsequently, the solid-liquid interface redox reaction is performed between Fe²⁺ and S₂O₈²⁻ ions in monomer solution to generate radical anion SO₄^{-•}, along with the dramatic reduction of the

decomposition activation energy, to initiate monomer polymerization at room temperature. Intuitively, the poly(acrylic acid)–poly(acrylamide) (PAA-PAM) hydrogel coating can be successfully grown on the surface of the Ti substrate by the SIL@UV-SCIRP method.

Our work (photo-detachable CNFs-mediated hydrogel enabled by P.F. reaction)

Abstract

Self-powered skin attachable and detachable electronics are under intense development to enable the internet of everything and everyone in new and useful ways. Existing on-demand separation strategies rely on complicated pretreatments and physical properties of the adherends, achieving detachable-on-demand in a facile, rapid, and universal way remains challenging. To overcome this challenge, we propose a rapid, reliable, and universal strategy to achieve both reversible tough adhesion and easy detachment by regulating its supramolecular network via the cellulose nanofiber (CNF)/Fe³⁺ (as a Fenton-like reagent)-mediated dopamine-poly(acrylic acid) dynamic hydrogel. This strategy enables the simple and rapid fabrication of strong yet reversible hydrogels with tunable toughness ((Value_{max}-Value_{min})/Value_{max} of up to 88%), on-demand adhesion energy ((Value_{max}-Value_{min})/Value_{max} of up to 98%), and stable conductivity up to 12 mS m⁻¹. We further extend this strategy to fabricate different CNF/Fe³⁺-based hydrogels from various biomacromolecules and petroleum polymers, and shed light on exploration of fundamental dynamic supramolecular network reconfiguration. Simultaneously, we prepare an adhesive-detachable triboelectric nanogenerator (TENG) as a human-machine interface for a self-powered wireless monitoring system based on this strategy, which can acquire the real-time, self-powered monitoring, and wireless whole-body movement signal, opening up possibilities for diversifying potential applications in e-skins and intelligent devices.

Figure 1. Reversible photo-detachable CNF-DA/PAA@Fe³⁺ dynamic hydrogel based on a light-driven supramolecular network engineering strategy via photo-Fenton reaction.

In our work, we propose a **rapid, reliable, and universal strategy to achieve both reversible tough adhesion and easy photodetachment of the hydrogel by regulating its supramolecular network via the cellulose nanofibers (CNFs)-mediated photo-Fenton-like reaction.** The developed strongly adhesive CNF-DA/PAA@Fe³⁺ hydrogel is composed of Fe³⁺, CNF-DA, and PAA networks, in which the CNF network serves as a Fenton-like reagent and the supporting framework of the hydrogel, and produces coordination interactions with Fe³⁺ for excellent mechanical and adhesive performance. Upon the UV radiation, the Fe³⁺ ions are reduced to Fe²⁺ ions in the photo-Fenton-like reaction, leading to dissociation and reconstruction of the Fe ions and CNF networks, further the

reconfiguration of the supramolecular network inside the hydrogel. As such, we achieve a strong and immediate adhesion on various substrates ranging from aqueous and nonaqueous systems, and rapid and human-friendly detachment of the hydrogel. More attractively, the mechanical and adhesion energy is tunable for diversifying applications, and the detached hydrogel can be easily self-recovered by air oxidation of the hydrogel within 5 min for reuse. Furthermore, the biocompatible and human-friendly hydrogel as an ionic conductor in flexible sensors or triboelectric nanogenerators can be applied in electronic skins to stably monitor human movements and physiological signals with a high signal-to-noise ratio, creating a wide-open space for potential applications, especially for personal health monitoring, patient rehabilitation, athletic performance monitoring, and recreational human motion tracking.

To illustrate the originality of our current work, we summarize the major differences between our work and the two previous works reported by **Ref. 1 (Adv. Mater. 2019, 31, 1806948)** and **Ref. 2 (Adv. Mater. 2022, 34, 2108889)**, as shown below.

- 1. CNFs-mediated supramolecular reconfiguration regulatory mechanism (Our work) VS interfacial iron-carboxyl chemistry regulatory mechanism and Fe^{2+} - $\text{S}_2\text{O}_8^{2-}$ redox reaction (Ref.1 & Ref.2).** In our work, cellulose nanofibrils (CNFs), as the supporting framework of hydrogel, are also the key to the regulation of the valence state of iron ions. CNF served as a reducing agent in the photo-Fenton reaction and meanwhile contributed to the reversibility and stability of coordination interactions in the CNF-DA/PAA@ Fe^{3+} hydrogel. Upon the UV radiation, due to the transformation of the valence state of Fe ions, the coordination interaction among Fe ions, CNFs, and PAA chains is dynamically dissociated and reconstructed, affecting the supramolecular network reconfiguration inside the hydrogel, further self-regulating the cohesive energy of the hydrogel. This is the first time that the photo-Fenton reaction involving the transformation of the Fe ions valence via CNFs is introduced to regulate the supramolecular network reconfiguration within the hydrogel, thus regulating the mechanical and adhesive properties of the hydrogel. By sharp contrast, Ref. 1 and Ref. 2 focused on the interfacial iron-carboxyl chemistry with Fe^{3+} -Citric acid aqueous solution and adherends, and uncontrolled-redox reaction between Fe^{2+} and $\text{S}_2\text{O}_8^{2-}$.
- 2. CNFs-mediated hydrogel with widely tunable mechanical and adhesive performance (Our work) VS single photo-detachable adhesion performance (Ref.1 & Ref.2).** In our work, the introduction of CNFs endows the hydrogel with excellent yet tunable mechanical and adhesive properties. For the first time, we easily utilize the photo-Fenton-like specialty of CNFs and iron ions in combination with UV light radiation to rapidly obtain dynamic hydrogels with tunable mechanical and adhesive properties, instead of monotonous photo-detachable glue in Ref.1. The mechanical and adhesive performance of the hydrogel is tunable by tuning the Fe contents, water content, and CNF/DA contents in the hydrogel, indicating excellent flexibility, adaptability, and applicability of our hydrogel. However, the adhesion energy is related to the permeation and gelation of PAA and the pH of the Fe^{3+} -citric acid serving as a monotonous adhesive glue in Ref.1, which can hardly satisfy the requirements of adjustable mechanical and adhesive properties for wide applications.
- 3. Fast adhesion process and rapid self-recovery adhesive performance (Our work) VS time-consuming and unrecoverable adhesion process (Ref.1 & Ref.2).** The hydrogels prepared in our work exhibit adhesion immediately, which is much faster and more efficient than the adhesion process reported in Ref. 1 (Less than 1 s VS 10 min). The abundant oxygen-containing

functionalized groups on CNFs (especially catechol) can directly produce strong interactions for adhesion with the adherends, which is an efficient path to rapidly form a conformal and perfect adhesion. More importantly, the interfacial adhesion of our hydrogel could be rebuilt upon exposure to white light by air oxidization. The adhesion strength and adhesion energy of the hydrogels can be recovered to their original state with 5-minute exposure to open air. By contrast, the overall adhesion process reported in Ref. 1 and Ref. 2 is extremely time-consuming because the Fe³⁺-citric acid aqueous solution needs to penetrate the adherends and generate coordination interactions with the functional groups of the internal polymer chains of the adherends.

4. **Universality of CNFs-mediated photo-detachable adhesion strategy (Our work) VS aqueous adhesion-detachable strategy (Ref.1 & Ref.2).** In our work, benefiting from introducing polymer chains with versatile functional groups, this strategy applies to a wide variety of hydrogels and adherends. As such, the hydrogel works as a universal gel for facile adhesion to diverse substrates (both aqueous and nonaqueous systems), without any additional interfacial chemical designs or pretreatments for adherends. On the other hand, the photo-detachable adhesion strategy based on the P.F. reaction is applicable to other biomacromolecules and petrochemical polymers, including gelatin, chitosan, alginate, starch, acrylamide (PAAm), and polyvinyl alcohol (PVA), in which CNF/Fe³⁺ serves as a Fenton-like reagent. By contrast, Fe³⁺-citric acid aqueous solution in Ref. 1 and Ref. 2 as an adhesive glue is inapplicable to nonaqueous adherends, and the adhesion performance depends on the choice between adherends types and operations.
5. **Multifunctional application (Our work) VS limited application (Ref.1 & Ref.2).** With excellent reversible and tunable features of viscoelasticity, self-healing, and ionic conductance, the dynamic gel demonstrates great potential as a flexible, transparent, designable, and biocompatible sensor device for real-world applications including e-skins, soft robots, energy storage, and intelligent devices. However, the Fe³⁺-Citric acid aqueous solution and hydrogel coating in Ref. 1 and Ref. 2 have limited application due to the applicable environment of the iron-carboxyl chemistry and redox reaction between Fe²⁺ and S₂O₈²⁻.

These major differences between our work and previous works are further listed in the following Table R4 for a clear and fair comparison.

Table R4. Comprehensive comparisons (differences) of our work with the two previously reported papers (Ref.1 and Ref.2).

Product (Study object)	Fabrication Method	Photo-detachable adhesion time (min)	Micro-structure	Photodetachment mechanism	Universality of adherends and strategy	Properties	Applications	Refs
CNF-mediated photo-detachable adhesive hydrogel	In-situ synthesis of hydrogels by copolymerization of CNF-DA and iron ions as the P. F. reagent with acrylic acid	Immediate adhesion; detaching after 5 mins (UV light, 40 mW cm ⁻²)	Reverse transformation, from the dense to loose micro-structure	Supramolecular network reconfiguration inside the CNF-mediated hydrogel to further enable self-regulation of the cohesive energy	1. Diverse substrates (both aqueous and non-aqueous systems) 2. Photo-detachable adhesion strategy applies to other biomacromolecules and petrochemical polymers	1. Reversible mechanical properties; 2. Tunable adhesive properties; 3. Self-healing properties; 4. Ionic conductivity 5. Biocompatibility	1. E-skins, soft robots, energy storage, and intelligent devices; 2. Wound dressing and transdermal drug delivery	This work

Fe ³⁺ -citric acid aqueous solution	FeCl ₃ and citric acid were dissolved in deionized water.	Adhesion by 10 mins penetration; detaching after 3 mins (UV light, 60 mW cm ⁻²)	Solution	Interfacial iron-carboxyl chemistry	Only aqueous systems	Photo-detachable adhesive properties	Need to transfer thin films of devices from a donor to a target substrate	Ref. 1
--	--	---	----------	-------------------------------------	----------------------	--------------------------------------	---	--------

References

- [1] Gao, Y., Wu, K. & Suo, Z. *Adv. Mater.* **2019**, *31*, 1806948.
 [2] Xu, R., et al. *Adv. Mater.* **2022**, *34*, 2108889.

In brief, our universal light-driven supramolecular reconfiguration approach to regulate the adhesion properties of CNF-mediated hydrogels is faster, safer, more reliable, scalable, and designable compared to the methods of regulating the adhesion between two hydrogels through the iron-carboxyl chemistry as reported in Ref.1. Our CNF-mediated hydrogel through this light-driven supramolecular reconfiguration strategy has the combined features of reversible and excellent viscoelasticity, flexible scalability, and satisfied designability, thus **substantially different** from that Fe³⁺-citric acid aqueous solution and SIL@UV-SCIRP method hydrogel coating in previous works (Ref.1 and Ref.2).

Moreover, to give a clearer picture of our innovative work according to the comments of Referee #2, we summarize the novelty of our work as follows:

(a) CNFs-mediated photo-Fenton reaction for the reversible photo-detachable dynamic adhesion

Available position-selective and abundant carboxy groups on the CNF surface offer abundant active sites to bond with Fe ions for coordination complexes in the hydrogel. Upon UV radiation without external additives, the environment of –COOH groups changes and –CHO groups are obtained in the CNF-DA/PAA@Fe³⁺ hydrogel, meanwhile, Fe³⁺ ions were transformed into Fe²⁺ ions in the photo-Fenton reaction. As such, the coordination interaction between CNF-DA, Fe ions, and PAA chains is dynamically transformed through the valence state switching of Fe ions, and the supramolecular network structure inside the hydrogel undergoes the dissociation-reconfiguration, facilitating the triggerable benign detachment of the hydrogel; Attractively, as the resultant hydrogel was exposed to air, the Fe²⁺ ions were oxidized to Fe³⁺ ions, leading to the reconfiguration of supramolecular network of the CNF-DA/PAA@Fe³⁺ hydrogel in air for strong yet reversible adhesion of the hydrogel.

(b) CNFs-mediated hydrogels with widely tunable properties (i.e., self-healing properties, mechanical properties, reversible photo-detachable adhesion behavior, and self-recovery adhesive performance)

The strong coordination interactions and CNF serving as the supporting skeleton of the internal network structure enable the excellent mechanical properties of the hydrogel. The CNF-DA/PAA@Fe³⁺ hydrogel with strong stretchability, durability, and stability displays excellent self-healing performance. Additionally, the mechanical properties and adhesive performance of the hydrogel are tunable by regulating the water or CNF contents in the hydrogel. The developed hydrogel exhibited the maximum mechanical properties (0.57 MJ m⁻³ for toughness, 34.8 MPa for compressive strength) and adhesive performance (1739 J m⁻² for interfacial toughness, 1017 kPa for shear strength, 1286 kPa for tensile strength). Upon the UV radiation, the hydrogel presents dramatical tunability ratios of the interfacial adhesion energy (by up to 94%) and toughness (by up to 83%) and thus

realizing the reversible photo-detachment of the hydrogel with various substrates. Encouragingly, a wide range of adhesion strength of the hydrogel can be achieved (more than 1739 J m^{-2} for interfacial toughness, more than 99% for tunable adhesion ratio) by tuning the UV intensity and radiation time or hydrogel water content, imparting the desired adhesion of the developed hydrogel for multi-scene applications. More attractively, being exposed to air for 5 min, we can achieve the switchable operability of the above processes with associated reversible adhesion, suggesting the great potential of the light-driven engineered hydrogel in rapid and efficient reuse.

(c) Universal and scalable photo-detachable adhesion strategy

The photo-Fenton reagent formed by CNF and iron ions can be introduced into a wide variety of hydrogel matrices, benefiting from the dynamic regulation of CNF chains with multifunctional functional groups and its internal network, and can facilely adhere and detach to diverse substrates (like aqueous and nonaqueous systems), without requiring any interfacial chemical design or pretreatment for adherends (more than 1739 J m^{-2} for interfacial toughness, 99% for adhesion adjustable ratio). CNF/ Fe^{3+} can be served as a Fenton-like reagent to realize the transformation between Fe^{3+} - Fe^{2+} ions under UV radiation, and then enables a tunable supramolecular hydrogel. We prepared different hydrogels consisting of CNF/ Fe^{3+} as a Fenton-like reagent, including those based on gelatin, chitosan, alginate, starch acrylamide (PAAm), and polyvinyl alcohol (PVA). All fabricated hydrogels show good potential in strong yet reversible flexibility and interfacial toughness, and the tensile toughness and interfacial toughness tunability ratios of all these hydrogels exceed 88%. The universality of this strategy will facilitate the reliable production of photo-detachable dynamic hydrogels from a variety of resources ranging from biomass to synthesized polymers.

In addition, we appreciate Referee #2's constructive suggestions and comments regarding the current manuscript. At the same time, we are aware that the academic quality of our manuscript should be further improved to dispel these doubts, and that is what we've been working on in the past few months. While addressing all the comments, we have further improved this work given more comprehensive experiments and simulations (Figure R1). **The highlights of our revisions added in the Revised Manuscript are summarized below:**

I. Computational simulation to evaluate the binding energies between Fe ions (Fe^{3+} , Fe^{2+}) and other macromolecules and polymers including gelatin, chitosan, alginate, starch, PAAm, and PVA

We further investigated the applicability of this strategy to different kinds of biomacromolecules and polymers to confirm its universality. Gaussian simulations were employed to get further insights into the binding energy between Fe ions (Fe^{3+} , Fe^{2+}) and gelatin, chitosan, alginate, starch, PAAm, and PVA, which creates a chance of preparation of reversible photo-detachable dynamic hydrogel from a variety of resources ranging from biomass to synthesized polymers.

II. Photo-detachable adhesive tests of a variety of CNFs-mediated adhesive hydrogels

We carried out tensile tests and 90-degree peeling tests to quantitative analysis of the stretchability and interfacial toughness of gelatin-based, chitosan-based, alginate-based, starch-based, PAAm-based, PVA-based hydrogels. All developed hydrogels formed by CNF- Fe^{3+} photo-Fenton reagents and

biomass materials or petroleum-based polymers exhibit excellent yet reversible flexibility and photodetachment performance.

III. Demonstration of the coordination interaction in the hydrogel after the air oxidization process

We added a series of compressive experimental analyses including in-situ Raman spectroscopy, X-ray photoelectron spectroscopy (XPS), 2D-COS synchronous and asynchronous spectroscopy, confocal (CLSM) and scanning electron microscopy (SEM) images, small-angle X-ray scattering (SAXS), the solid-state ^{13}C NMR spectroscopy, and ultraviolet-visible (UV-vis) spectroscopy to further demonstrate supramolecular reconfiguration in the CNF-DA/PAA@Fe $^{3+}$ hydrogel before and after UV radiation as well as after air oxidation (See Revised Figure 2).

IV. Tunable and stable mechanical and adhesive performance by regulating water contents of the hydrogel

We carried out tensile, compression, and 90-degree peeling tests to explore the mechanical and adhesive performance of the CNF-DA/PAA@Fe $^{3+}$ hydrogel with a water content ranging from 50 wt% to 80 wt%. The CNF-DA/PAA@Fe $^{3+}$ hydrogel showed the maximum values of mechanical (0.57 MJ m $^{-3}$ for toughness, 34.8 MPa for compressive strength) and adhesive properties (1739 J m $^{-2}$ for interfacial toughness, 1017 kPa for shear strength, 1286 kPa for tensile strength).

V. Tunable and stable mechanical and adhesive performance by regulating CNF contents of the hydrogel

We fabricated a series of CNF-DA/PAA@Fe $^{3+}$ hydrogels with a fixed water content of 50 wt% and different CNF-DA contents to further improve the mechanical properties of the hydrogel. According to the CNF-DA content to the mass of the total hydrogel as x, the prepared hydrogel is denoted as CNF-DA/PAA@Fe $^{3+}$ _x, where x is 7.4 wt%, 10 wt%, 12 wt%, 15 wt%. For the case of CNF-DA/PAA@Fe $^{3+}$ hydrogels with varying CNF-DA contents, the mechanical properties of the hydrogels are correspondingly improved with the increase of CNF-DA contents.

VI. Tunable and stable mechanical and adhesive performance by regulating Fe contents of the hydrogel

To investigate the influence of Fe $^{3+}$ contents on the mechanical and adhesive performance of CNF-DA/PAA@Fe $^{3+}$ hydrogels, we prepared series of hydrogels with different Fe $^{3+}$ contents. According to the ratio of Fe $^{3+}$ to the mass of the CNF-DA as x, the prepared hydrogel is denoted as CNF-DA/PAA@Fe $^{3+}$ _x, where x is 0.5 wt%, 1 wt%, 2 wt%, and 4 wt%. We conducted compression and tensile tests to investigate the mechanical performance of the CNF-DA/PAA@Fe $^{3+}$ hydrogel with varying Fe contents. To explore the adhesive properties of these CNF-DA/PAA@Fe $^{3+}$ hydrogels, we performed 90-degree peeling, lap-shear, and tensile tests on the substrates of freshly excised porcine skin and engineering glass.

VII. Self-recovery adhesive performance of the CNF-DA/PAA@Fe $^{3+}$ hydrogel over the air oxidation time

To investigate the adhesive recovery behavior of the photodetached hydrogel on the freshly excised porcine skin in terms of the adhesive strength during the air-oxidation process, we conducted the 90-degree peel, lap-shear, and tensile tests of the hydrogels with different air oxidation time on the substrate of the freshly excised porcine skin.

VIII. Excellent and stable adhesive performance of the CNF-DA/PAA@Fe³⁺ hydrogel under different environment temperatures

To exclude the influence of temperature on the hydrogel, we evaluated the adhesion performance of the CNF-DA/PAA@Fe³⁺ hydrogel on the skin at different temperatures (20, 25, 30, and 35 °C). Furthermore, we investigated the adhesion strength of the CNF-DA/PAA@Fe³⁺ hydrogel on glass under varied UV intensity, and it can be observed that the adhesion strength displays a significant decreasing trend with the increase of UV intensity.

IX. Stable mechanical and adhesive performance of the hydrogel over the air oxidation time

To explore the mechanical and adhesive stability of the CNF-DA/PAA@Fe³⁺ hydrogel, we placed the hydrogel in the atmosphere (temperature is 25 °C and relative humidity is 70%) for 12, 24, 48, and 96 h, respectively, and then quantitatively tested mechanical and adhesive properties of these hydrogels. For the different oxidation times, we denote the hydrogel as CNF-DA/PAA@Fe³⁺_x, where x is 12, 24, 48, and 96 h. As a control, the initially prepared hydrogel is denoted as CNF-DA/PAA@Fe³⁺_0 h.

I. Computational simulation to evaluate the binding energies between Gelatin, Chitosan, Alginate, Starch, PAAm, PVA and Fe ions (Fe³⁺, Fe²⁺)

II. Detailed 90-degree peeling and tensile toughness tests

III. Detailed air-oxidation process investigation

IV. Excellent yet reversible mechanical and adhesive properties of CNF-DA/PAA@Fe³⁺ hydrogel prepared with different water content

(i) High yet reversible photodetachable adhesion of CNF-DA/PAA@Fe³⁺ hydrogel prepared with different water content

(ii) Strong yet tunable flexibility and compressibility of CNF-DA/PAA@Fe³⁺ hydrogel prepared with different water content

V. Excellent mechanical properties of CNF-DA/PAA@Fe³⁺ hydrogel prepared with different CNF ratio

(i) Robust and strong mechanical properties of CNF-DA/PAA@Fe³⁺ hydrogel prepared with different CNF ratio

VI. Excellent yet reversible mechanical and adhesive properties of CNF-DA/PAA@Fe³⁺ hydrogel prepared with different iron content

(i) Excellent yet reversible stretchability and photodetachable adhesion of CNF-DA/PAA@Fe³⁺ hydrogel prepared with different iron content

VII. Strong adhesive properties of CNF-DA/PAA@Fe³⁺ hydrogel with the extension of oxidation time on the freshly excised porcine skin

(i) Strong adhesive properties of CNF-DA/PAA@Fe³⁺ hydrogel with the extension of oxidation time on the freshly excised porcine skin

VIII. Adhesion strength as a function of several variables

(i) Excellent photodetachable adhesion of CNF-DA/PAA@Fe³⁺ hydrogel with different environment temperature after UV light

IX. Stable mechanical and adhesive properties of CNF-DA/PAA@Fe³⁺ hydrogel placed at different time

(i) Stable stretchability, compressability and adhesive properties of CNF-DA/PAA@Fe³⁺ hydrogel placed at different time

Figure R1 Summary of our experimental and simulation efforts during the past few months to provide more solid evidences towards wide, excellent yet tunable mechanical and adhesive properties and insights of the structure-property-function relationship of the developed dynamic hydrogel.

Updates to the revised manuscript: According to Referee #2's comments, we have added a more careful description to further explain the supramolecular network in the revised manuscript on Pages 2-4 and 17-18.

“Self-powered skin attachable and detachable electronics are under intense development to enable the internet of everything and everyone in new and useful ways. Existing on-demand separation strategies rely on complicated pretreatments and physical properties of the adherends, achieving detachable-on-demand in a facile, rapid, and universal way remains challenging. To overcome this challenge, we propose a rapid, reliable, and universal strategy to achieve both reversible tough adhesion and easy detachment by regulating its supramolecular network via the cellulose nanofiber (CNF)/Fe³⁺ (as a Fenton-like reagent)-mediated dopamine-poly(acrylic acid) dynamic hydrogel. This strategy enables the simple and rapid fabrication of strong yet reversible hydrogels with tunable toughness ((Value_{max}-Value_{min})/Value_{max} of up to 88%), on-demand adhesion energy ((Value_{max}-Value_{min})/Value_{max} of up to 98%), and stable conductivity up to 12 mS m⁻¹. We further extend this strategy to fabricate different CNF/Fe³⁺-based hydrogels from various biomacromolecules and petroleum polymers, and shed light on exploration of fundamental dynamic supramolecular network reconfiguration. Simultaneously, we prepare an adhesive-detachable triboelectric nanogenerator (TENG) as a human-machine interface for a self-powered wireless monitoring system based on this strategy, which can acquire the real-time, self-powered monitoring, and wireless whole-body movement signal, opening up possibilities for diversifying potential applications in e-skins and intelligent devices.” On Page 2.

“Existing skin-attachable electronics with autonomous powering ability are desired for obtaining accurate and reliable biological/physical information and can be reversibly attached to arbitrary surfaces and detached without leaving residues^{1, 2}. Thus the utilization of reversible adhesion hydrogels is of great significance for self-powered electronic skins^{3, 4}. In the past few years, great efforts have been devoted to realizing the reversible adhesion of hydrogel with a variety of hard and soft materials^{5, 6}. Reversible adhesion can be achieved using chemical connection consisting of reversible bonds^{7, 8}, including dynamic covalent bonds, noncovalent bonds and specific chemical groups, and physically topological entanglement through external stimuli, such as pH⁹, temperature¹⁰, current¹¹, and rays¹². These current hydrogel adhesion-detachment strategies suffer from general limitations in their universality. For reversible adhesion through reversible bonds, this method only works for the hydrogels that have been chemically designed or the adherends that can be surface modified. The procedures for hydrogel synthesis with tailored chemical structure and modification of adherends usually need rigorous reaction conditions, time-consuming pretreatments, or the usage of toxic agents. For adhesion through physical topological entanglement, it requires the adherends to have a porous microstructure. This method only works for porous adherend like hydrogels and living tissues. In this regard, the reversible integration of hydrogels and diverse materials calls for a fast, facile, and universal strategy.

Here we report a rapid, reliable, and universal strategy to achieve both reversible tough adhesion and easy detachment by regulating the supramolecular network of the dynamic hydrogel via the cellulose nanofiber (CNF)/Fe³⁺ (as a Fenton-like reagent)-mediated photo-Fenton (P.F.) reaction for self-powered e-skins (Fig. 1a and Supplementary Fig. 1). CNF derived from biomass cellulose—the most abundant polysaccharide on earth, featuring high mechanical strength, renewability, and biocompatibility, has been regarded as a multifunctional building block for applications in electronics and medicine. Furthermore, TEMPO-oxidized CNF contains position-selective and abundant carboxy groups on the surface, which offers abundant active sites to bond with Fe ions for coordination complexes. Strong coordination interactions between Fe³⁺ and carboxyl groups and the presence of catechol groups in the CNF-DA/PAA@Fe³⁺ hydrogel allow tough adhesion of the hydrogel. As directly exposed to UV light without external additives, through the transformation of the valence state of iron ions, the coordination interaction among Fe ions, CNFs, and PAA chains is dynamically dissociated and reconstructed, leading to the reconfiguration of the supramolecular network inside the hydrogel, and further making the cohesive strength of hydrogel self-regulatable (Fig. 1b and Supplementary Fig. 2). More attractively, by varying the exposure time in the air, we achieve the switchable operability of the above processes with associated reversible adhesion, representing great potentials for the reuse of the light-driven engineered hydrogel. Benefiting from the rich multi-type oxygen-containing functional groups of CNF chains and excellent mechanical properties, the hydrogel is able to stick quickly and conforms perfectly to the subject's skin surface as well as mild triggerable benign detachment with no visible residue and redness on the skin surface when exposed to the human-friendly UV radiation (Fig. 1c, Supplementary Fig. 3, and Supplementary Movie 1). With a switchable light-driven supramolecular network, the photo-detachable hydrogel exhibits excellent tunable adhesive strength change (before UV radiation: after UV radiation = 15.6 times) (Fig. 1d). Combined with the rapid, reliable, and reproducible regulating method, as well as adhesion-on-demand properties based on supramolecular network engineering, the developed convertible hydrogel holds great potential for practical application in photo-detachable self-powered e-skins” On Pages 3-4.

“To further verify the universality of the photo-detachable adhesion strategy based on the P.F. reaction, six additional hydrogels derived from biomass and synthesized polymers were prepared. First, Gaussian simulations were performed to investigate the feasibility of the dynamic structural dissociation and reconfiguration based on the interactions between iron ions and six coordination polymer frameworks including gelatin, chitosan, alginate, starch, polyacrylamide (PAAm), and polyvinyl alcohol (PVA). It can be observed that the binding energy between Fe³⁺ and gelatin (−127.64 kcal/mol), chitosan (−115.01 kcal/mol), alginate (−95.28 kcal/mol), starch (−88.51 kcal/mol), PAAm (−94.61 kcal/mol), and PVA (−72.14 kcal/mol) was much higher than that between Fe²⁺ and these polymers (Fig. 5a, b and Supplementary Fig. 40). Such a difference enables the desirable dissociation of coordination complexes involved in the P.F. reaction and thus triggerable detachment of these prepared hydrogels with the exposure to UV light.

Beyond the theoretical simulations, all these fabricated hydrogels were experimentally shown to be strong yet reversible flexibility and interfacial toughness. Upon UV radiation, the tensile toughness of the hydrogels decreased from 0.47 MJ m⁻³ to 0.057 MJ m⁻³ (for gelatin), 0.33 MJ m⁻³ to 0.050 MJ m⁻³ (for chitosan), 0.45 MJ m⁻³ to 0.044 MJ m⁻³ (for alginate), 0.40 MJ m⁻³ to 0.047 MJ m⁻³ (for starch), 0.42 MJ m⁻³ to 0.043 MJ m⁻³ (for PAAm), 0.31 MJ m⁻³ to 0.050 MJ m⁻³ (for PVA), respectively. As shown in Fig. 5c and Supplementary Fig. 41, the tensile toughness and interfacial toughness tunability ratios of all these hydrogels exceed 88%. These results demonstrated the facile yet versatile strategy for the fabrication of adhesive and photo-detachable dynamic hydrogels for reversible, strong and photo-detachable adhesion, which facilitates the reliable production of adhesive and photo-detachable hydrogels from a variety of resources, creating a wide-open space for diversifying potential applications, especially for self-powered electronic skins and intelligent devices.” On Pages 17-18.

Referee #3:

Comments:

*In this manuscript, the authors reported a light-driven supramolecular topology network engineering strategy for the construction of a dynamic hydrogel with adhesive and photo-detachable performance and evaluated its function and application potential as self-powered e-skins. This strategy used UV light to drive the supramolecular topology network transformation via the photo-Fenton-like reaction (P.F. reaction) of common copolymers with cellulose nanofibrils, which in turn triggers the conversion of Fe^{3+} and Fe^{2+} ions to regulate the properties of hydrogels. This strategy is interesting and practical, and the resulting CNF-DA/PAA@ Fe^{3+} hydrogel is superior to existing dynamically adhesive hydrogels in terms of adhesive properties while maintaining high tunability. The obtained results and discussions for designing and fabricating the CNF-DA/PAA@ Fe^{3+} hydrogel were provided through extensive state-of-the-art experimental characterizations and theoretical calculations. The manuscript is well-written. I would like to recommend the publication of this work in *Nature Communications* after some minor revisions to address the following points.*

Reply to the Referee: We thank Referee #3 for the positive comments on our work, especially pointing out its extraordinary performances and scientific interest, and the suggestion of publication of our work in *Nature Communications*.

1. In Fig. 2d, the diagram should be drawn in the form of $q-Iq^2$ to analyze the formation of dispersible microstructure in hydrogel with UV light.

Reply to the Referee: We genuinely thank Referee #3 for the valuable comments. We have revised the SAXS spectra in the form of $q-Iq^2$ to analyze the formation of the topological network in photo-detachable hydrogels with UV radiation and air oxidation. (See Revised Figure 2d)

Revised Figure 2d SAXS curves for the photo-detachable hydrogels before, after UV radiation, and after air oxidation.

2. In Fig. 2e, the peak-splitting curves in the XPS spectrum should be better represented.

Reply to the Referee: We thank Referee #3 for the kind reminders. We have revised the XPS spectrum in Revised Figure 2e in the revised manuscript to clearly represent the appearance of Fe^{3+} and Fe^{2+} peaks. The appearance of the peak at 710.6 eV of the hydrogel after UV radiation demonstrated the generation of Fe^{2+} in the system, verifying that Fe^{3+} was partially reduced to Fe^{2+} triggered by UV light, in which two free radicals ($\text{O}_2^{\cdot-}$ and $\text{HO}^{\cdot-}$) also participate in the P.F. reaction process. As the photo-detachable hydrogel was exposed to air, the Fe^{3+} was partially oxidized to Fe^{2+} , facilitating the self-recovery of the supramolecular network in the hydrogel.

Revised Figure 2e XPS spectra of Fe 2p regions of the CNF-DA/PAA@Fe³⁺ hydrogel before and after UV radiation and after air oxidation.

3. Both experimental and theoretical simulation show strong coordination interactions between Fe^{3+} and $-\text{COOH}$ groups. However, the Fe^{3+} -catechol coordination also play an important role in the topology and photo-detachable property of hydrogel. Please discuss the interactions of Fe^{3+} -catechol coordination and hydrogen bonding in the hydrogel in details.

Reply to the Referee: We genuinely thank Referee #3 for the valuable comments. In the CNF-DA/PAA@Fe³⁺ hydrogel system, Fe^{3+} ions and catechol groups of CNF-DA form Fe^{3+} -catechol coordination interactions, and the hydrogen bonding interaction is also generated between catechol groups. These impressive bonds contribute to the tight network architecture of the developed hydrogel, which is consistent with related previous works^{1,2}. Furthermore, under the stimulation of UV light, the resultant weak Fe^{2+} -catechol interaction due to the transformation of Fe^{3+} ions to Fe^{2+} ions leads to a loose topological network, which facilitates the easily peeled adhesion for photo-detachable performance.

Updates to the revised SI: According to Referee #3's comments, we have added a more careful description to discuss the interactions of Fe³⁺-catechol coordination and hydrogen bonding in the hydrogel in the revised Supporting Information on Pages 51-52.

“In the CNF-DA/PAA@Fe³⁺ hydrogel system, Fe³⁺ ions and catechol groups of CNF-DA form Fe³⁺-catechol coordination interactions, and the hydrogen bonding interaction is also generated between catechol groups. These impressive bonds facilitate a tight network architecture and are consistent with related works^{8,9}. Furthermore, under the stimulation of UV light, the weaker Fe²⁺-catechol interaction accompanied by the transformation of Fe³⁺ ions to Fe²⁺ ions leads to a loose topological network, which promotes the easily peeled adhesion behavior for photo-detachable performance.”

References:

1. Shannon, D. P. et al. Modular Synthesis and Patterning of High-Stiffness Networks by Postpolymerization Functionalization with Iron-Catechol Complexes. *Macromolecules* 56, 2268-2276 (2023).
2. Zhang, Z. et al. Eco-Friendly, Self-Healing Hydrogels for Adhesive and Elastic Strain Sensors, Circuit Repairing, and Flexible Electronic Devices. *Macromolecules* 52, 2531-2541 (2019).

4. The catechol groups could be oxidized gradually by O₂ and increase the crosslinking density of hydrogel. How about the stability of CNF-DA/PAA@Fe³⁺ hydrogel.

Reply to the Referee: We genuinely thank Referee #3 for the valuable comments. With the gradual oxidation of CNF-DA/PAA@Fe³⁺ hydrogel by O₂, the crosslinking density of the hydrogel increases, which leads to changes in hydrogel properties including mechanics and adhesion behavior. To further explore the stability of the CNF-DA/PAA@Fe³⁺ hydrogel, we placed the hydrogel in the atmosphere (temperature is 25 °C and relative humidity is 70%) for 12, 24, 48, and 96 h, respectively, and then quantitatively tested their mechanical and adhesive properties. For the difference in oxidation time, we denote the hydrogel as CNF-DA/PAA@Fe³⁺_x, where x is 12, 24, 48, and 96 h. As a control, the initially prepared hydrogel is denoted as CNF-DA/PAA@Fe³⁺_0 h.

As shown in Figure R15a, the initial CNF-DA/PAA@Fe³⁺_0 h hydrogel, even after being compressed to extreme deformation (up to 95% compression strain), shows no significant fractures and cracks, indicating its excellent mechanical properties. With a continuous increase of O₂ oxidation time, no obvious cracks can be observed in all hydrogels, which further proves the mechanical stability of the CNF-DA/PAA@Fe³⁺ hydrogel.

Furthermore, tensile and compressive stress-strain tests were conducted to analyze quantitatively the mechanical properties of the corresponding hydrogels with prolonged oxidation time. As shown in Figure R15b, as expected, the hydrogels exhibit increased tensile stress and decayed tensile strain with

extending oxidation time, which is due to the increased crosslinking density of the hydrogel by O₂ oxidation. In particular, with the oxidation time up to 96 h, the tensile strength and strain of the CNF-DA/PAA@Fe³⁺₉₆ h hydrogel were 0.088 MPa and 721%, respectively. Likewise, similar mechanical behavior is also obtained in compressive stress-strain tests (Figure R15c). Compared with the tensile Young's modulus of 0.0054 MPa and the compressive strength of 18.1 MPa of the CNF-DA/PAA@Fe³⁺₀ h hydrogel, the CNF-DA/PAA@Fe³⁺₉₆ h hydrogel correspondingly improves to 0.013 MPa and 26.2 MPa, increased by 2.4 times and 1.4 times, respectively (Figure R15d). These interesting mechanical observations jointly demonstrate the high strength and stiffness of the CNF-DA/PAA@Fe³⁺ hydrogel accompanied by oxidative growth.

Beyond the mechanical performance, the adhesion properties of these hydrogels with varying oxidation times were further investigated by 90-degree peeling tests on skin and glass (Figure R15e and R15f). The hydrogels with adhesion to skin and glass substrates showed similar peel force-displacement curves, that is, the maximum peel force of the hydrogels displayed a slightly decreasing trend with increasing oxidation time. Typically, compared to CNF-DA/PAA@Fe³⁺₀ h hydrogel, the interfacial toughness of CNF-DA/PAA@Fe³⁺₉₆ h hydrogel applied on skin and glass decreased by 1.3 times (from 88 J m⁻² to 66 J m⁻²) and 1.5 times (from 80 J m⁻² to 53 J m⁻²), respectively. Such considerable degradation is attributed to the catechol groups in the hydrogel being partially oxidated, leading to weak interfacial interactions between the hydrogel and substrates (Figure R15g).

In summary, with the presence of O₂, the catechol groups in the hydrogel were partially oxidized, leading to an increase in the crosslinking density. Although enhanced strength and stiffness of the hydrogel itself and a slightly decreased interfacial toughness, the hydrogel still maintains excellent toughness and adhesive strength, indicating its considerable stability and functionality.

Figure R15 a The digital images of the CNF-DA/PAA@Fe³⁺ hydrogels with different times before and after compression. **b** Tensile stress-strain curves of the CNF-DA/PAA@Fe³⁺ hydrogels with different times. **c** Compressive stress-strain curves of the CNF-DA/PAA@Fe³⁺ hydrogels prepared at different times. **d** Comparison of the toughness, tensile Young's modulus, and compressive strength of CNF-DA/PAA@Fe³⁺ hydrogels at different times. **e, f** 90-degree peel force-displacement curves of the CNF-DA/PAA@Fe³⁺ hydrogels with different times on the substrates of freshly excised porcine skin and engineering glass, respectively. **g** Comparison of the interfacial toughness of CNF-DA/PAA@Fe³⁺ hydrogels with different times on the substrates of freshly excised porcine skin and engineering glass, respectively. Values represent the mean and the standard deviation (n = 3).

Updates to the revised manuscript: According to Referee #3's comments, we have added a more careful description to clarify the uniqueness of this system, including the effect of O₂ on the stability of hydrogel in terms of mechanical and adhesive properties in the revised manuscript on Pages 10 and 14.

“In addition, the CNF-DA/PAA@Fe³⁺ hydrogel is able to be stretched by 10 times under external force without breaking. Moreover, the CNF-DA/PAA@Fe³⁺ hydrogel can still maintain its good flexibility even being exposed to air for a long time of 96 h. Those combined demonstrates its strong stretchability, durability, and stability (Fig. 3b, Supplementary Figs. 14 and 15).” On Page 10

“Attractively, the adhesive properties of the CNF-DA/PAA@Fe³⁺ hydrogel can be further enhanced substantially by decreasing the water content of the hydrogel. The resultant hydrogel with a water content of 50 wt% presented a high interfacial toughness of 1739 J m⁻², shear strength of 1017 kPa, and tensile strength of 1286 kPa on the skin; interfacial toughness of 1062 J m⁻², shear strength of 777 kPa, and tensile strength of 1557 kPa on the glass. It is worth noting that the decreasing of water content does not compromise the tunability of adhesive performance of the CNF-DA/PAA@Fe³⁺ hydrogel, as evidenced by its tunable adhesive performance ratio (Determined by $(Value_{max}-Value_{min})/Value_{max}$) of up to 98% in terms of the interfacial toughness, shear strength, and tensile strength (Supplementary Figs. 28–32).” On Page 14

5. How to exclude the influence of temperature on the performance of adjusted adhesion with UV light?

Reply to the Referee: We genuinely thank Referee #3 for the valuable comments. In order to exclude the influence of temperature on the hydrogel, we evaluated the adhesion performance of the CNF-DA/PAA@Fe³⁺ hydrogel at different temperatures (including 20, 25, 30, and 35 °C).

As shown in Figure R16a, as the environment temperature increases from 0 to 35 °C, the CNF-DA/PAA@Fe³⁺ hydrogel still stably adhere to the skin, showing its excellent and stable adhesion performance with varying temperature. Furthermore, we investigated the adhesion strength of the CNF-DA/PAA@Fe³⁺ hydrogel on glass under varied UV intensity, and it can be observed that the adhesion strength displays a significant decreasing trend with the increase of UV intensity (Figure R16b). In particular, at a UV intensity of 40 mW cm⁻², the adhesion strength of the hydrogel is 15 N m⁻¹, showing 82.5% tunability compared to 80 N m⁻¹ before UV radiation. It is further observed that the adhesive strength of the hydrogel to the glass exhibits negligible fluctuations with UV light at elevated temperatures, revealing that temperature has no significant effect on the adhesive properties of the hydrogel (Figure R16c). In summary, the effect of temperature on the performance of CNF-DA/PAA@Fe³⁺ hydrogel is almost negligible.

Figure R16 a The peeling process digital images of the CNF-DA/PAA@Fe³⁺ hydrogels with different environment temperature after UV light. **b, c** Relationship of adhesion strength of the CNF-DA/PAA@Fe³⁺ hydrogel to the UV intensity and environment temperature. The experiment for each value of the variable was repeated with three samples.

Updates to the revised manuscript: According to the Referee #3's comments, we have added a more careful description in the revised manuscript on Pages 14-15.

“Encouragingly, a wide range of adhesion strength of the CNF-DA/PAA@Fe³⁺ hydrogel can be achieved by tuning the UV intensity and radiation time, and the hydrogel still exhibits excellent adhesive strength as the temperature varies, which also imparts the desired adhesion of the developed hydrogel for multi-scene applications (Supplementary Figs. 36 and 37).”

6. How does photo-detachable cellulosic CNF-DA/PAA@Fe³⁺ hydrogel relate to its application in self-powered electronic skin?

Reply to the Referee: We genuinely thank Referee #3 for the valuable comments. To better illustrate the relationship between our photo-detachable adhesive property and practical applications, we summarize the hydrogel-based devices for real-world applications in e-skin and TENG into four parts, including (a) weak adhesive hydrogel for epidermal electronics; (b) tough adhesive hydrogel for use in real world; (c) reversible adhesive hydrogel for epidermal electronics; (d) reversible adhesive hydrogel with benign triggers for epidermal electronics.

(a) Weak adhesive hydrogel for epidermal electronics

Common hydrogels with weak adhesion require the participation of external activities such as bio-tapes or bandages when applied to human epidermal electronics, as reported in many previously reported works. For example, Mingxin Ye and co-workers reported a protein-based hydrogel and used it as an e-skin for sensing human activities; unfortunately, poor adhesion led to the need to tape the hydrogel fixed at the joint¹. Xinhua Liu and co-workers developed a collagen-based hydrogel to adhere to human skin for monitoring various movements, but the hydrogel is inevitably attached to the joint with adhesive tape². In addition, the application in TENG can also be observed by anchoring the hydrogel on the elbow with a bandage for energy harvesting studied in Shien-Ping Feng's work³. Commonly, on account of the lack of sufficient adhesion, these hydrogels have to compromise with tapes or bandages to complete the assembly of electronic devices, and there are more or less gaps in the interface between the hydrogel and the skin, as well as limited by external fixtures, which easily lead to increased signal distortion and reduced portability.

(b) Tough adhesive hydrogel for epidermal electronics

To break through the limitations of auxiliary anchors, in recent years, increased attention has been paid to tough adhesive hydrogel materials design. Particularly, introducing tailored functional groups, constructing topologically entangled networks, and designing bioinspired microstructures are innovative techniques used to prepare hydrogels with strong and secure adhesion when applied on skins^{4,5}. However, these common hydrogels with strong adhesion are usually difficult to remove from the skin after use or are poorly reusable, which is consistent with the work reported by Xuanhe Zhao and co-workers⁶. Similar observations are also found in Tiger H. Tao's work, that is, the commercial patch causes severe skin damage during/after peeling⁷.

(c) Reversible adhesive hydrogel for epidermal electronics

Therefore, to improve the applicability of hydrogels in the real world, hydrogels with reversible adhesion have emerged. Such hydrogels can achieve reversible and switchable adhesive properties under external environment or stimulation to meet the requirement of real-world applications⁸. For instance, as introduced by Qinghua Lu and co-workers, a reversibly adhesive gel with strong adhesion properties was obtained by controlling the melting-crystallization process of liquid crystals. The relatively low temperatures (below 40 °C) led to the crystallization with strong adhesion, while relatively high temperatures (above 40 °C) resulted in melting crystals with weak adhesion. Even if the reversible adhesion is induced by temperature, the gel-substrate detachment conditions are relatively harsh, consequently, the epidermal electronics may be needed to further study⁹. Wei Hong and co-workers developed a switchable adhesive hydrogel by controlling wrinkles, but this structural design is hard to achieve on the skin¹⁰.

(d) Reversible adhesive hydrogel with a benign trigger for epidermal electronics

Based on the above discussion, the development of a reversibly adhesive hydrogel with a benign trigger is essential and desirable for use in human skin as epidermal electronics. Our work reported a strategy to regulate hydrogel-skin on-demand adhesion separation due to the dynamic regulation of the topological network triggered by UV light-mediated Fe³⁺ ions valence state transition. In this detachment process, only soft UV light is used to trigger benign separation. Therefore, when the hydrogel is applied on the skin as an ionic conductor, it can not only realize stable contact with the skin to enhance the signal-to-noise ratio of the output signal, but also reduce the weight of the participants to facilitate its boost portability and conformability. More importantly, when the signal is collected, the device can be securely and quickly removed without evident damage to the skin. Combined with reversible adhesion properties and skin-friendly features, we believe that our developed hydrogel has great potential for real-world applications including but not limited to e-skin and TENG.

Updates to the revised manuscript: According to Referee #3's comments, we have added a more careful description to emphasize the relationship between the photo-detachable adhesive property and the practical application performance in the revised manuscript on Page 19.

“Taking advantage of the excellent mechanical properties, ionic conductivity, biocompatibility, and photo-tunable adhesion performance, the resulting hydrogel holds great promise for applications as epidermal electronics in the soft UV-light. Therefore, we exploit the supramolecular network reconfiguration as UV light-mediated valence transition of iron ions to regulate hydrogel-skin on-demand adhesion-detachment. Of note, human-friendly UV light ($\leq 40 \text{ mW cm}^{-2}$, 5 min) alone is required to trigger benign detachment without additional activity. When the hydrogel is integrated on the skin as an ionic conductor, it can not only realize stable contact with the skin, improve the signal-to-noise ratio of the output signal, but also reduce the weight of the participants, and boost portability and conformability. More importantly, when the signal is collected, the device is securely and quickly removed without damage to the skin. As a proof-of-concept demonstration, the obtained hydrogel serving as an ionic conductor in flexible sensors or triboelectric nanogenerators is applied in electronic skins to stably monitor human movements and physiological signals with a high signal-to-noise ratio (Supplementary Figs. 42–44, Supplementary Movies 3 and 4).”

References:

1. Chen, B. et al. Liquid metal-tailored gluten network for protein-based e-skin. *Nat. Commun.* **13**, 1206 (2022).
2. Bai, Z. et al. Mechanically robust and transparent organohydrogel-based e-skin nanoengineered from natural skin. *Adv. Funct. Mater.* **1**, 2212856 (2023).

3. Wu, Y. et al. Biomechanical energy harvesters based on ionic conductive organohydrogels via the Hofmeister effect and electrostatic interaction. *ACS Nano* **15**, 13427–13435 (2021).
4. Yang, J., Bai, R., Chen, B. & Suo, Z. Hydrogel adhesion: a supramolecular synergy of chemistry, topology, and mechanics. *Adv. Funct. Mater.* **30**, 1901693 (2019).
5. Ma, Z., Bao, G. & Li, J. Multifaceted design and emerging applications of tissue adhesives. *Adv. Mater.* **33**, e2007663 (2021).
6. Yuk, H., Zhang, T., Lin, S., Parada, G. A. & Zhao, X. Tough bonding of hydrogels to diverse non-porous surfaces. *Nat. Mater.* **15**, 190–196 (2016).
7. Zhang, Y. & Tao, T. H. Skin-friendly electronics for acquiring human physiological signatures. *Adv. Mater.* **31**, e1905767 (2019).
8. Liu, Z. & Yan, F. Switchable adhesion: on-demand bonding and debonding. *Adv. Sci.* **9**, e2200264 (2022).
9. Xi, S. et al. Reversible dendritic-crystal-reinforced polymer gel for bioinspired adaptable adhesive. *Adv. Mater.* **33**, e2103174 (2021).
10. Li, Q., Zhang, P., Yang, C., Duan, H. & Hong, W. Switchable adhesion between hydrogels by wrinkly. *Extreme. Mech. Lett.* **43**, 101193 (2021).

7. *What is the benefit of using cellulose nanofibrils? Please explain in more details.*

Reply to the Referee: We genuinely thank Referee #3 for the valuable comments. Here, we propose a design strategy that combines the interfacial chemical bonding and internal cellulose nanofibers (CNF)-mediated topological entanglement-dissociation, achieving both reversible tough adhesion and on-demand easy detachment by regulating the supramolecular topological network structure in the cellulose nanofiber reinforced dopamine-poly(acrylic acid)-iron metal (CNF-DA/PAA@Fe³⁺) hydrogel based on a photo-initiated Fe ions valence state transition in the photo-Fenton-like reaction (P.F. reaction).

(a) CNFs enable the photo-Fenton reaction for the reversible photo-detachable dynamic adhesion

CNFs are abundantly available from various biomass sources (e.g., wood, bamboo, straw), featuring an aligned, one-dimensional (1D) hierarchical structure rich in oxygen-containing polar functional groups (e.g., hydroxyl groups) in the form of repeating anhydroglucose units (AGUs) that make up the cellulose molecular chains. Such functional structure contains position-selective and abundant carboxy groups on the surface, which offers abundant active sites to bond with Fe ions for coordination complexes. Upon UV radiation without external additives, the environment of –COOH groups changes and –CHO groups are obtained in the CNF-DA/PAA@Fe³⁺ hydrogel, meanwhile, Fe³⁺ ions were transformed into Fe²⁺ ions in the photo-Fenton reaction. As such, the coordination interaction between CNF-DA, Fe ions, and PAA chains is dynamically transformed through the valence state switching of Fe ions, and the topological network structure inside the hydrogel undergoes the dissociation-reconfiguration, facilitating the triggerable benign detachment of the hydrogel;

Attractively, as the resultant hydrogel was exposed to air, the Fe^{2+} ions were oxidized to Fe^{3+} ions, leading to the reconfiguration of the supramolecular network of the CNF-DA/PAA@ Fe^{3+} hydrogel under air for strong yet reversible adhesion of the hydrogel.

In our work, aiming at the intrinsic flaws of low strength, poor toughness, and weak adhesion of the common PAA hydrogel, we directly addressed the mentioned multiple trade-offs by one-step introducing sustainable CNF in an in-situ polymerization manner for smart material design. CNFs not only serve as the supporting skeleton of the internal network of the hydrogel but also act as a reducing agent with abundant carboxyl groups that directly participate in the photo-Fenton reaction, contributing to the reversibility of coordination interactions and thus the dissociation-reconstruction of the topological network structure for reversible strong yet photo-detachable adhesion.

(b) CNFs-enabled hydrogels with widely tunable properties (i.e., self-healing properties, mechanical properties, reversible photo-detachable adhesion behavior, and self-recovery adhesive performance)

In our work, aiming at the intrinsic flaws of low strength, poor toughness, and weak adhesion of the common PAA hydrogel, we directly addressed the mentioned multiple trade-offs by one-step introducing sustainable CNF in an in-situ polymerization manner for smart material design. The strong coordination interaction between CNF-DA, Fe ions, and PAA chains, and CNFs serving as the supporting skeleton of the internal network structure enable the enhanced mechanical properties of the fabricated CNF-DA/PAA@ Fe^{3+} hydrogel. The CNF-DA/PAA@ Fe^{3+} hydrogel with strong stretchability, durability, and stability, taking advantage of the dynamic and reversible metal coordination interactions, can still restore the original intact shape after being torn apart, demonstrating its excellent self-healing performance. Additionally, the CNF-DA/PAA@ Fe^{3+} hydrogel exhibited the maximum tensile stress of 0.046 MPa at a fracture strain of 1850%, and the maximum tensile stress and fracture strain was 5 times higher than those of the PAA hydrogel. Similarly, the CNF-DA/PAA@ Fe^{3+} hydrogel displayed high compressive strength of 18.1 MPa at extreme compressibility up to 95% strain, 12 times higher than the poly(acrylic acid) (PAA) hydrogel (1.5 MPa). The excellent mechanical performance of the hydrogel guarantees its strong adhesion to the substrates.

Due to the presence of cellulose, under the exposure of UV light, the triggerable detachment of the hydrogel is realized by the dissociation of coordination complexes along with the partial reduction of Fe^{3+} to Fe^{2+} in the P.F. reaction, hence reducing the adhesion strength with tissue surfaces, which enables benign detachment of the hydrogel. Specifically, for the CNF-DA/PAA@ Fe^{3+} hydrogel, upon direct exposure to ultraviolet (UV) radiation, leading to dramatical tunable ratios of the interfacial adhesion energy (by up to 94%) and toughness (by up to 83%) and thus realizing reversibility of the flexibility and photo-detachment of the hydrogel with various substrates. Encouragingly, a wide range of adhesion strength of the CNF-DA/PAA@ Fe^{3+} hydrogel can be achieved (more than 1739 J m^{-2} for interfacial toughness, more than 99% for tunable adhesion ratio) by tuning the UV intensity and

radiation time or hydrogel water content, imparting the desired adhesion of the developed hydrogel for multi-scene applications.¹ More attractively, by varying the exposure time in the air, we achieve the switchable operability of the above processes with associated reversible adhesion, representing the great potential for the rapid and efficient reuse of the light-driven engineered hydrogel. As a proof-of-concept, the CNF-DA/PAA@Fe³⁺ hydrogel maintained stable adhesion and photo-detachment even more than 20 cycles of continuous UV radiation and air oxidation process, indicating outstanding stability and persistence of excellent and reversible adhesive properties strong and reversible adhesion performances of the developed hydrogel.

(c) CNFs-enabled hydrogels with the universality of photo-detachable adhesion strategy

This strategy applies to a wide variety of hydrogels and adherends, benefiting from introducing CNF chains with versatile functional groups. Especially, the high versatility of catechol in the CNF chains mainly arises from its unique structure with a benzene ring bearing two neighboring hydroxyl groups.² This structure interacts actively with many material systems: the inorganic solid composed of silicate can form hydrogen bonds with bidentate hydroxyl on catechol;³⁻⁵ organic solid containing alkane chains or benzene rings can interact effectively with the benzene ring on catechol through hydrophobic interaction or π - π stacking;^{6,7} metals can form metal-ion coordination bonds with the bidentate hydroxyl;^{8,9} and living tissues with active amine or sulfhydryl groups can form covalent bonds with oxidized catechol through Michael addition.¹⁰ Therefore, these impressive results demonstrate the strong adhesion and broad-range adhesion properties of the CNF-DA/PAA@Fe³⁺ hydrogel, thereby promising to improve the adaptability and universality of our fabricated hydrogel.

CNF/Fe³⁺ can be served as a Fenton-like reagent to realize the transformation between Fe³⁺-Fe²⁺ ions under UV light, and then enables a tunable supramolecular hydrogel. We prepared different hydrogels consisting of CNF/Fe³⁺ as a Fenton-like reagent, including those based on acrylamide (PAAm), polyvinyl alcohol (PVA), gelatin, chitosan, alginate, and starch. All fabricated hydrogels show good potential in strong yet reversible flexibility and interfacial toughness, and the tensile toughness and interfacial toughness tunability ratios of all these hydrogels exceed 88%.

Updates to the revised manuscript: According to Referee #3's comments, we have added a more careful description to further explain the topological network in the revised manuscript on Page 4.

“Benefiting from the rich multi-type oxygen-containing functional groups of CNF chains and excellent mechanical properties, the hydrogel is able to stick quickly and conforms perfectly to the subject's skin surface as well as mild triggerable benign detachment with no visible residue and redness on the skin surface when exposed to the human-friendly UV radiation (Fig. 1c, Supplementary Fig. 3, and Supplementary Movie 1). With a switchable light-driven supramolecular network, the photo-detachable hydrogel exhibits excellent tunable adhesive strength change (before UV radiation: after UV radiation = 15.6 times) (Fig. 1d). Combined

with the rapid, reliable, and reproducible regulating method, as well as adhesion-on-demand properties based on supramolecular network engineering, the developed convertible hydrogel holds great potential for practical application in photo-detachable self-powered e-skins.”

References:

1. Wang, S., et al. Strong, tough, ionic conductive, and freezing-tolerant all-natural hydrogel enabled by cellulose-bentonite coordination interactions. *Nat. Commun.* **13**, 3408 (2022).
2. Saiz-Poseu, J., Mancebo-Aracil, J., Nador, F., Busqué, F. & Ruiz-Molina, D. The chemistry behind catechol-based adhesion. *Angew. Chem. Int. Ed.* **58**, 696 (2019).
3. Mian, S. A., et al. A fundamental understanding of catechol and water adsorption on a hydrophilic silica surface: exploring the underwater adhesion mechanism of mussels on an atomic scale. *Langmuir* **30**, 6906 (2014).
4. Lin, Q., Gourdon, D., Sun, C. & Israelachvili, J. N. Adhesion mechanisms of the mussel foot proteins mfp-1 and mfp-3. *Angew. Chem. Int. Ed.* **58**, 696 (2019). *Proc. Natl. Acad. Sci. USA* **104**, 3782 (2007).
5. Mian, S. A., et al. Density functional theory study of catechol adhesion on silica surfaces. *J. Phys. Chem. C* **114**, 20793 (2010).
6. Yu, J., et al. Adaptive hydrophobic and hydrophilic interactions of mussel foot proteins with organic thin films. *Proc. Natl. Acad. Sci. USA* **110**, 15680 (2013).
7. Leng, C., et al. Interfacial structure of a DOPA-inspired adhesive polymer studied by sum frequency generation vibrational spectroscopy. *Langmuir* **29**, 6659 (2013).
8. Xu, Z. P., et al. Mechanics of metal-catecholate complexes: The roles of coordination state and metal types. *Sci. Rep.* **3**, 2914 (2013).
9. Pierpont, C. G. & Lange, C. W. The Chemistry of Transition Metal Complexes Containing Catechol and Semiquinone Ligands. *Proc. Inorg. Chem.* **41**, 331 (1994).
10. Lee, H., Scherer, N. F. & Messersmith, P. B. Single-molecule mechanics of mussel adhesion. *Proc. Natl. Acad. Sci. USA* **103**, 12999 (2006).

8. The presented photo-Fenton-like reaction induced reversible transformation of supramolecular topology network strategy is interesting. I am interested to know whether such strategy is applicable for other hydrogel or ionic gel materials as well? The authors are suggested to provide further discussion on this point.

Reply to the Referee: We genuinely thank Referee #3 for the valuable comments. In our system, the hydrogels with supramolecular network structures were designed based on the photo-Fenton-like reaction, in which UV-induced CNF to generate free hydroxyl radicals ($\cdot\text{OH}$), and Fe^{3+} ions were transformed into Fe^{2+} ions, thereby promoting network to transfer from tight to loose architectures. Noting that, in contrast to the traditional application of containing-iron Fenton-like systems, which

inevitably involves the utilization of hydrogen peroxide (H_2O_2) to produce free hydroxyl radicals ($\cdot\text{OH}$), the presented strategy utilizes only the inherent substrate for CNF and Fe^{3+} completed the whole reaction process¹⁻³. Therefore, we promise that CNF/ Fe^{3+} can be served as a Fenton-like reagent to realize the transformation between Fe^{3+} - Fe^{2+} ions under UV light, and then enables a tunable supramolecular hydrogel. To further validate this concept, we prepared different hydrogels consisting of CNF/ Fe^{3+} as a Fenton-like reagent, including those based on acrylamide (PAAm), polyvinyl alcohol (PVA), gelatin, chitosan, alginate, and starch. Furthermore, we investigated the significance of the mechanics and adhesion of these hydrogels on the glass before and after UV irradiation.

To reveal the mechanism of the universality of CNF/ Fe^{3+} -mediated photo-detachable adhesion strategy, we used Gaussian simulations to investigate the dynamically dissociated and restructured coordination interaction between iron ions and gelatin, chitosan, alginate, starch or PAAm, PVA. It can be observed that the binding energy between Fe^{3+} and gelatin (-127.64 kcal/mol), chitosan (-115.01 kcal/mol), alginate (-95.28 kcal/mol), starch (-88.51 kcal/mol), PAAm (-94.61 kcal/mol), and PVA (-72.14 kcal/mol) is much stronger than that between Fe^{2+} and gelatin (-81.94 kcal/mol), chitosan, alginate, starch or PAAm, PVA (Revised Figure 5a, b and Figure R17). This indicates that when the Fe^{3+} ions were transformed into Fe^{2+} ions with exposure to UV light, the coordination interaction between iron ions and polymer chains can be easily broken. Simultaneously, we experimentally found that various hydrogels prepared by this strategy, including but not limited to gelatin, chitosan, alginate, starch, PAAm, and PVA, all show good potential in strong yet reversible flexibility and interfacial toughness. Beyond the theoretical simulations, all these fabricated hydrogels were experimentally shown to be strong yet reversible flexibility and interfacial toughness. Upon UV radiation, the tensile toughness of the hydrogels decreased from 0.47 MJ m^{-3} to 0.057 MJ m^{-3} (for gelatin), 0.33 MJ m^{-3} to 0.050 MJ m^{-3} (for chitosan), 0.45 MJ m^{-3} to 0.044 MJ m^{-3} (for alginate), 0.40 MJ m^{-3} to 0.047 MJ m^{-3} (for starch), 0.42 MJ m^{-3} to 0.043 MJ m^{-3} (for PAAm), 0.31 MJ m^{-3} to 0.050 MJ m^{-3} (for PVA), respectively. As shown in Fig. 5c and Supplementary Fig. 41, the tensile toughness and interfacial toughness tunability ratios of all these hydrogels exceed 88%. These results demonstrated the facile yet versatile strategy for the fabrication of adhesive and photo-detachable dynamic hydrogels for reversible strong and photo-detachable adhesion, which facilitates the reliable production of photo-detachable hydrogels from a variety of resources, creating a wide-open space for diversifying potential applications, especially the self-powered electronic skins.

Revised Figure 5 Universality of the photo-detachable adhesion strategy. **a** Binding energy among gelatin, chitosan, alginate, starch, PAAm, PVA and Fe ions (Fe^{3+} , Fe^{2+}). The illustrations are snapshots of the gelatin, chitosan, alginate, starch, PAAm, PVA chain with Fe^{3+} , respectively. **b** Extending this photo-detachable adhesion strategy to different kinds of biomass materials and petroleum-based polymers, namely, gelatin, chitosan, alginate, starch, PAAm, and PVA. **c** Mechanical and adhesive impacts of the different kinds of biomass materials and petroleum-based polymers compared to non-UV light and UV radiation.

Figure R17 Structural formula of the Fe²⁺ and gelatin, chitosan, alginate, starch, PAAm, and PVA.

Figure R18 Universality of the photo-detachable adhesion strategy. **a** Digital photograph of the gelatin, chitosan, alginate, starch, PAAm, and PVA-based hydrogels. **b, c** Tensile curves of the gelatin, chitosan, alginate, starch, PAAm, and PVA-based hydrogels before and after UV radiation. **d, e** 90-

degree peeling curves of the gelatin, chitosan, alginate, starch, PAAm, and PVA-based hydrogels before and after UV radiation. **f** Schematic adhesion mechanism.

1. Ruppert G, Bauer R, Heisler G. The photo-Fenton reaction—an effective photochemical wastewater treatment process. *Journal of Photochemistry and Photobiology A: Chemistry* 73, 75-78 (1993).
2. Wang J, et al. In situ photo-Fenton-like tandem reaction for selective gluconic acid production from glucose photo-oxidation. *ACS Catalysis* 13, 2637-2646 (2023).
3. Zhang Y et al. Weakly hydrophobic nanoconfinement by graphene aerogels greatly enhances the reactivity and ambient stability of reactivity of MIL-101-Fe in Fenton-like reaction. *Nano Research*, 1-7 (2021).

Updates to the revised manuscript: According to Referee #3's comments, we have added a more careful description in the revised manuscript on Pages 17-18.

“To further verify the universality of the photo-detachable adhesion strategy based on the P.F. reaction, six additional hydrogels derived from biomass and synthesized polymers were prepared. First, Gaussian simulations were performed to investigate the feasibility of the dynamic structural dissociation and reconfiguration based on the interactions between iron ions and six coordination polymer frameworks including gelatin, chitosan, alginate, starch, polyacrylamide (PAAm), and polyvinyl alcohol (PVA). It can be observed that the binding energy between Fe^{3+} and gelatin (-127.64 kcal/mol), chitosan (-115.01 kcal/mol), alginate (-95.28 kcal/mol), starch (-88.51 kcal/mol), PAAm (-94.61 kcal/mol), and PVA (-72.14 kcal/mol) was much higher than that between Fe^{2+} and these polymers (Fig. 5a, b and Supplementary Fig. 40). Such a difference enables the desirable dissociation of coordination complexes involved in the P.F. reaction and thus triggerable detachment of these prepared hydrogels with the exposure to UV light.”

“Beyond the theoretical simulations, all these fabricated hydrogels were experimentally shown to be strong yet reversible flexibility and interfacial toughness. Upon UV radiation, the tensile toughness of the hydrogels decreased from 0.47 $MJ\ m^{-3}$ to 0.057 $MJ\ m^{-3}$ (for gelatin), 0.33 $MJ\ m^{-3}$ to 0.050 $MJ\ m^{-3}$ (for chitosan), 0.45 $MJ\ m^{-3}$ to 0.044 $MJ\ m^{-3}$ (for alginate), 0.40 $MJ\ m^{-3}$ to 0.047 $MJ\ m^{-3}$ (for starch), 0.42 $MJ\ m^{-3}$ to 0.043 $MJ\ m^{-3}$ (for PAAm), 0.31 $MJ\ m^{-3}$ to 0.050 $MJ\ m^{-3}$ (for PVA), respectively. As shown in Fig. 5c and Supplementary Fig. 41, the tensile toughness and interfacial toughness tunability ratios of all these hydrogels exceed 88%. These results demonstrated the facile yet versatile strategy for the fabrication of adhesive and photo-detachable dynamic hydrogels for reversible, strong and photo-detachable adhesion, which facilitates the reliable production of adhesive and photo-detachable hydrogels from a variety of resources, creating a wide-open space for diversifying potential applications, especially for self-powered electronic skins and intelligent devices.”

REVIEWER COMMENTS

Reviewer #1 (Remarks to the Author):

The authors tried to convince reviewers with plenty of discussion. However, the very long discussion makes reviewer harder to focus on the key point. It is recommended to briefly list the critical points when responding to the comments.

1. The authors should get insight into the intrinsic mechanism between the current work and Suo group's work on the photo-detachable hydrogel (Adv. Mater. 2019, 31, 1806948). The both works took UV light to enable the transformation of Fe³⁺ to Fe²⁺ for the regulation of the adhesion energies. This point is exactly same for two works. The different point is the current work added DA into the polymer network. The DA is capable of providing initial adhesion to the tested surface. The transformation of Fe³⁺ to Fe²⁺ adjusted the mechanical properties of hydrogel, so that regulating the interfacial adhesion, because the adhesion is highly dependent on the mechanical strength of hydrogel. Therefore, the conceptual novelty on the intrinsic mechanism should be enhanced. What's the new concept that readers can learn from?

2. It is interesting on the results of Fig. 5c. After UV light, the interfacial toughness is almost the same for all the samples. This result implies that the hydrogel itself is not important. The only important factor is UV light. The authors may need to reconsider the real mechanism on the interfacial adhesion. Did UV light decrease the DA-induced interfacial adhesion interaction?

3. The authors figured out the low interfacial adhesion by decreasing the water content of hydrogel. In this case, what's the critical role of Fe³⁺ ions on the interfacial adhesion?

4. The hydrogel would be dehydrated and all properties would change upon long-term exposure to air. How to maintain its performance when it was used as e-skin?

Reviewer #2 (Remarks to the Author):

The manuscript can be accepted in the current form.

Reviewer #3 (Remarks to the Author):

The authors made very impressive revisions to address the comments carefully. The clarification of uniqueness of the developed strategy and the insightful discussions of the structure-property relationship were well-presented. I would like to recommend the acceptance of this interesting and inspiring work for publication in Nature Communications.

Itemized list of response to the Referees' remarks
(Black: Referees' remarks; Blue type: Our response)

Referee #1 (Remarks to the author):

The authors tried to convince reviewers with plenty of discussion. However, the very long discussion makes reviewer harder to focus on the key point. It is recommended to briefly list the critical points when responding to the comments.

Reply to the Referee: We thank Referee #1 for the constructive comments. We have briefly listed the critical points in response to the comments and revised the manuscript.

1. The authors should get insight into the intrinsic mechanism between the current work and Suo group's work on the photo-detachable hydrogel (Adv. Mater. 2019, 31, 1806948). The both works took UV light to enable the transformation of Fe^{3+} to Fe^{2+} for the regulation of the adhesion energies. This point is exactly same for two works. The different point is the current work added DA into the polymer network. The DA is capable of providing initial adhesion to the tested surface. The transformation of Fe^{3+} to Fe^{2+} adjusted the mechanical properties of hydrogel, so that regulating the interfacial adhesion, because the adhesion is highly dependent on the mechanical strength of hydrogel. Therefore, the conceptual novelty on the intrinsic mechanism should be enhanced. What's the new concept that readers can learn from?

Reply to the Referee: We agree with Referee #1 on the critical role of DA in providing initial adhesion to the adherends and the regulation of adhesion behavior of the hydrogel by the transformation of Fe^{3+} to Fe^{2+} ions. However, the intrinsic mechanism of our work is conceptionally different from Suo's previous work, as briefly quoted below.

Conceptual novelty on the intrinsic mechanism: CNF-mediated manifold dynamic synergy for the reversible photo-detachable adhesion of the hydrogel

In our work, we successfully **constructed the hydrogel with both reversible tough adhesion and easy photodetachment via ingenious CNF-mediated manifold dynamic synergy**. In this manifold dynamic synergy, the CNFs reinforced-network structure and the physical crosslinking of the polymer network, as well as the strong coordination between Fe^{3+} ions and polymer networks in the CNF-DA/PAA@ Fe^{3+} hydrogel endow its excellent comprehensive mechanical properties and cohesive strength. Meanwhile, the chemical modification of CNFs by DA further improves the interfacial combination of the fabricated hydrogel with diverse adherends. Such CNF-mediated manifold dynamic synergy strategy for strong yet reversible adhesion is conceptually different from that in Suo's group (Ref.1), where the prerequisite spreading of a solution of additional polymer chains to the adherends surface and one-way stitching of adherends by coordination complexation of diffused

Fe³⁺ ions and carboxyl groups on the polymer chains. Thus, the solution-adhesion strategy for strong adhesion in Suo's work (Ref.1) was strongly affected by the rheology of added polymer gelation. On the other hand, upon the UV radiation, in our work, the transformation of Fe³⁺ to Fe²⁺ ions via photo-Fenton-like reaction leads to the dissociation of network structures, thereby regulating the interfacial adhesion behavior of the hydrogel. The photodetach-trigger (i.e., Fe³⁺/Fe²⁺) in our work is the same as Suo's group, however, their mechanisms are conceptually different. In our work, the transformation of Fe ions was realized based on a photo-Fenton-like reaction, where CNFs in the hydrogel material served as a green Fenton-like reducing agent for photodetachment under mild operating conditions. The CNF-mediated photo-Fenton-like reaction and the CNF backbone structure also afforded the reversibility of the coordination interactions in the hydrogel, further achieving the continuous utility of the hydrogel for multicycle adhesion and detachment. In contrast, in Suo's work (Ref.1), citric acid with a controlled pH value within 3 was used to help the one-way transformation of Fe ions without any reversibility and recyclability.

Additionally, benefiting from this CNF-mediated manifold dynamic synergy strategy, we achieve a strong and immediate adhesion on various substrates ranging from aqueous and nonaqueous systems, and rapid and human-friendly detachment of the hydrogel. Moreover, the mechanical and adhesion energy of our developed hydrogel is tunable for diversifying applications, and the detached hydrogel can be easily self-recovered by air oxidation of the hydrogel for reuse. Furthermore, the biocompatible and human-friendly hydrogel as an ionic conductor in flexible sensors or triboelectric nanogenerators can be applied in electronic skins to stably monitor human movements and physiological signals with a high signal-to-noise ratio, creating a wide-open space for potential applications, especially for personal health monitoring and recreational human motion tracking. These features imparted by our new CNF-mediated manifold dynamic synergy strategy further distinguish our dynamic hydrogel materials from those developed by Suo's team (a solution-based adhesive, Ref.1), which has a controlled low pH value of ~3 and requires a time-consuming permeation process and accessibility of the diffusion substrate, making it relatively difficult to apply this photo-detachable system to human skin.

Major differences between our work and Suo group's previous work (Adv. Mater. 2019, 31, 1806948)

To further illustrate the originality of our current work, we summarize the major differences between our work and the Suo group's previous work (Adv. Mater. 2019, 31, 1806948), as shown below (Table R1).

1. **Photodetachment mechanism: CNFs-mediated manifold dynamic synergy mechanism** (Our work) VS interfacial iron-carboxyl chemistry regulatory mechanism (Ref.1 Adv. Mater. 2019, 31, 1806948).
2. **Fabrication method of adhesives: hydrogels prepared by *in-situ* copolymerization of CNF-DA, Fe³⁺ ions, and acrylic acid** (Our work) VS the spreading of an aqueous solution of polymer chains and Fe³⁺/citric acid with a controlled pH value of 1.5~3 (Ref.1 Adv. Mater. 2019, 31,

1806948).

3. **Interfacial adhesion source: CNFs-enhanced network and DA adhesive groups** (Our work) VS a third network in topological entanglement with the two preexisting networks of the adherends and coordination with Fe³⁺ ions (Ref.1 Adv. Mater. 2019, 31, 1806948).
4. **Micro-structure: reversible transformation from a dense to loose micro-structure** (Our work) VS an aqueous solution without reversibility (Ref.1 Adv. Mater. 2019, 31, 1806948).
5. **Properties: reversible mechanical properties, tunable adhesive behavior, self-healing ability, ionic conductivity, and biocompatibility** (Our work) VS photodetachable adhesive properties (Ref.1 Adv. Mater. 2019, 31, 1806948).
6. **Applications: e-skins, soft robots, energy storage, and intelligent devices** (Our work) VS no demonstration in the paper (Ref.1 Adv. Mater. 2019, 31, 1806948).

Table R1. Comprehensive comparisons (differences) of our work with the Suo group’s previously reported paper (Ref.1 Adv. Mater. 2019, 31, 1806948).

Adhesive materials (Study object)	Fabrication method	Interfacial adhesion source	Photodetachment mechanism	Micro-structure	Properties	Applications	Refs
CNF-mediated photo-detachable adhesive hydrogel	Hydrogels fabricated by in-situ copolymerization of CNF-DA, iron ions, and acrylic acid	CNFs-reinforced network and DA (dopamine) adhesive groups	CNFs-mediated manifold dynamic synergy mechanism	Reversible transformation from a dense to loose micro-structure	1. Reversible mechanical properties; 2. Tunable adhesive properties; 3. Self-healing properties; 4. Ionic conductivity 5. Biocompatibility	1. E-skins, soft robots, energy storage, and intelligent devices; 2. Wound dressing and transdermal drug delivery	This work
PAA aqueous solution, and Fe ³⁺ -citric acid aqueous solution	FeCl ₃ and citric acid were dissolved in deionized water.	Fe ³⁺ ions cross-linked PAA chains into two adherends	Interfacial iron-carboxyl chemistry	Solution	Photodetachable adhesive properties	No demonstration	Ref. 1

In brief, our CNF-mediated manifold dynamic synergy strategy approach to regulate the adhesion properties of CNF-mediated hydrogels is conceptionally different from previously reported strategies, which is faster, safer, more reversible, reliable, and designable.

Updates to the revised manuscript: According to Referee #1’s comments, we have added a more careful description to further explain the CNF-mediated manifold dynamic synergy mechanism in the revised manuscript on Pages 2-4 and Page 23.

“To overcome this challenge, an ingenious cellulose nanofiber (CNF)-mediated manifold dynamic synergy strategy is developed to construct a supramolecular hydrogel with both reversible tough adhesion and easy photodetachment. The CNF-reinforced network and the coordination between Fe ions and polymer chains endow the dynamic reconfiguration of supramolecular networks and the adhesion behavior of the hydrogel.” on Page 2.

“Reversible adhesion can be achieved using chemical connection consisting of reversible bonds^{7, 8}, including dynamic covalent bonds, noncovalent bonds and specific chemical groups, and physically topological entanglement through external stimuli, such as pH⁹, temperature¹⁰, current¹¹, and rays¹². Suo’s group⁷ developed a smart solution adhesive, where the spreading of an aqueous solution of polymer chains on the surface of two adherends to crosslink them together for topological adhesion. To further shorten the interfacial adhesion time and afford the effective interfacial linking between the bulk hydrogel and adherends to transmit force and elicit energy dissipation, there is a need to design a dynamic adhesion strategy with quick and strong long-term adhesion while leaving a window period for reversible easy detachment.” on Page 3.

“Herein, we successfully constructed a supramolecular hydrogel with both reversible tough adhesion and easy photodetachment via an ingenious cellulose nanofiber (CNF)-mediated manifold dynamic synergy strategy (Fig. 1a and Supplementary Fig. 1). Cellulose—the most abundant polysaccharide on earth, featuring high mechanical strength, renewability, and biocompatibility, has been regarded as a multifunctional building block for developing high-performance materials. Additionally, TEMPO-oxidized CNF and poly(acrylic acid) (PAA) chains with position-selective and abundant carboxy groups on the surface enable their coordination complexation with Fe³⁺ ions. The CNF-reinforced supramolecular network, strong coordination interactions between Fe³⁺ and polymer chains, as well as the dopamine (DA) adhesive groups collaboratively endow the excellent comprehensive mechanical properties and cohesive strength of the fabricated CNF-DA/PAA@Fe³⁺ hydrogel. As directly exposed to UV light, Fe³⁺ ions are reduced to Fe²⁺ ions based on the photo-Fenton-like reaction (P.F. reaction) in the hydrogel system, where the CNF serves as a green Fenton-like reagent, leading to dissociation and reconfiguration of the Fe ions and CNF networks, thereby making the self-regulatable adhesive behavior of the hydrogel for on-demand photodetachment (Fig. 1b and Supplementary Fig. 2).” on Pages 3-4.

“In summary, we constructed a supramolecular CNF-DA/PAA@Fe³⁺ hydrogel with both reversible tough adhesion and easy photodetachment via a CNF-mediated manifold dynamic synergy strategy. The CNF-reinforced network, the physical entanglement between CNF and PAA chains, as well as the strong coordination between Fe³⁺ ions and CNF networks endow the excellent reversible mechanical and adhesive properties of the hydrogels.” on Page 23.

References

1. Gao, Y., Wu, K. & Suo, Z. *Adv. Mater.* **31**, 1806948 (2019).

2. It is interesting on the results of Fig. 5c. After UV light, the interfacial toughness is almost the same for all the samples. This result implies that the hydrogel itself is not important. The only important factor is UV light. The authors may need to reconsider the real mechanism on the interfacial adhesion. Did UV light decrease the DA-induced interfacial adhesion interaction?

Reply to the Referee: We thank Referee #1 for the thoughtful comments.

Regarding the comment “After UV light, the interfacial toughness is almost the same for all the samples. This result implies that the hydrogel itself is not important. The only important factor is UV light.”:

In our universal design of photodetachable hydrogel, UV light serves as a trigger to stimulate the transformation of Fe^{3+} to Fe^{2+} ions based on CNFs-mediated photo-Fenton-like reaction, leading to the dissociation and reconstruction of the supramolecular networks and the regulatory of the cohesive strength of the hydrogel. As shown in Figure R1, all universal hydrogels delivered a decreased tensile strength. Owing that the adhesion is highly dependent on the mechanical strength of hydrogel, the cohesive strength of all universal hydrogels was reduced after UV radiation, thereby affecting the interfacial bonding between the hydrogel and other adherends. Therefore, the hydrogel itself with the photo-responsive CNF- Fe^{3+} -DA supramolecular network design is important, while UV light just serves as a trigger to regulate the hydrogel’s supramolecular network and properties.

Figure R1 a, b Tensile curves of the gelatin, chitosan, alginate, starch, PAAm, and PVA-based hydrogels with Fe ions before and after UV radiation. **c** Comparison of the toughness of the gelatin, chitosan, alginate, starch, PAAm, and PVA-based hydrogels with Fe ions before and after UV radiation. **d, e** 90-peeling curves of the gelatin, chitosan, alginate, starch, PAAm, and PVA-based hydrogels with Fe ions before and after UV radiation on the substrate of glass. **f** Comparison of the interfacial toughness of the gelatin, chitosan, alginate, starch, PAAm, and PVA-based hydrogels with Fe ions before and after UV radiation.

Regarding the comment “The authors may need to reconsider the real mechanism on the interfacial adhesion. Did UV light decrease the DA-induced interfacial adhesion interaction?”:

To investigate the influence of UV light on the DA-induced interfacial adhesion interaction, we synthesized universal hydrogels without Fe^{3+} ions and compared their mechanical and interfacial adhesion properties with those with Fe^{3+} ions. As shown in Figure R2, various hydrogels without Fe^{3+} ions showed irreversible tensile toughness and interfacial peeling toughness before and after UV radiation, indicating that the UV light does not decrease the DA-induced interfacial adhesion of the hydrogels. This result further confirms our conclusion that UV light just serves as a trigger to regulate the hydrogel's photo-responsive CNF- Fe^{3+} -DA supramolecular network and properties, where the Fe^{3+} ions play a crucial role by forming strong yet photo-responsive CNF- Fe^{3+} coordination interactions and DA play a synergic role by providing initial interfacial adhesion.

It is worth noting that, by comparing the tensile strength and toughness of the hydrogels themselves (Figure R2: without Fe^{3+} ions VS Figure R1: with Fe^{3+} ions), we found that the formation of strong coordination between the Fe^{3+} ions and CNF networks also contributes to the formation of a more robust supramolecular hydrogel network, further highlighting the benefits of our CNF-mediated manifold synergic design.

Figure R2 a, b Tensile curves of the gelatin, chitosan, alginate, starch, PAAm, and PVA-based hydrogels without Fe^{3+} ions before and after UV radiation. **c** Comparison of the toughness of the gelatin, chitosan, alginate, starch, PAAm, and PVA-based hydrogels without Fe ions before and after UV radiation. **d, e** 90-peeling curves of the gelatin, chitosan, alginate, starch, PAAm, and PVA-based hydrogels without Fe ions before and after UV radiation on the substrate of glass. **f** Comparison of the interfacial toughness of the gelatin, chitosan, alginate, starch, PAAm, and PVA-based hydrogels without Fe ions before and after UV radiation.

Updates to the revised manuscript: According to Referee #1's comments, we have added a more careful description to further explain the intrinsic adhesion mechanism of the universality in the revised manuscript on Page 17.

“To further verify the universality of the photo-detachable adhesion strategy based on the P.F. reaction, six additional hydrogels derived from biomass and synthesized polymers were prepared. First, Gaussian simulations were performed to investigate the feasibility of the dynamic structural dissociation and reconfiguration based on the interactions between iron ions and six coordination polymer frameworks including gelatin, chitosan, alginate, starch, polyacrylamide (PAAm), and polyvinyl alcohol (PVA). It can be observed that the binding energy between Fe^{3+} and gelatin (-127.64 kcal/mol), chitosan (-115.01 kcal/mol), alginate (-95.28 kcal/mol), starch (-88.51 kcal/mol), PAAm (-94.61 kcal/mol), and PVA (-72.14 kcal/mol) was much higher than that between Fe^{2+} and these polymers (Fig. 5a, b and Supplementary Fig. 42). Such a difference enables the coordination dissociation of supramolecular networks via CNF-mediated manifold dynamic synergy, which decreases the cohesive strength of the hydrogels, thereby allowing triggerable detachment of these prepared hydrogels with the exposure to UV light.”

3. The authors figured out the low interfacial adhesion by decreasing the water content of hydrogel. In this case, what's the critical role of Fe^{3+} ions on the interfacial adhesion?

Reply to the Referee: We thank Referee #1 for the thoughtful suggestions.

Regarding the comment “The authors figured out the low interfacial adhesion by decreasing the water content of hydrogel.”:

We would like to correct Referee #1 that the interfacial toughness of CNF-DA/PAA@ Fe^{3+} hydrogel gradually increases by decreasing the water content of the hydrogels. The decrease in the water content of a hydrogel leads to the shrinkage of the polymer network, which may increase the surface chain density and result in the increase of the adhesion energy. Besides, the shrinking polymer network will also affect the bulk properties of the hydrogel, including the bulk energy dissipation and elastic modulus, thereby indirectly affecting the adhesion energy of a hydrogel¹⁻³.

Regarding the comment “In this case, what's the critical role of Fe^{3+} ions on the interfacial adhesion?”:

The strong coordination with Fe^{3+} ions and CNF networks endows the excellent comprehensive mechanical properties and cohesive strength of hydrogels. In our adhesion tests on hydrogels with different water contents, the decrease of the water content of a hydrogel leads to the shrinkage of the

polymer network, which may increase the coordination cross-links density and result in the increase of the cohesive strength, further affecting the adhesive energy. Besides, upon the UV radiation, the Fe^{3+} ions are reduced to Fe^{2+} ions in the photo-Fenton-like reaction (the CNF fibers serve as a Fenton-like reagent), leading to dissociation and reconstruction of the Fe ions and CNF networks, further the regulatory of the cohesive strength of the hydrogels. Owing that the adhesion is highly dependent on the mechanical strength of hydrogel, during the UV light process, the valence state transition of Fe ions causes the dissociation of the internal network structure of the hydrogel and reduces the cohesive strength of the hydrogel, which in turn affects the interfacial bonding between the hydrogel and other materials.

To investigate the influence of Fe^{3+} ions on the interfacial adhesion, we synthesized a series of Fe^{3+} ions-free CNF-DA/PAA hydrogels with different water content and compared their mechanical properties with those with Fe^{3+} ions. As shown in Figure R3, various hydrogels prepared without Fe ions exhibit weaker, irreversible tensile toughness and interfacial peeling toughness before and after UV irradiation. By contrast, all iron-containing CNF-DA/PAA@ Fe^{3+} hydrogels with different water content exhibited photo-triggered mechanical and adhesion properties before and after UV irradiation (Figure R4). These results clearly demonstrate the critical role of Fe^{3+} ions in strengthening the hydrogel network and providing reversible tuning of the adhesive properties via forming strong yet photo-responsive coordination Fe^{3+} -CNFs-DA supramolecular network (Figure R3: without Fe^{3+} VS Figure R4: with Fe^{3+} ions).

Figure R3 a, b Tensile curves of the CNF-DA/PAA hydrogels (without Fe ions) with different water content (50%, 60%, 70%, 80%) before and after UV radiation. c Comparison of the toughness of CNF-DA/PAA hydrogels with different water content (50%, 60%, 70%, 80%) before and after UV radiation. d, e 90-peeling curves of the CNF-DA/PAA hydrogels with different water content (50%, 60%, 70%, 80%) before and after UV radiation on the substrate of glass. f Comparison of the

interfacial toughness of the CNF-DA/PAA hydrogels with different water content (50%, 60%, 70%, 80%) before and after UV radiation.

Figure R4 a, b Tensile curves of the CNF-DA/PAA@Fe³⁺ hydrogels (with Fe ions) with different water content (50%, 60%, 70%, 80%) before and after UV radiation. **c, d** Comparison of the toughness of CNF-DA/PAA@Fe³⁺ hydrogels with different water content (50%, 60%, 70%, 80%) before and after UV radiation. **e, f** 90-peeling curves of the CNF-DA/PAA@Fe³⁺ hydrogels with different water content (50%, 60%, 70%, 80%) before and after UV radiation on the substrate of glass. **g, h** Comparison of the interfacial toughness of the CNF-DA/PAA@Fe³⁺ hydrogels with different water content (50%, 60%, 70%, 80%) before and after UV radiation.

References:

1. Zhou, Z., Lei, J. & Liu, Z. Effect of water content on physical adhesion of polyacrylamide hydrogels. *Polymer* **246**, 124730 (2022).
2. Skelton, S., et al. Biomimetic adhesive containing nanocomposite hydrogel with enhanced materials properties. *Soft Matter* **9**, 3825 (2013).
3. Zhang, X., et al. A Tough and stiff hydrogel with tunable water content and mechanical properties based on the synergistic effect of hydrogen bonding and hydrophobic interaction. *Macromolecules* **51**, 8136–8146 (2018).

Updates to the Revised Manuscript: According to Referee #1's comments, we have added a more careful description to further explain the strong coordination between Fe ions and CNFs networks in the revised manuscript on Page 11.

“In addition, the CNF-DA/PAA@Fe³⁺ hydrogel also shows excellent and tunable compression strength and toughness ((Value_{max}-Value_{min})/Value_{max} of up to 88% for toughness) by adjusting the water or CNF contents of the hydrogel (Supplementary Figs. 20–22). These results demonstrated the mechanical properties tunability of the CNF-DA/PAA@Fe³⁺ hydrogel enabled by the transformation of Fe ions during the UV-stimulus process, which is of great significance to the self-powered e-skins with satisfactory mechanical performance.”

Updates to the Revised SI: According to Referee #1's comments, we have added a more careful description to further explain the strong coordination between Fe ions and CNFs networks in the revised manuscript on Pages 66 and 86.

“We carried out the tensile tests of a series of CNF-DA/PAA hydrogels with different water contents to explain the role of Fe ions in enhancing the cross-linking of the polymer network. As shown in Supplementary Figure 21, the CNF-DA/PAA hydrogels with different water contents show weaker tensile toughness (0.042 MJ m⁻³ for 50%, 0.045 MJ m⁻³ for 60%, 0.049 MJ m⁻³ for 70%, 0.057 MJ m⁻³ for 80%) in comparison to those CNF-DA/PAA@Fe³⁺ hydrogels with different water contents, and the toughness of the CNF-DA/PAA hydrogels basically maintained a stable state during the UV radiation process, where the Fe ions play a key role via forming strong yet photo-responsive coordination Fe³⁺-CNFs-DA supramolecular network.” on Page 66

“We explored the interfacial toughness of a series of CNF-DA/PAA hydrogels without Fe ions with different water contents on the glass before and after UV radiation. The interfacial toughness of all CNF-DA/PAA hydrogels basically maintains a stable state (919.04 J m⁻² for 50%, 570.20 J m⁻² for 60%, 277.60 J m⁻² for 70%, 107.53 J m⁻² for 80%) during the UV radiation process. These results demonstrated that Fe ions as part of the internal network of the hydrogel participated in the formation of the network structure, and cooperated with CNF fibers and PAA chains to enhance the cross-linking density of the hydrogel. Simultaneously, upon the UV radiation, the transformation of Fe³⁺ ions to Fe²⁺ ions in the photo-Fenton-like reaction (the CNF fibers serves as a Fenton-like reagent) led to the dissociation and reconstruction of Fe ions and CNF networks, achieving the regulatory of the cohesive strength of the hydrogels, thereby affecting the interfacial bonding interactions.” on Page 86

4. The hydrogel would be dehydrated and all properties would change upon long-term exposure to air. How to maintain its performance when it was used as e-skin?

Reply to the Referee: We thank Referee #1 for the constructive comments.

Dehydration is a common challenge for all the hydrogel-e-skins¹⁻⁹:

Hydrogels are a group of crosslinked polymers that reveal a growing number of applications in emerging fields with credit to their intrinsic stretchability, good biocompatibility, and mechanical properties. However, traditional hydrogels always lose their flexibility and functions in dry environments with long-term operation because the internal water inevitably undergoes evaporation. Thus, enhancing the water retention capacity of hydrogel becomes a critical challenge in the fields. So far, a few strategies focused on anti-dehydration have been reported (Table R2).

(i) Introducing highly hydratable salts in the bulk hydrogel is a facile and effective approach (Ref. 7). The water molecules with hydrogel can form hydrogel ions with salt cations/anions, and the hydrates have to break the bonds to evaporate while free water molecules evaporate naturally.

(ii) Adding to alcohols in the bulk hydrogels to deter dehydration has been demonstrated as another effective approach (Ref. 10). Owing to the low vapor pressure and ice-inhibiting effect of alcohols, both water retention capacity and anti-freezing properties can be improved. However, the presence of organic solvents like alcohols will greatly change some properties of the hydrogel, such as the electrical and mechanical properties.

(iii) Additionally, constructing a surface coating to encapsulate hydrogel is also a preferable strategy to prevent air-drying (Ref. 11). Encapsulating hydrogels with elastomers such as PDMS or Ecoflex and building a hydrophobic coating on the surface of the hydrogel can effectively inhibit hydrogel dehydration. One potential limitation of this strategy is that it is difficult to selectively modify only the exposed surface of the hydrogel without changing the internal structure and chemical composition of the hydrogel.

Table R2 Comprehensive comparisons (differences in water retention strategy) among the previously reported studies.

Product (Study object)	Fabrication Method	Water retention mechanism	Water retention	Other properties	Applications	Refs
VSNPs-PAA-Zr ⁴⁺ hydrogel	In-situ synthesis of hydrogels consisting of VSNPs and metal coordination double cross-linked between Zr ⁴⁺ and -COOH in the PAA chain	Inorganic salt, multibond network (MBN) strategy	90% (20 °C)	1. Excellent mechanical properties; 2. Good adhesive properties; 3. Ionic conductivity.	Intelligent and wearable strain sensors	Ref. 7
Double-hydrophobic-coating encapsulated hydrogel PAAm-A ₆₀ S ₆₀ O ₅₀₀	A double-hydrophobic coating was introduced to encapsulate the hydrogel to reduce water evaporation (APTES & STA Hydrophobic-polymer-coated, oil-coated).	Double-hydrophobic coating encapsulated strategy	71.9% ± 1.8% (25 °C)	1. Excellent mechanical properties.	Multifarious surface functionalization of hydrogel and broadens the range of hydrogel applications.	Ref. 10
Solketal-PPG-DMAPS hydrogel	Solketal and PPG are introduced into poly(DMAPS) hydrogels via a one-pot polymerization method.	Hydrophobic oligomeric solvents	80%~99% (25°C)	1. Excellent mechanical properties; 2. Ionic conductivity.	Intelligent and wearable strain sensors	Ref. 11

Our proposed strategies to improve water retention:

To maintain the performance of our hydrogel as e-skin, we carried out two packaging methods: top-down encapsulation and wrap-around encapsulation, and explored their water retention in different environments (4 °C, 25 °C, 40 °C). As shown in Figure R5, pure hydrogels show the weakest water retention (16.32% for 4 °C, 16.98% for 25 °C, 17.54% for 40 °C), and the top-down encapsulation hydrogel e-skins exhibit much-improved water retention in comparison with pure hydrogel, and relatively better than the wrap-around encapsulation (62.43% VS 43.50% for 4 °C, 63.05% VS 37.80% for 25 °C, 59.99% VS 37.55% for 40 °C). In our E-skin configuration, we adapted top-down encapsulation to improve the water retention of our hydrogel materials for potential long-term use.

Figure R5 Water retention properties of the CNF-DA/PAA@Fe³⁺ hydrogel. **a** Encapsulation methods of the hydrogel to improve its water retention performance: top-down encapsulation and wrap-around encapsulation. **b-d** The water loss curves of different hydrogel e-skins at varying temperatures of (b) 4 °C, (c) 25 °C, and (d) 40 °C, respectively.

References:

1. Zhao, J., et al. Hazardous gases-responsive photonic crystals cryogenic sensors based on antifreezing and water retention hydrogels. *ACS Appl. Mater. Interfaces* **15**, 42046–42055 (2023).
2. Wang, Y., et al. Highly Sensitive zwitterionic hydrogel sensor for motion and pulse detection with water retention, adhesive, antifreezing, and self-healing properties. *ACS Appl. Mater. Interfaces* **14**, 47100–47112 (2022).
3. Wang, W., et al. Physically cross-linked silk fibroin-based tough hydrogel electrolyte with exceptional water retention and freezing tolerance. *ACS Appl. Mater. Interfaces* **12**, 25353–25362 (2020).
4. Tang, J., Xing, T., Chen, S. & Feng, J. A shape memory hydrogel with excellent mechanical properties, water retention capacity, and tunable fluorescence for dual encryption. *Small*, 2305928 (2023).
5. Rajan, K., et al. Sustainable hydrogels based on lignin-methacrylate copolymers with enhanced water retention and tunable material properties. *Biomacromolecules* **19**, 2665–2562 (2018).
6. Han, X., et al. PVA/Agar interpenetrating network hydrogel with fast healing, high strength, antifreeze, and water retention. *Macromol. Chem. Phys.*, 2000237 (2020).
7. Li, Y., Liu, Y. & Xie, X. One-step in situ synthesis of tough and highly conductive ionohydrogels with water-retentive and antifreezing properties. *ACS Appl. Mater. Interfaces* **15**, 30859–30869 (2023).
8. Sun, F., et al. Dopamine/zinc oxide doped poly(N-hydroxyethyl acrylamide)/agar dual network hydrogel with super self-healing, antibacterial and tissue adhesion functions designed for transdermal patch. *J. Mater. Chem. B* **9**, 5492 (2021).
9. Zhao, L., et al. MXene-induced flexible, water-retention, semi-interpenetrating network hydrogel for ultra-stable strain sensors with real-time gesture recognition. *Adv. Sci.* **10**, 2303922 (2023).
10. Zhu, T., et al. Skin-inspired double-hydrophobic-coating encapsulated hydrogels with enhanced water retention capacity. *Adv. Funct. Mater.*, 2102433 (2021).
11. Zhao, Y., et al. Wide-humidity range applicable, anti-freezing, and healable zwitterionic hydrogels for ion-leakage-free iontronic sensors. *Adv. Mater.* **35**, 2211617 (2023).

Updates to the Revised Manuscript: According to Referee #1’s comments, we have added a more careful description to further explain the water retention of the top-down encapsulation hydrogel-e-skins in the revised manuscript on Pages 20-21.

“Specifically, a single-electrode mode PdA-TENG with a two-layer-like architecture (Fig. 6a, b, Supplementary Figs. 48 and 49) is fabricated to demonstrate the potential of the hydrogel via a

series of measurements in **long-term** self-powered e-skin devices. The CNF/PAA@Fe³⁺ hydrogel was used as the wearable substrate and ionic current collector and **top-down encapsulated by a PDMS layer to improve its water retention properties** (Supplementary Figs. 50–52 and Supplementary Movie 5).”

Updates to the Revised SI: According to Referee #1’s comments, we have added a more careful description to further explain the water retention of the top-down encapsulation hydrogel-e-skins in the revised manuscript on Pages 106-107.

“To stabilize the overall performance of our hydrogel as e-skin, we carried out two packaging methods to improve the water retention of the hydrogel materials: top-down encapsulation and wrap-around encapsulation. The water retention properties of the hydrogels were evaluated in different temperature environments (4 °C, 25 °C, 40 °C). As shown in Supplementary Fig. 52, pure hydrogels show weaker water retention (16.32% for 4 °C, 16.98% for 25 °C, 17.54% for 40 °C), and the top-down encapsulation hydrogel e-skins exhibit the excellent water retention in comparison with the wrap-around encapsulation (62.43% VS 43.50% for 4 °C, 63.05% VS 37.80% for 25 °C, 59.99% VS 37.55% for 40 °C). In our E-skin configuration, we adapted top-down encapsulation to improve the water retention of our hydrogel materials for potential long-term use.”

Referee #2 (Remarks to the Author):

The manuscript can be accepted in the current form.

Reply to the Referee: We thank Referee #2 for the suggestion of publication of our work in *Nature Communications*.

Referee #3 (Remarks to the Author):

The authors made very impressive revisions to address the comments carefully. The clarification of uniqueness of the developed strategy and the insightful discussions of the structure-property relationship were well-presented. I would like to recommend the acceptance of this interesting and inspiring work for publication in Nature Communications.

Reply to the Referee: We thank Referee #3 for the positive comments on our work and the suggestion of publication of our work in *Nature Communications*.

REVIEWERS' COMMENTS

Reviewer #1 (Remarks to the Author):

I think the authors have addressed my comments. I thus recommend publication of this work in its current version.

Itemized list of response to the reviewers' remarks
(Black: Reviewers' remarks; Blue type: Our response)

Reviewer #1 (Remarks to the Author):

Comments:

I think the authors have addressed my comments. I thus recommend publication of this work in its current version.

Reply to the Referee: We would like to thank you again for the positive comments and recommendation. Your valuable comments and suggestions have helped improve our manuscript substantially.